# Conditional Independent Component Analysis for Estimating Causal Structure with Latent Variables

**Yewei Xia**[1,5]    **Zhengming Chen**[2]    **Haoyue Dai**[3]    **Fuhong Wang**[4]    **Yixin Ren**[1]
**Yiqing Li**[5]    **Kun Zhang**[3,5,∗]    **Shuigeng Zhou**[1,∗]
[1]Fudan University          [2]Shantou University          [3]Carnegie Mellon University
[4]Guangdong University of Technology  [5]Mohamed bin Zayed University of Artificial Intelligence

## ABSTRACT

Identifying latent variables and their induced causal structure is fundamental in various scientific fields. Existing approaches often rely on restrictive structural assumptions (e.g., purity assumption) and may become invalid when these assumptions are violated. We introduce Conditional Independent Component Analysis (CICA), a new principle that extracts components that are conditionally independent given latent variables. Under mild conditions, CICA can be optimized using a tractable proxy such as rank-deficiency constraints. Building on CICA, we establish an identifiability theory for linear non-Gaussian acyclic models with latent variables: solving CICA and then applying an appropriate row permutation to the sparsest CICA solution enables recovery of the causal structure. Accordingly, we propose an estimation method based on the identifiability theory and substantiate the algorithm with experiments on both synthetic and real-world datasets.

## 1 INTRODUCTION

Understanding causal structures is essential in numerous scientific domains, such as biology (Woodward, 2010), psychology (Eronen, 2020), and economics (Hicks et al., 1980). To uncover the underlying causal structures in a data-driven manner, various methods have been proposed (Peters et al., 2017). Most traditional causal discovery methods rely on the causal sufficiency assumption (Spirtes et al., 2000), i.e., no latent confounders exist between any pair of observed variables. However, in many real-world applications, it is often infeasible to measure all the underlying causal variables. For example, in psychology, researchers investigate the impact of social behavior on mental health, while intelligence or personality may often act as latent confounders. It is difficult to precisely measure these variables, yet ignoring such latent confounders can lead to misleading conclusions. Generally, identifying the presence of latent variables and recovering the causal structure involving both observed and latent variables remains a significant challenge.

Some approaches attempt to address the challenge by exploiting conditional independence constraints, such as the FCI algorithm (Spirtes et al., 1995) and its variants (Colombo et al., 2012). However, their results capture only the causal relationships among observed variables. To further discover causal relationships between latent variables, additional parametric assumptions are typically required. For linear Gaussian causal models, several methods leverage rank-deficiency constraints to recover the underlying structure, including latent variables, up to the Markov equivalence class (Silva et al., 2006; Kummerfeld & Ramsey, 2016; Huang et al., 2022; Dong et al., 2023). To take into account higher-order statistics, (Xie et al., 2020) develops a generalized independent noise (GIN) condition and establishes its corresponding estimation algorithm for linear non-Gaussian data. TIN (Dai et al., 2022) defines the independent linear transformation subspace and its dimension can be used to further improve the identifiability of causal discovery with measurement error.

Although these methods have achieved some progress, they typically involve certain structural assumptions to simplify the problem. In particular, the purity assumption (Cai et al., 2019; Xie et al.,

---

∗Corresponding Authors

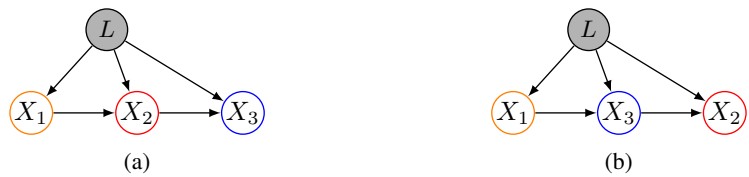

Figure 1: An example of a non-identifiability issue of most existing methods.

[2020]) rules out edges between observed variables. Violating these assumptions can lead to failures in determining the true causal graph. For example, in Fig. 1, the two graphs cannot be distinguished by most existing methods. Only a few methods can theoretically distinguish these two graphs, primarily overcomplete ICA (OICA) (Eriksson & Koivunen, 2004)-based methods and higher-order cumulant-based methods (Schkoda et al., 2024; Chen et al., 2024a). However, OICA typically relies on the expectation maximization (EM) procedure along with approximate inference, which is computationally prohibitive and prone to local optima (Cai et al., 2023). On the other hand, higher-order statistics can be very sensitive to outliers in the data (Hyvärinen & Oja, 2000), reliably estimating higher-order cumulants requires massive samples (Nikias & Mendel, 1993). This raises an important question: can we strike a better balance between identifiability and practical feasibility? Our findings indicate that this could be possible.

Concretely, by analyzing why GIN and TIN conditions fail to distinguish Fig. (1a) and (1b), we argue that relying solely on a one-sided projection $\omega^\top \mathbf{Y} \perp\!\!\!\perp \mathbf{Z}$ ($\mathbf{Y}, \mathbf{Z}$ are two subsets of observed variables) could be restricted. Instead, two-sided projections $\omega_1^\top \mathbf{Y} \perp\!\!\!\perp \omega_2^\top \mathbf{Z}$ may leave additional identifiable traces. Accordingly, we seek a unified procedure that estimates latent causal structure by searching for non-zero $\omega_1, \omega_2$ with $\omega_1^\top \mathbf{Y} \perp\!\!\!\perp \omega_2^\top \mathbf{Z}$. Motivated by this, we introduce a new principle named **conditional independent component analysis (CICA)**, which extracts components that are conditionally independent given latent variables. Under mild conditions, CICA can be optimized using a tractable proxy such as rank-deficiency constraints, which avoid involving the estimation of high-order cumulants like OICA or cumulant-based methods. Building on CICA, we establish an identifiability theory and an estimation algorithm for linear non-Gaussian acyclic models with latent variables: first solve CICA, then apply an appropriate row permutation to the sparsest CICA solution to recover the underlying causal structure, yielding a more general identifiability result with a tolerable computational burden.

**Contributions**: (1) We introduce conditional independent component analysis (CICA), a novel principle that extracts components conditionally independent given latent variables. CICA fundamentally overcomes the theoretical bottlenecks of existing methods in highly impure structures. (2) We establish an identification theory and an estimation algorithm for recovering the underlying causal structure based on CICA, where sparsity is a provable property in resolving CICA's inherent indeterminacies. (3) We validate our framework through extensive synthetic and real-world experiments.

## 2 BACKGROUND

### 2.1 PROBLEM SETUP

Suppose a directed acyclic graph (DAG) $G = (\mathbf{V}, \mathcal{E})$, where the variables $\mathbf{V}$ are generated following a linear causal structural model:

$$\mathbf{V} = \mathbf{B}\mathbf{V} + \mathbf{E}. \tag{1}$$

Here $\mathbf{V} = \mathbf{X} \cup \mathbf{L}$ contains observed variables $\mathbf{X} = \{X_i\}_{i=1}^m$ and latent variables $\mathbf{L} = \{L_i\}_{i=1}^d$. $\mathbf{E}$ are mutually independent non-Gaussian exogenous noises. The adjacency matrix $\mathbf{B}$ captures direct causal effects ($\mathbf{B}_{j,i} \neq 0 \iff V_i \to V_j \in \mathcal{E}$). Assuming observed variables do not cause latent ones ($\mathbf{B}_{\mathbf{L},\mathbf{X}} = \mathbf{0}$), $\mathbf{B}$ can be naturally partitioned into three sub-matrices $\mathbf{B}_{\mathbf{L},\mathbf{L}}$, $\mathbf{B}_{\mathbf{X},\mathbf{X}}$, and $\mathbf{B}_{\mathbf{X},\mathbf{L}}$. Specifically, $\mathbf{B}_{\mathbf{L},\mathbf{L}}$ and $\mathbf{B}_{\mathbf{X},\mathbf{X}}$ encode the direct causal effects among the latent and observed variables, respectively, while $\mathbf{B}_{\mathbf{X},\mathbf{L}}$ captures the direct causal effects from the latent to the observed one. Through simple mathematical manipulation, $\mathbf{V}$ can also be expressed as a linear mixture $\mathbf{V} = \mathbf{A}\mathbf{E}$ with the mixing matrix $\mathbf{A} = (\mathbf{I} - \mathbf{B})^{-1}$. The objective of latent causal structure learning is to fully recover the underlying causal structure $G$ from pure observational data.

**Notations.** For a matrix $M$, we denote by $M_{\mathbf{S},:}$ the rows in $M$ indexed by set $\mathbf{S}$, and similarly by $M_{:,\mathbf{S}}$ the columns. In addition, let $\mathrm{GL}(m)$[1] be the invertible matrix $\mathbf{W} \in \mathbb{R}^{m \times m}$. Further, we use $\mathrm{Pa}(V_i), \mathrm{Ch}(V_i), \mathrm{Anc}(V_i), \mathrm{De}(V_i)$ as parents, children, ancestors and descendants of $V_i$, respectively. We use $\mathrm{LPa}(\mathbf{S})$ for a subset $\mathbf{S} \subseteq \mathbf{V}$ to denote the set that contains all the common latent parents of any two nodes in $\mathbf{S}$, excluding the variables in $\mathbf{S}$. Denote equivalence relation $\mathbf{M}_1 \sim \mathbf{M}_2$ if and only if there exist a row permutation matrix $\mathbf{P}$ and a non-singular diagonal matrix $\mathbf{D}$ that makes $\mathbf{PDM}_1 = \mathbf{M}_2$. By default, $\mathbf{Y}$ and $\mathbf{Z}$ denote two subsets of observed random variables.

### 2.2 PRELIMINARIES

We first briefly review the Independent Noise (IN) condition, a fundamental principle that exploits asymmetry to discover causal directions in linear non-Gaussian acyclic models (LiNGAM) without latent variables, and then explain how it was extended to scenarios involving latent variables.

**Definition 1** (IN condition (Shimizu et al., 2011)). *Let $Y$ be a single variable and $\mathbf{Z}$ be a set of variables. Suppose all variables follow LiNGAM. We say that $(\mathbf{Z}, Y)$ follows the IN condition, if and only if the residual of regressing $Y$ on $\mathbf{Z}$ is statistically independent from $\mathbf{Z}$.*

With hidden confounders behind observed variables, independence between regressor and residual typically does not exist. However, we could still benefit from a similar idea to leverage the non-Gaussianity of exogenous noises to bring asymmetry. That is, involving other observed variables to serve as a proxy of latent variables and calculate "pseudo-residual". This finding was formalized in the Generalized Independent Noise (GIN) condition:

**Definition 2** (GIN condition (Xie et al., 2020)). *Let $\mathbf{Y}$ and $\mathbf{Z}$ be two observed random vectors. Suppose that the variables follow LiNGAM. We say $(\mathbf{Z}, \mathbf{Y})$ satisfies the GIN condition, if and only if the following two conditions are satisfied: 1) $\exists$ non-zero $\omega \in \mathbb{R}^{|\mathbf{Y}|}$ that solves the equation $\mathrm{cov}(\mathbf{Z}, \mathbf{Y})\omega = \mathbf{0}$, and 2) Any such solution $\omega$ makes the linear transformation $\omega^\top \mathbf{Y} \perp\!\!\!\perp \mathbf{Z}$.*

GIN condition needs to be equipped with enough pure children, which is defined as follows:

**Definition 3** (Purity (Xie et al., 2024)). *Let $\widetilde{\mathbf{L}}$ be a set of latent variables, and $\mathbf{S}$ be a subset of descendant nodes of $\widetilde{\mathbf{L}}$, i.e., $\mathbf{S} \subseteq \mathrm{De}(\widetilde{\mathbf{L}})$. We say $\mathbf{S}$ is a pure set relative to $\widetilde{\mathbf{L}}$ iff i) $V_a \perp\!\!\!\perp V_b | \widetilde{\mathbf{L}}$ for any $V_a, V_b \in \mathbf{S}$, and ii) $\mathbf{S} \perp\!\!\!\perp \{\mathbf{V} \setminus \mathrm{De}(\widetilde{\mathbf{L}})\} | \widetilde{\mathbf{L}}$. In addition, we say that a variable $V_c \in \mathbf{S}$ relative to $\widetilde{\mathbf{L}}$ is a pure variable if $\mathbf{S}$ is a pure set relative to $\widetilde{\mathbf{L}}$. Specifically, if $\mathbf{S} \subseteq \mathrm{Ch}(\widetilde{\mathbf{L}})$, we say that each variable $V_c \in \mathbf{S}$ is a pure child relative to $\widetilde{\mathbf{L}}$.*

In the definition of the GIN condition, the coefficient vector $\omega$ is only characterized by the covariance matrix, which may limit the power of non-Gaussianity. Therefore, TIN condition are proposed to bring more higher-order information.

**Definition 4** (TIN condition (Dai et al., 2022)). *Let $\mathbf{Z}$ and $\mathbf{Y}$ be two subsets of random variables. Denote the independent linear transformation subspace $\Omega_{\mathbf{Z};\mathbf{Y}} := \{\omega \in \mathbb{R}^{|\mathbf{Y}|} \mid \omega^\top \mathbf{Y} \perp\!\!\!\perp \mathbf{Z}\}$. The TIN condition of $\mathbf{Z}$ and $\mathbf{Y}$ is defined as: $\mathrm{TIN}(\mathbf{Z}, \mathbf{Y}) := |\mathbf{Y}| - \dim(\Omega_{\mathbf{Z};\mathbf{Y}})$, where $\dim(\Omega_{\mathbf{Z};\mathbf{Y}})$ denotes the dimension of the subspace $\Omega_{\mathbf{Z};\mathbf{Y}}$, i.e., the degree of freedom of $\omega$.*

Although TIN condition generalizes GIN condition, it is fundamentally restricted by the utility of only one-sided projections $\omega^\top \mathbf{Y} \perp\!\!\!\perp \mathbf{Z}$. As we will reveal in the next section, relying solely on such one-sided projections remains non-informative when faced with impure causal structures.

## 3 METHOD

In this section, we develop a principled framework for causal discovery in the presence of latent confounders. We first describe our motivation by analyzing why existing tools that rely on constructing independence fail (§3.1). We then formalize our proposed tool, conditional independent component analysis (CICA), and discuss its identifiability and indeterminacy (§3.2), optimization criterion (§3.3). Next, we provide a comprehensive introduction to the identifiability guarantee of latent causal structure based on CICA (§3.4). Finally, we discuss the connection between CICA and independent subspace analysis (ISA) and why ISA is not informative in our settings (§3.5).

---

[1]the general linear group of degree $n$

### 3.1 MOTIVATION: BEYOND ONE-SIDED PROJECTIONS

As discussed, existing criteria such as GIN and TIN conditions are built on one-sided projections $\omega^\top \mathbf{Y} \perp\!\!\!\perp \mathbf{Z}$. To ensure identifiability, these methods require that latent variables $\mathbf{L}$ have enough pure children (Xie et al., 2024). The rationale is that pure children are mutually conditionally independent given $\mathbf{L}$. With sufficient pure children, one can construct a linear combination of $\mathbf{Y}$ to remove the dependence entirely attributable to the common ancestors $\mathbf{L}$ and thus induce independence.

In contrast, in Fig. 1a and 1b, every pair of observed variables share not only $L$ but also $E_1$. In this case, no one-sided projection of the form $\omega^\top \mathbf{Y} \perp\!\!\!\perp \mathbf{Z}$ with non-zero $\omega$ can eliminate both sources of dependence simultaneously. As a result, the GIN and TIN conditions fail to distinguish between the two graphs since both exhibit no non-degenerate independence pattern under one-sided projections.

This limitation highlights the insufficiency of these tools based on one-sided projections in the presence of multiple latent influences. In fact, not all constructive independence patterns can be expressed as $\omega^\top \mathbf{Y} \perp\!\!\!\perp \mathbf{Z}$. A natural step forward is to consider two-sided projections of the form $\omega_1^\top \mathbf{Y} \perp\!\!\!\perp \omega_2^\top \mathbf{Z}$, to remove the dependence from both sides. The following lemma shows that the independence patterns in the form of $\omega^\top \mathbf{Y} \perp\!\!\!\perp \mathbf{Z}$ are a subset of those of $\omega_1^\top \mathbf{Y} \perp\!\!\!\perp \omega_2^\top \mathbf{Z}$.

**Lemma 1.** *Let $\mathbf{Z}$ and $\mathbf{Y}$ be two subsets of random variables. If $\omega_1^\top \mathbf{Y} \perp\!\!\!\perp \mathbf{Z}$ has a non-zero solution $\omega_1$, then there must exist a non-zero vector $\omega_2$ makes $\omega_1^\top \mathbf{Y} \perp\!\!\!\perp \omega_2^\top \mathbf{Z}$.*

Essentially, the richer the independence information that a principle exploits, the stronger its identification power. As shown next, Fig. 1a and Fig. 1b fall into different equivalent classes when using the information contained in $\omega_1^\top \mathbf{Y} \perp\!\!\!\perp \omega_2^\top \mathbf{Z}$.

**Remark 1.** *The causal structures in Fig. 1a and Fig. 1b exhibit asymmetry independence information. Specifically, non-zero solutions $(\omega_1, \omega_2)$ satisfying $\omega_{1,1} X_2 + \omega_{1,2} X_3 \perp\!\!\!\perp \omega_{2,1} X_1 + \omega_{2,2} X_2$ exist in Fig. 1a but not in Fig. 1b. Conversely, the alternative independence constraint $\omega_{1,1} X_2 + \omega_{1,2} X_3 \perp\!\!\!\perp \omega_{2,1} X_1 + \omega_{2,2} X_3$ admits non-zero solutions in Fig. 1b, but not in Fig. 1a.*

Motivated by these asymmetries, although two causal graphs cannot be distinguished using only a one-sided projection $\omega^\top \mathbf{Y} \perp\!\!\!\perp \mathbf{Z}$, two-sided projections $\omega_1^\top \mathbf{Y} \perp\!\!\!\perp \omega_2^\top \mathbf{Z}$ can leave additional identifiable traces for the causal direction. This prompts a natural question: Can we develop a unified procedure that searches for non-zero $\omega_1, \omega_2$ with $\omega_1^\top \mathbf{Y} \perp\!\!\!\perp \omega_2^\top \mathbf{Z}$ to enhance identifiability?

### 3.2 CONDITIONAL INDEPENDENT COMPONENT ANALYSIS

A direct route to construct $\omega_1^\top \mathbf{Y} \perp\!\!\!\perp \omega_2^\top \mathbf{Z}$ is to use overcomplete ICA (OICA), which separates more mutually independent sources from fewer observed signals. However, OICA is known to be computationally and statistically ineffective (Ding et al., 2019). An alternative solution is to brute-force searching for each "two-sided projection" $(\omega_1, \omega_2)$. However, it is difficult to guarantee that all feasible $(\omega_1, \omega_2)$ have been found.

Instead of fully separating all latent sources as in OICA, we propose to factor out the shared influences explicitly and only require independence conditional on a latent vector. Concretely, we seek an invertible transform $\mathbf{W}$ such that $\mathbf{Z} = \mathbf{W}\mathbf{X}$ has mutually independent coordinates given some latent $\mathbf{L} \in \mathbb{R}^p$. This approach is powerful for two reasons: 1. As we will show in Section 3.3, when $p$ is known, this principle allows for more tractable optimization proxies, avoiding the statistical and computational burdens of OICA. 2. As we will prove in Lemma 3, any solution that satisfies this generative principle $(Z_i \perp\!\!\!\perp Z_j | \mathbf{L})$ provably induces the two-sided projections $\omega_1^\top \mathbf{Y} \perp\!\!\!\perp \omega_2^\top \mathbf{Z}$ required for identifiability. We formalize this core generative principle as follows:

**Assumption 1** (Linear mixing with conditionally independent sources). *Let $\mathbf{X}$ be an observed variable set with $|\mathbf{X}| = m$. There exist a constant invertible matrix $\mathbf{A} \in \mathbb{R}^{m \times m}$, $p$ latent variables $\mathbf{L}$ with $\Sigma_\mathbf{L} \succ 0$, a constant matrix $\mathbf{M} \in \mathbb{R}^{m \times p}$, and noise variables $\mathbf{E} = (E_1, \dots, E_m)^\top$ such that*

$$\mathbf{X} = \mathbf{A}\mathbf{S}, \qquad \mathbf{S} = \mathbf{M}\mathbf{L} + \mathbf{E}, \qquad \mathbf{E} \perp\!\!\!\perp \mathbf{L}. \qquad (2)$$

*$\{E_i\}_{i=1}^m$ are mutually independent with finite, non-zero variances, and at most one $E_i$ is Gaussian. $\Sigma_\mathbf{E}$ is not a scalar multiple of the identity matrix $\mathbf{I} \in \mathbb{R}^{m \times m}$.*

**Definition 5** ($p$-order Conditional Independent Component Analysis (CICA)). *Let $\mathbf{X}$ be $m$ observed variables. An matrix $\mathbf{W} \in \mathrm{GL}(m)$ is called a $p$-order CICA solution for $\mathbf{X}$ if there exists $p$ latent*

variables $\mathbf{L}$ s.t., the components of $\mathbf{Z} := \mathbf{WX} = (Z_1, \ldots, Z_m)^\top$ are mutually conditionally independent given $\mathbf{L}$. Besides, we introduce $p_{min}(\mathbf{X}) := \min\{k : k \in \mathbb{N}, k\text{-order CICA solution of } \mathbf{X}$ exists$\}$ to measure the size of the minimal latent conditional set of $\mathbf{X}$.

When $p = 0$, conditionally independent reduces to mutual independence of $\mathbf{Z}$, and then CICA coincides with ICA, where the unmixing matrix can be identified up to a permutation and scaling equivalence class (Comon, 1994). In CICA, does the introduction of latent conditional variables create new, complex mixing ambiguities? The following lemma formally establishes the results.

**Lemma 2** (Identifiability and Indeterminacy of CICA). *Suppose Assumption 1 holds, let $\mathbf{X}$ be $m$ observed variables. If there exist $p_{min}(\mathbf{X})$ latent variables $\mathbf{L}$ such that for two valid CICA solutions $\mathbf{W}_1$ and $\mathbf{W}_2$, the components of $\mathbf{Z}^{(k)} := \mathbf{W}_k\mathbf{X}$ are mutually conditionally independent given $\mathbf{L}$ for $k \in \{1, 2\}$, then there uniquely exist a permutation matrix $\mathbf{P}_\pi$ (for some permutation $\pi$ of $[m]$) and a non-singular diagonal matrix $\mathbf{D}$ such that $\mathbf{W}_2 = \mathbf{P}_\pi \mathbf{D} \mathbf{W}_1$.*

In particular, when $p = 0$ (the ICA case), the conclusion reduces to the classical permutation and scaling indeterminacy of ICA. Therefore, Lemma 2 tells us that CICA introduces an additional indeterminacy about the conditional set $\mathbf{L}$ compared to ICA. In addition, based on the CICA solution, one can naturally induce two-sided projections $\omega_1^\top \mathbf{Y} \perp\!\!\!\perp \omega_2^\top \mathbf{Z}$.

**Lemma 3.** *Let $\mathbf{X}$ be $m$ observed variables, and $\mathbf{W}$ be a $p$-order CICA solution of $\mathbf{X}$. Let $\mathbf{X}' = \mathbf{WX}, \mathbf{Y}$ and $\mathbf{Z}$ are two subsets of $\mathbf{X}'$, then if $\max\{|\mathbf{Y}|, |\mathbf{Z}|\} > p$, $\omega_1^\top \mathbf{Y}' \perp\!\!\!\perp \omega_2^\top \mathbf{Z}'$ has a non-zero solution $(\omega_1, \omega_2)$ for $(\mathbf{Y}', \mathbf{Z}')$, where $\mathbf{Y}' = \{X_i | \sum_{X_k \in \mathbf{Y}} \mathbf{W}_{k,i} \neq 0\}$, $\mathbf{Z}'$ are defined similarly.*

**Example 1.** *The following structural causal model serves as an instantiation of Fig. 1a, where $L, E_1, E_2, E_3$ are independent non-Gaussian variables, $a, b, c, u, v$ are non-zero coefficients. The identity matrix $\mathbf{I} \in \mathrm{GL}(3)$ is a 3-order CICA solution of $\mathbf{X}$ (the conditional set can be $\{L, E_1, E_2\}$). The right-hand side below shows an example of a 1-order CICA solution of $\mathbf{X}$ (the conditional set is $\{L\}$). The existence of $L$ leads to the absence of a 0-order CICA (i.e., ICA) solution of $\mathbf{X}$.*

$$\begin{cases} X_1 = aL + E_1, \\ X_2 = bL + uX_1 + E_2, \\ X_3 = cL + vX_2 + E_3. \end{cases} \qquad \overbrace{\begin{bmatrix} 1 & 0 & 0 \\ -u & 1 & 0 \\ 0 & -v & 1 \end{bmatrix}}^{\mathbf{W}} \begin{bmatrix} X_1 \\ X_2 \\ X_3 \end{bmatrix} = \begin{bmatrix} a \\ b \\ c \end{bmatrix} L + \begin{bmatrix} E_1 \\ E_2 \\ E_3 \end{bmatrix}$$

*Besides, we can construct two-sided projections $\omega_1^\top \mathbf{Y} \perp\!\!\!\perp \omega_2^\top \mathbf{Z}$ with non-zero $\omega_1, \omega_2$, based on the CICA solution of $\mathbf{X}$. Taking $\mathbf{Y} = \{X_1', X_2'\}$, $\mathbf{Z} = \{X_3'\}$ as an example, denoting $\mathbf{X}' = \mathbf{WX}$, then we have $aX_2' - bX_1' \perp\!\!\!\perp X_3'$. i.e., $-(au + b)X_1 + bX_2 \perp\!\!\!\perp X_3 - vX_2$. A non-zero solution $\omega_1 = [-(au + b), b]^\top$, $\omega_2 = [-v, 1]^\top$ exists for $(\mathbf{Y}' = \{X_1, X_2\}, \mathbf{Z}' = \{X_2, X_3\})$.*

## 3.3 Optimization Criterion for CICA

Since the conditional set is latent, the definition of CICA does not specify a testable optimization objective. A practical question arises: which optimization criterion should we use for CICA? Inspired by (Huang et al., 2022; Dong et al., 2023), we characterize conditional independence by introducing the following rank-deficiency constraint.

**Lemma 4.** *For an observed variable set $\mathbf{X}$ with $|\mathbf{X}| = m$, denote $p = p_{min}(\mathbf{X})$. Suppose $m \geq 2p + 2$, and set $\mathbf{X}' := \mathbf{WX}$, then $\mathbf{W}$ is a $p$-order CICA solution of $\mathbf{X}$ if and only if for every pair of disjoint coordinate subsets $\mathbf{X_1}, \mathbf{X_2}$ of $\mathbf{X}'$ with $|\mathbf{X_1}| = |\mathbf{X_2}| = p + 1$, $\det(\Sigma_{\mathbf{X_1}, \mathbf{X_2}}) = 0$, where $\Sigma := \mathrm{Cov}(\mathbf{X}')$ denotes the covariance matrix on $\mathbf{X}'$ and $\Sigma_{\mathbf{X_1}, \mathbf{X_2}}$ is the $(p+1) \times (p+1)$ sub-matrix of $\Sigma$ with rows indexed by $\mathbf{X_1}$ and columns by $\mathbf{X_2}$.*

In fact, here $m \geq 2p + 2$ is not a strict restriction; we can relax it by replacing the covariance matrix with a higher-order cumulant tensor. More details are included in Appendix B.3. When $p_{min}(\mathbf{X}) = 1$, we can use another proxy objective of CICA, equipped with a weaker condition.

**Definition 6** (Triad constraint (Cai et al., 2019)). *Define the pseudo-residual of $\{X_i, X_j\}$ relative to $X_k$ as $E_{(i,j|k)} := \mathrm{Cov}(X_j, X_k) \cdot X_i - \mathrm{Cov}(X_i, X_k) \cdot X_j$. We say that the pair of variables $\{X_i, X_j\}$ and $X_k$ satisfy the Triad constraint if $E_{(i,j|k)} \perp\!\!\!\perp X_k$.*

**Lemma 5.** *For an observed variable set $\mathbf{X}$ with $|\mathbf{X}| = m$, suppose that $p_{min}(\mathbf{X}) = 1$ and $m \geq 3$ hold, set $\mathbf{X}' \triangleq \mathbf{WX}$, then the invertible matrix $\mathbf{W}$ is a 1-order CICA solution of $\mathbf{X}$ if and only if for every ordered triple $(X_i', X_j', X_k')$ of $\mathbf{X}'$, $\{X_i', X_j'\}$ and $X_k'$ satisfies the Triad constraint.*

In both Lemma 4 and 5, we assume $p_{min}(\mathbf{X})$ is known, then characterize $p_{min}(\mathbf{X})$-order CICA using the zero-determinant and independence constraint, respectively. In our estimation algorithm, we can determine the value of $p_{min}(\mathbf{X})$ in principle, without requiring prior knowledge (see Lemma 11). Since both the determinant and dependence measures (e.g., HSIC (Gretton et al., 2005)) used in Def. 6 are differentiable, these lemmas actually provide an optimization criterion for CICA.

### 3.4 IDENTIFIABILITY OF LATENT CAUSAL STRUCTURE BASED ON CICA

In this section, we establish an identifiability theory for causal structure in the linear non-Gaussian acyclic models with latent variables. Once CICA is solved, when and how can the causal structure be recovered from the CICA solutions $\mathbf{W}$? First, we have the following basic assumptions.

**Assumption 2** (Rank Faithfulness Assumption (Spirtes, 2013))**.** *We assume that the distribution $P$ is (linearly) rank-faithful to a DAG $G$ if every rank constraint on a sub-covariance matrix that holds in $P$ is entailed by every free-parameter linear structural model with a path diagram equal to $G$.*

Assumption 2 holds generically, since the set of values of the free parameters of the SCM for which the rank is not faithful is of Lebesgue measure 0 (Spirtes, 2013). Besides, recovering the latent structure requires the latent variables to leave a sufficient structural footprint in the observed variables, we then introduce the following standard requirement:

**Condition 1.** *Each latent variable in $G$ has at least three neighbors and two children (which can be latent or observed).*

In this section, for the sake of brevity, we will primarily discuss the results under the one-factor scenario (observed variables share at most one common latent confounder). Most results can be extended into the multi-factor scenario (observed variables can share multiple latent variables) directly. Since directly recovering the global causal structure is inherently challenging due to the entanglement of causal effects from latent and observed variables, we decouple the objective by first recovering the causal structure among the observed variables $\mathbf{B_{X,X}}$. As we will demonstrate, our CICA framework exactly isolates $\mathbf{B_{X,X}}$ from latent confounding. The following lemma shows that $\mathbf{B_{X,X}}$ naturally constitutes a valid CICA solution when conditioning on the true latent confounders.

**Lemma 6.** $\mathbf{I} - \mathbf{B_{X,X}}$ *is a $p_{min}(\mathbf{X})$-order CICA solution of $\mathbf{X}$ with latent conditional set $\mathrm{LPa}(\mathbf{X})$.*

To identify the causal structure based on CICA, we must resolve all inherent indeterminacies. Inspired by (Shimizu et al., 2006), the permutation and scaling indeterminacies can be eliminated by the acyclicity of the causal graph. Therefore, we have the following lemma.

**Lemma 7.** *Suppose $\mathbf{W}$ is a $p_{min}(\mathbf{X})$-order CICA solution of $\mathbf{X}$ whose latent conditional set is $\mathrm{LPa}(\mathbf{X})$, then $\mathbf{W} \sim \mathbf{I} - \mathbf{B_{X,X}}$ and we can uniquely recover the true causal matrix among observed variables $\mathbf{B_{X,X}}$ from $\mathbf{W}$.*

Based on Lemma 6 and Lemma 7, we can correctly recover $\mathbf{B_{X,X}}$ provided that the CICA solution whose latent conditional set is $\mathrm{LPa}(\mathbf{X})$. As stated in Lemma 2, CICA introduces an additional indeterminacy in the choice of the latent conditional set. If $\mathbf{W}$ is a $p_{min}(\mathbf{X})$-order CICA solution of observed variables $\mathbf{X}$, the conditional set does not need to coincide with the latent confounders $\mathrm{LPa}(\mathbf{X})$. Instead, it may correspond to the exogenous noise of the observed variables. Therefore, to solve the indeterminacy of the latent conditional set, we must further identify the CICA solution that aligns with the ground-truth causal structure.

As shown in Fig. 2, for $\mathbf{X} = \{X_1, X_2, X_3\}$, $\mathbf{W_1}$ is a 1-order CICA solution of $\mathbf{X}$ given $L$, thus $\mathbf{W_1} \sim \mathbf{I} - \mathbf{B_{X,X}}$ according to Lemma 2. In contrast, the ambiguity of the latent conditional set allows alternative solutions, such as $\mathbf{W_2}$, to also qualify as feasible 1-order CICA solutions of $\mathbf{X}$, although without direct correspondence to $\mathbf{B_{X,X}}$. Essentially, $\mathbf{W_2 X}$ can be interpreted as swapping the roles of $L$ and $E_1$ on $\mathbf{W_1 X}$. Although conditional independence is preserved after swapping the latent variables, the sparsity of the solution matrix changes. Specifically, it becomes denser. This observation highlights that sparsity can serve as an additional discriminative signal: the sparsest CICA solution better aligns with the underlying causal structure.

**Lemma 8.** $\mathbf{I} - \mathbf{B_{X,X}} \in \arg\min\{\|\mathbf{W}\|_0 : \mathbf{W}$ *is a $p_{min}(\mathbf{X})$-order CICA solution of $\mathbf{X}\}$.*

Lemma 8 shows that $\mathbf{I} - \mathbf{B_{X,X}}$ is a $p_{min}(\mathbf{X})$-order CICA solution of $\mathbf{X}$ with the minimum number of non-zero entries. Notably, we do not assume that real-world causal structure is maximally sparse. On

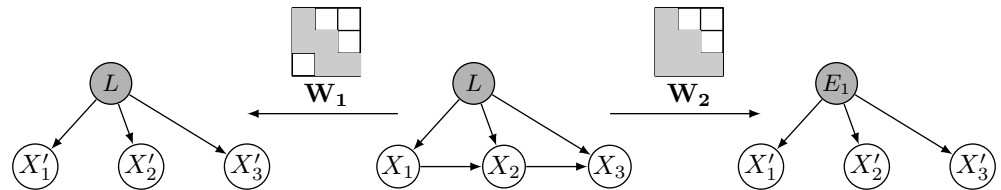

Figure 2: An example of different 1-order CICA solutions for $\mathbf{X}$. $\mathbf{W_1}$ is a 1-order CICA solution that renders $\mathbf{X}'$ conditionally independent given $L$, while $\mathbf{W_2}$ renders $\mathbf{X}'$ conditionally independent given $E_1$, the exogenous noise of $X_1$. The gray/white rectangle denotes non-zero/zero entries.

the contrary, it can be arbitrarily dense. The minimal sparsity principle is not a prior assumption or a convenient choice we impose on the causal structure. Instead, it is a provable theoretical property that emerges from the CICA framework itself, which we then exploit for identifiability. To ensure identifiability, we seek conditions under which $\mathbf{I} - \mathbf{B_{X,X}}$ is the unique sparsest $p_{min}(\mathbf{X})$-order CICA solution of $\mathbf{X}$, up to some permutation and scale indeterminacies.

**Condition 2.** *For any $X_i \in \mathbf{X}$, $\exists X_j \in \mathbf{X} \setminus \{X_i\}$ with $\mathrm{LPa}(\{X_i, X_j\}) \neq \emptyset$, $X_i \nrightarrow X_j$.*

Intuitively, Condition 2 ensures that every observed variable has at least one "confounded sibling" that it does not directly cause, making it possible to disentangle the latent variable's effect uniquely.

**Example 2.** *In the figure on the left below, since $X_1$ is not the parent of $X_3$, $X_2$ and $X_3$ are not the parents of $X_1$, then Condition 2 holds. In contrast, in the figure on the right below, $X_1$ is both the parent of $X_2$ and $X_3$, thus Condition 2 does not hold.*

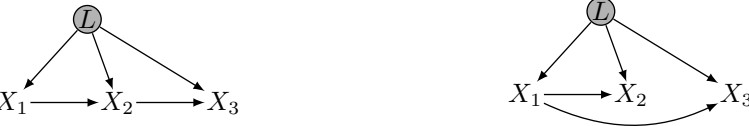

Figure 3: Two example graphs to understand Condition 2.

**Lemma 9.** *If Condition 2 holds, $\mathbf{W} \in \arg\min\{\|\tilde{\mathbf{W}}\|_0 : \tilde{\mathbf{W}} \text{ is a } p_{min}(\mathbf{X})\text{-order CICA solution of} \mathbf{X}\}$ if and only if $\mathbf{W} \sim \mathbf{I} - \mathbf{B_{X,X}}$.*

Under Condition 2, Lemma 9 establishes that the sparsest $p_{min}(\mathbf{X})$-order CICA solution recovers $\mathbf{I} - \mathbf{B_{X,X}}$ up to permutation and scale indeterminacies. By Lemma 7, the remaining gap can be eliminated by acyclicity. Consequently, $\mathbf{B_{X,X}}$ is uniquely identified, including both the causal graph among the observed variables and its edge coefficients.

Conversely, when Condition 2 does not hold, $\mathbf{I} - \mathbf{B_{X,X}}$ is non-identifiable: there exists a distinct $p_{min}(\mathbf{X})$-order CICA solution $\mathbf{W}'$ with an equal number of non-zero entries. Surprisingly, although $\mathbf{W}'$ has different parameters from $\mathbf{I} - \mathbf{B_{X,X}}$, their support matrix remains the same. Therefore, the causal structure among observed variables is identifiable, which we summarized as follows.

**Theorem 1.** *All latent variables in $\mathrm{LPa}(\mathbf{X})$ can be identified. Besides, the causal edges of $\mathrm{LPa}(\mathbf{X})$ to $\mathbf{X}$ and the causal edges between the observed variables are also identifiable.*

When the variables form a hierarchical structure and some latent variables may have no observed children, due to the linearity assumption and the transitivity of linear causal relations, we can use a certain observed descendant of the latent variables to implement CICA and apply Theorem 1 recursively. The question is, which one is suitable to serve as a surrogate for the latent variable?

**Lemma 10.** *Let $L$ be a latent variable discovered in the current iteration. Denote $\mathbf{S} = \mathrm{Ch}(L)$. Let $S_k$ have the highest causal order in $\mathbf{S}$ whose index in $\mathbf{S}$ is $k$, and $\mathbf{W}$ be the sparsest $p_{min}(\mathbf{S})$-order CICA solution of $\mathbf{S}$. $\mathbf{P}$ is the permutation matrix that makes $\mathbf{PW}$ have non-zero diagonal elements, simultaneously. Denote $\mathbf{Z} = \mathbf{PWS}$, then the value of $Z_k$ can be a suitable surrogate for $L$.*

**Example 3.** *Taking Fig. 1a as an example, denote $\mathbf{W}$ as the sparsest 1-order CICA solution of $\mathbf{X} = \{X_1, X_2, X_3\}$, $\mathbf{P}$ is the permutation matrix that makes $\mathbf{PW}$ whose diagonal elements have non-zero values, simultaneously. Let $\mathbf{Z} = \mathbf{PWX}$. As $X_1$ is the variable that has the highest causal order in $\mathrm{Ch}(L)$, then we can take the value of $Z_1$ as the surrogate of $L$.*

Equipped with the surrogate selection strategy provided by Lemma 10, the identifiability guarantee of Theorem 1 can be naturally generalized to a whole hierarchical structure.

**Theorem 2.** *If Condition 1 holds, then the underlying causal graph $G$ is fully identifiable, including both the location of latent variables $\mathbf{L}$ and the recovery of causal relationships $\mathcal{E}$.*

Based on the identifiability guarantee, we develop an estimation algorithm named CICA-LiNGAM to recover the latent causal structure from the CICA solution. Suppose that some observed variables $\mathbf{S}$ form a causal cluster, we can determine the value of $p_{min}(\mathbf{S})$ using the GIN condition. Here we say that an observed variable set $\mathbf{S}$ is a causal cluster if the variables in $\mathbf{S}$ partially share the same latent parents that satisfy $\mathbf{S} = \mathrm{Ch}(\mathrm{LPa}(\mathbf{S}))$, or $\mathrm{LPa}(\mathbf{S})$ $d$-separates $\mathbf{S}$ and $\mathrm{Ch}(\mathrm{LPa}(\mathbf{S})) \setminus \mathbf{S}$. The causal cluster serves as a basic unit that helps us quickly locate the latent variables. The following lemma states a basic criterion for identifying causal clusters from active variables $\mathbf{A}$ (active variables contain some variables that may form causal clusters in the bottom-up recursive procedure).

**Lemma 11** (Identifying Causal Clusters (Xie et al., 2022)). *Let $\mathbf{A}$ be the active variable set and $\mathbf{S} \subset \mathbf{A}$. Then $\mathbf{S}$ is a causal cluster with $|\mathrm{LPa}(\mathbf{S})| = p_{min}(\mathbf{S}) = 1$ if: 1) for any subset $\tilde{\mathbf{S}}$ of $\mathbf{Y}$ with $|\tilde{\mathbf{S}}| = 2, (\mathbf{A} \setminus \mathbf{S}, \tilde{\mathbf{S}})$ follows the GIN condition, and 2) no proper subset of $\mathbf{S}$ satisfies 1).*

---

**Algorithm 1** CICA-LiNGAM

---

**Require:** Observed variables $\mathbf{X}$.
**Ensure:** Fully identified causal structure $G$ on $\mathbf{X}$ and discovered latent variables.
 1: Initialize active variable set $\mathbf{A} = \mathbf{X}$ and $G = \emptyset$.
 2: **while** $\mathbf{A} \neq \emptyset$ **do**
 3:     Identify causal clusters in the current active variable set $\mathbf{A}$ (Lemma 11).
 4:     Obtain the sparsest CICA solution $\mathbf{W}$ of each cluster (Lemma 4 or 5).
 5:     Find a permutation matrix $\mathbf{P}$ to make the diagonal elements of $\mathbf{PW}$ non-zero (Lemma 7).
 6:     Obtain causal structure within a causal cluster (Theorem 1).
 7:     Merge clusters share the common latent parent (Proposition 1 in Appendix B).
 8:     Determine whether new latent variables should be introduced (Corollary 2 in Appendix B).
 9:     Update the active variable set $\mathbf{A}$ according to Lemma 10.
10: **end while**
11: Return $G$.

---

The algorithm adopts a recursive procedure (see Algorithm 1 for pseudo code). In each iteration, it performs four steps: i) identify causal clusters (line 3); ii) infer the causal structure within each cluster based on the sparsest CICA solution (lines 4∼6); iii) merge the clusters share the common latent parent and determine how many new latent variables are required in the current iteration (lines 7∼8, details see Appendix B); and iv) update the active variable set accordingly (line 9).

## 3.5   Connection with Independent Subspace Analysis

Local ISA-LiNG (Dai et al., 2024) leverages independent subspace analysis (ISA) instead of OICA for local causal discovery. Inspired by this, we then ask whether ISA remains a suitable surrogate of OICA in the presence of latent confounders and what the relationship is between CICA and ISA. To answer these questions, we first review the basic terminology of ISA.

**Definition 7** (Irreducible). *An $m$-dim random vector $\mathbf{Z}$ is irreducible if it contains no lower-dim independent components. In other words, no invertible matrix $\mathbf{W} \in \mathrm{GL}(m)$ can decompose $\mathbf{WZ} = (\mathbf{Z}_1', \mathbf{Z}_2')^\top$ into $\mathbf{Z}_1' \perp\!\!\!\perp \mathbf{Z}_2'$.*

**Definition 8** (ISA solution (Theis, 2006)). *For an $m$-dim random vector $\mathbf{X}$, an invertible matrix $\mathbf{W}$ is called an independent subspace analysis (ISA) solution of $\mathbf{Y}$ if $\mathbf{WX} = (\mathbf{Z}_1^\top, \ldots, \mathbf{Z}_k^\top)^\top$ consists of mutually independent, irreducible random vectors $\mathbf{Z}_i$. The corresponding partition $\mathbf{\Gamma_W}$ of the indices $[m]$ is called the ISA partition associated with $\mathbf{W}$.*

Although ISA seeks separation "as independent as possible", the following theorem shows that ISA is actually not informative enough in the presence of latent confounders.

**Theorem 3** (Interpretations of ISA in LiNGAM model). *Let the graph obtained after removing all the outgoing edges of $\mathbf{X}$ in $\mathcal{G}$ be named by $\mathcal{G}'$, which form several connected components of observed*

variables $\mathbf{X}'_{C_1}, \mathbf{X}'_{C_2}, \cdots, \mathbf{X}'_{C_k}$, *where $k$ be the number of connected components in $\mathcal{G}'$. For an ISA solution $\mathbf{W}$, let $\mathbf{WX} = (\mathbf{Z}_1^\top, \ldots, \mathbf{Z}_k^\top)^\top$. Then there is a permutation $\pi$ of $[k]$ s.t. for any $i \in [k]$, $\exists \mathbf{W}_i \in \mathrm{GL}(|C_i|)$ makes $\mathbf{Z}_{\pi(i)} = \mathbf{W}_i \mathbf{X}'_{C_i}$.*

**Example 4.** *Here we present a concrete example to aid in understanding Theorem 3. After removing all outgoing edges of $\mathbf{X}$ in $\mathcal{G}$ (the graph in Fig. 4a), $\mathcal{G}'$ (the graph in Fig. 4b) form three connected components of observed variables, $\{X'_1\}$, $\{X'_2, X'_4\}$ and $\{X'_3, X'_5\}$. Then $\mathbf{WX} = (\mathbf{Z}_1^\top, \mathbf{Z}_2^\top, \mathbf{Z}_3^\top)^\top, \exists \pi, \mathbf{W}_1, \mathbf{W}_2, \mathbf{W}_3, \text{ s.t. } \mathbf{Z}_{\pi(1)} = \mathbf{W}_1 X'_1, \mathbf{Z}_{\pi(2)} = \mathbf{W}_2 X'_{[2,4]}, \mathbf{Z}_{\pi(3)} = \mathbf{W}_3 X'_{[3,5]}.$*

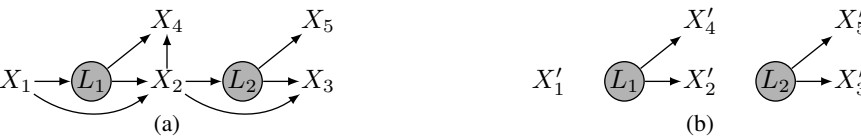

Figure 4: An example to understand the procedure of ISA in the LiNGAM model.

According to Lemma 3, any matrix $\mathbf{W} \in \mathrm{GL}(3)$ is an ISA solution of Fig. 1a and 1b. Therefore, they are "ISA equivalent", which we summarize in the following remark.

**Remark 2.** *The two causal graphs in Fig. 1a and Fig. 1b cannot be identified by ISA.*

The fundamental reason why ISA fails to be informative in the presence of latent confounders is that, although it seeks components that are "as independent as possible", ISA does not impose constraints within each irreducible subspace. Consequently, regardless of how variables are connected within a subspace, the corresponding graphs belong to the same equivalence class under ISA. In contrast, the absence of constraints within each subspace can be addressed by CICA. For example, the sparsest 1-order CICA solution on $\{X_2, X_4\}$ makes the edge $X_2 \to X_4$ identifiable.

## 4 EXPERIMENTS

In this section, we first present simulation studies on synthetic data to demonstrate that our algorithm effectively identifies causal structure. We also conduct a simulation study to quantify the gap between the sparsest and the alternative CICA solution to demonstrate the stability of CICA for practical use. Due to space limitations, real-world experiments are presented in Appendix C.

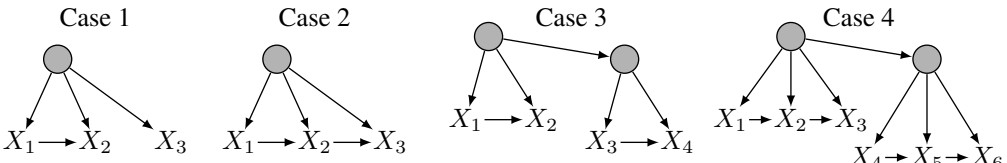

Figure 5: Causal structures used in synthetic experiments.

**Dataset, Baselines, Metrics**. We generate data from some typical graph structures that satisfy Condition 1 (see Fig. 5). We consider different sample sizes $N = \{5k, 10k, 20k\}$. The causal strengths $B_{i,j}$ are generated uniformly from $[-2, -0.5] \cup [0.5, 2]$, and the non-Gaussian noise terms are generated from the square of exponential distributions. In each setting, the results are obtained after averaging the values in the 10 tests. We report both the average results and standard errors. We consider the following four methods as baselines for comparing: RLCD (Dong et al., 2023), PO-LiNGAM (Jin et al., 2023), LaHME (Xie et al., 2024), CDHS (Li et al., 2025). To evaluate the precision of the estimated graph, we used the following three metrics as (Li & Liu, 2025). 1) Error in Latent Variables: the absolute difference between the estimated number of latent variables and the ground-truth one; 2) Correct Ordering Rate: the number of correctly estimated causal orderings divided by that of ground-truth causal orderings; 3) F1 score of causal edges.

**Results and Analysis**. The experimental results are summarized in Table 1. For CDHS, the algorithm fails in the fully impure setting as its "Homologous Surrogates" condition (Li et al., 2025) is violated, preventing any valid output. While LaHME and PO-LiNGAM are relatively stable on key evaluation metrics, they are unable to produce correct results in fully impure scenarios (e.g., cases 2 and 4) because their clustering step fails. RLCD is inapplicable to cases 1 and 2, as its

Table 1: Comparison on synthetic data. ↑ means higher is better while ↓ means lower is better.

| Graph | Method | Error in Latent Variables ↓ | | | Correct-Ordering Rate ↑ | | | F1-Score ↑ | | |
|---|---|---|---|---|---|---|---|---|---|---|
| | | 5k | 10k | 20k | 5k | 10k | 20k | 5k | 10k | 20k |
| Case 1 | CDHS | 0.30±0.46 | 0.20±0.40 | 0.40±0.49 | **0.65±0.45** | **0.80±0.40** | 0.60±0.49 | 0.67±0.45 | **0.80±0.40** | 0.60±0.49 |
| | LaHME | 0.00±0.00 | 0.10±0.30 | 0.00±0.00 | 0.50±0.00 | 0.45±0.15 | 0.50±0.00 | 0.67±0.00 | 0.60±0.20 | 0.67±0.00 |
| | PO-LiNGAM | 0.00±0.00 | 0.00±0.00 | 0.00±0.00 | 0.50±0.00 | 0.50±0.00 | 0.50±0.00 | 0.66±0.03 | 0.67±0.00 | 0.67±0.00 |
| | RLCD | 1.00±0.00 | 1.00±0.00 | 1.00±0.00 | 0.00±0.00 | 0.00±0.00 | 0.00±0.00 | 0.00±0.00 | 0.00±0.00 | 0.00±0.00 |
| | Ours | **0.00±0.00** | **0.00±0.00** | **0.00±0.00** | 0.65±0.25 | 0.60±0.35 | **0.75±0.25** | **0.75±0.25** | 0.67±0.38 | **0.77±0.46** |
| Case 2 | CDHS | 1.00±0.00 | 1.00±0.00 | 1.00±0.00 | 0.00±0.00 | 0.00±0.00 | 0.00±0.00 | 0.00±0.00 | 0.00±0.00 | 0.00±0.00 |
| | LaHME | 1.00±0.00 | 1.00±0.00 | 1.00±0.00 | 0.00±0.00 | 0.00±0.00 | 0.00±0.00 | 0.00±0.00 | 0.00±0.00 | 0.00±0.00 |
| | PO-LiNGAM | 1.00±0.00 | 1.00±0.00 | 1.00±0.00 | 0.00±0.00 | 0.00±0.00 | 0.00±0.00 | 0.00±0.00 | 0.00±0.00 | 0.00±0.00 |
| | RLCD | 1.00±0.00 | 1.00±0.00 | 1.00±0.00 | 0.00±0.00 | 0.00±0.00 | 0.00±0.00 | 0.00±0.00 | 0.00±0.00 | 0.00±0.00 |
| | Ours | **0.00±0.00** | **0.00±0.00** | **0.00±0.00** | 0.60±0.25 | 0.60±0.25 | 0.66±0.27 | 0.67±0.44 | 0.67±0.44 | 0.72±0.48 |
| Case 3 | CDHS | 2.00±0.00 | 1.90±0.30 | 2.00±0.00 | 0.00±0.00 | 0.00±0.00 | 0.00±0.00 | 0.00±0.00 | 0.00±0.00 | 0.00±0.00 |
| | LaHME | 0.00±0.00 | 0.20±0.60 | 0.10±0.30 | 0.44±0.00 | 0.40±0.13 | 0.40±0.13 | 0.73±0.00 | 0.65±0.22 | 0.65±0.22 |
| | PO-LiNGAM | 0.00±0.00 | **0.00±0.00** | 0.20±0.60 | 0.44±0.00 | 0.44±0.00 | 0.40±0.13 | 0.73±0.00 | **0.73±0.00** | 0.65±0.22 |
| | RLCD | 0.10±0.30 | 0.10±0.30 | **0.00±0.00** | 0.60±0.25 | 0.60±0.25 | 0.58±0.16 | 0.70±0.24 | 0.70±0.24 | 0.73±0.08 |
| | Ours | **0.00±0.00** | 0.20±0.60 | 0.10±0.00 | **0.66±0.18** | **0.61±0.31** | **0.61±0.31** | **0.78±0.31** | 0.72±0.35 | **0.78±0.31** |
| Case 4 | CDHS | 2.00±0.00 | 2.00±0.00 | 2.00±0.00 | 0.00±0.00 | 0.00±0.00 | 0.00±0.00 | 0.00±0.00 | 0.00±0.00 | 0.00±0.00 |
| | LaHME | 0.25±0.54 | 0.20±0.40 | 0.10±0.44 | 0.30±0.15 | 0.30±0.15 | 0.36±0.08 | 0.56±0.28 | 0.56±0.28 | 0.67±0.15 |
| | PO-LiNGAM | 2.00±0.00 | 2.00±0.00 | 2.00±0.00 | 0.00±0.00 | 0.00±0.00 | 0.00±0.00 | 0.00±0.00 | 0.00±0.00 | 0.00±0.00 |
| | RLCD | 0.50±0.81 | 1.10±0.83 | 0.70±0.90 | 0.28±0.19 | 0.11±0.17 | 0.20±0.17 | 0.30±0.22 | 0.13±0.20 | 0.23±0.20 |
| | Ours | **0.25±0.54** | **0.20±0.40** | **0.10±0.44** | **0.52±0.27** | **0.52±0.27** | **0.68±0.39** | **0.68±0.43** | **0.68±0.43** | **0.74±0.40** |

underlying rank test requires at least four observed variables; it also struggled to resolve the causal structure in the remaining scenarios. In contrast, our proposed algorithm demonstrated optimal performance across all cases. It consistently identified and characterized the impure connections among the observed variables, showcasing its advantages in handling impure structures.

**Robustness Verification.** Besides, we quantify the gap between the sparsest and the alternative CICA solution with a simulation study. We analyze all possible causal graphs with 1 latent variable and $n$ observed variables $(X_1, ..., X_n)$, for $n \in \{3, 4, 5, 6\}$. To make the enumeration feasible, we assume a causal order $X_1 \rightarrow ... \rightarrow X_n$ and then enumerate all $2^{n(n-1)/2}$ possible graphs by randomizing all possible causal edges between the observed variables. For each of these possible graphs, we compute the number of non-zero entries for all $n + 1$ possible valid CICA solutions.

| n | #graphs | $\text{Sum}_0$ (True) | $\text{Sum}_1$ | $\text{Sum}_2$ | $\text{Sum}_3$ | $\text{Sum}_4$ | $\text{Sum}_5$ | $\text{Sum}_6$ |
|---|---|---|---|---|---|---|---|---|
| 3 | 8 | 36 | 43 | 50 | 59 | - | - | - |
| 4 | 64 | 448 | 531 | 602 | 682 | 809 | - | - |
| 5 | 1024 | 10240 | 12015 | 13368 | 14857 | 16802 | 19758 | - |
| 6 | 32768 | 442368 | 513675 | 563799 | 619008 | 685666 | 769179 | 895521 |

Table 2: Simulation results of the sparsity gap.

Table 2 collects the total sum of non-zero entries for each of the $n + 1$ solutions (sorted from sparsest to densest) aggregated across all enumerated graphs, which provides clear evidence for the gap between the sparsest solution and the alternative solution. For example, when $n = 6$, the average number of non-zero entries in $\mathbf{I} - \mathbf{B}_{\mathbf{X},\mathbf{X}}$ is 13.5 ($\text{Sum}_0$/#graphs), and it increases 2.176 (($\text{Sum}_1$-$\text{Sum}_0$)/#graphs, 16.12% relative increase rate) per graph compared with the second sparsest CICA solution. This confirms that the sparsity gap is a generic property, and provides practical stability and robustness guarantees for CICA in finding the true causal graph.

## 5 CONCLUSION

In this paper, we introduce a new principle, Conditional Independent Component Analysis (CICA), which aims to extract components that are mutually independent given a certain number of latent variables. CICA naturally induces two-sided projections $\omega_1^\top \mathbf{Y} \perp\!\!\!\perp \omega_2^\top \mathbf{Z}$, which carry a richer identification signal than one-sided projections $\omega^\top \mathbf{Y} \perp\!\!\!\perp \mathbf{Z}$ used in GIN/TIN conditions, thus improving the identifiability in latent causal structure learning. Although CICA involves additional indeterminacy on the latent conditional set, we show that sparsity resolves this ambiguity and yields full identification of the latent variables and causal relationships. Building on our theoretical results, we derive an estimation algorithm for latent causal structure recovery. Synthetic and real-world experiments show the superiority of our methods in dealing with impure structures. One of the future research directions is extending the linear mixing assumption to the non-linear setting, existing techniques, e.g., Chen et al. (2024b; 2025b), may help to mitigate this issue.

## ACKNOWLEDGMENTS

The authors thank anonymous reviewers for their helpful comments. We also acknowledge the support from National Natural Science Foundation of China (NSFC) under grant No. 62372116, 62476163, U24A20233, National Key R&D Program of China under grant No. 2025YFC3410003, NSF Award No. 2229881, Guangdong Basic and Applied Basic Research Foundation under grant number 23201910250000514, 2023B1515120020, AI Institute for Societal Decision Making (AI-SDM), the National Institutes of Health (NIH) under Contract R01HL159805, and grants from Quris AI, Florin Court Capital, MBZUAI-WIS Joint Program, and the Al Deira Causal Education project. ZM would like to acknowledge the support by STU Scientific Research Initiation Grant under grant No. NTF25024T. The computations in this research were mainly performed using the CFFF platform of Fudan University.

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

ORGANIZATION OF APPENDICES

## A  DEFINITIONS, EXAMPLES, AND PROOFS

### A.1  DEFINITIONS

**Definition 9** (Treks ([Sullivant et al., 2010](#))). *In $\mathcal{G}$, a trek from X to Y is an ordered pair of directed paths $(P_1, P_2)$ where $P_1$ has a sink X, $P_2$ has a sink Y, and both $P_1$ and $P_2$ have the same source Z.*

**Definition 10** (T-separation ([Sullivant et al., 2010](#))). *Let $\mathbf{A}, \mathbf{B}, \mathbf{C_A}$, and $\mathbf{C_B}$ be four subsets of $\mathbf{V}_{\mathcal{G}}$ in graph $\mathcal{G}$ (not necessarilly disjoint). $(\mathbf{C_A}, \mathbf{C_B})$ t-separates $\mathbf{A}$ from $\mathbf{B}$ if for every trek $(P_1, P_2)$ from a vertex in $\mathbf{A}$ to a vertex in $\mathbf{B}$, either $P_1$ contains a vertex in $\mathbf{C_A}$ or $P_2$ contains a vertex in $\mathbf{C_B}$.*

**Lemma 12** (Rank and T-separation ([Sullivant et al., 2010](#))). *Given two sets of variables $\mathbf{A}$ and $\mathbf{B}$ from a linear model with graph $\mathcal{G}$, we have $\mathrm{rank}\,(\Sigma_{\mathbf{A},\mathbf{B}}) = \min\{|\mathbf{C_A}| + |\mathbf{C_B}| \,:\, (\mathbf{C_A}, \mathbf{C_B})\, t\text{-}separates\ \mathbf{A}\ from\ \mathbf{B}\ in\ \mathcal{G}\}$, where $\Sigma_{\mathbf{A},\mathbf{B}}$ is the cross-covariance over $\mathbf{A}$ and $\mathbf{B}$.*

### A.2  EXAMPLES

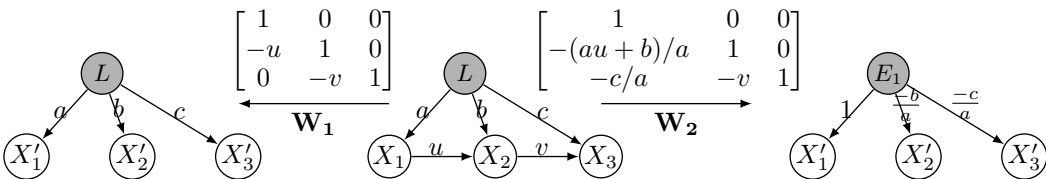

Figure 6: An example of different CICA solutions for $\mathbf{X}$. $\mathbf{W_1}$ is a CICA solution that renders $\mathbf{X}'$ conditionally independent given $L$, while $\mathbf{W_2}$ renders $\mathbf{X}'$ conditionally independent given $E_1$, the exogenous noise of $X_1$. The gray/white rectangle denotes non-zero/zero entries.

### A.3  PROOF

#### A.3.1  PRELIMINARIES

**Lemma 13** (Darmois-Skitovich Theorem ([Darmois, 1953](#); [Skitovich, 1954](#))). *Given $n$ independent scalar random variables $X_1, \ldots, X_n$ that are not necessarily identically distributed. Consider two linear statistics $L_1 = \sum \alpha_i X_i$, $L_2 = \sum \beta_i X_i$, where $\alpha_i, \beta_i$ are constant coefficients. $L_1$ and $L_2$ are independent if and only if the random variables $X_j$ for which $\alpha_j \beta_j \neq 0$ follow a normal distribution.*

**Lemma 14** (Graphical implication of TIN ([Dai et al., 2022](#))). *Let $\mathbf{Z}$, $\mathbf{Y}$ be two subsets of variables, we have:*

$$\mathrm{TIN}(\mathbf{Z}, \mathbf{Y}) = \min\{|\mathbf{S}| \mid \mathbf{S} \text{ is a vertex cut from } \mathrm{Anc}(\mathbf{Z}) \text{ to } \mathbf{Y}\}. \tag{3}$$

In a linear non-Gaussian system, the Darmois–Skitovich theorem ([Darmois, 1953](#)) plays a key role in determining the independence of two linear statistics. It tells us that two linear combinations of independent non-Gaussian variables are independent if they do not share any non-Gaussian component. As $\omega^\top \mathbf{X}$ is a linear combination of independent noises of $\mathbf{V}$, characterizing all possible independence that can be constructed from observational data requires understanding which noise combinations can be represented by $\omega^\top \mathbf{X}$. To this end, we introduce a new definition that describes the noise combinations attainable through linear combinations of observed variables.

**Definition 11** (Constructible Noise Combination). *A noise combination $\mathbf{Z} \subseteq \mathbf{E}$, which consists of some independent noises of variables in $\mathbf{V}$. The noise combination $\mathbf{Z}$ is constructible by some observed variables $\mathbf{X}$ if there exists a coefficient vector $\omega$ such that $\omega^\top \mathbf{X}$ is a linear combination of the noise variables in $\mathbf{Z}$ with non-zero coefficients, i.e., $\omega^\top \mathbf{X} = \sum_{E_i \in \mathbf{Z}} \nu_i E_i (\nu_i \neq 0)$. In other words, $\omega^\top \mathbf{X}$ contains and only contains noise variables in $\mathbf{Z}$.*

**Example 5.** *In the figure below, $L$ is the latent confounder of two observed variables $X$ and $Y$. We have $\emptyset, \{E_L, E_X\}, \{E_L, E_Y\}, \{E_X, E_Y\}$ and $\{E_L, E_X, E_Y\}$ are constructible while the other noise combinations are not.*

**Definition 12** (Bottleneck). *Let $\mathbf{J}, \mathbf{K}$ and $\mathbf{B}$ be three subsets of $\mathbf{V}$ that are not necessarily disjoint. We say that $\mathbf{B}$ is a bottleneck from $\mathbf{J}$ to $\mathbf{K}$ if, for every $j \in \mathbf{J}$ and every $k \in \mathbf{K}$, each directed path from $j$ to $k$ includes some $b \in \mathbf{B}$.*

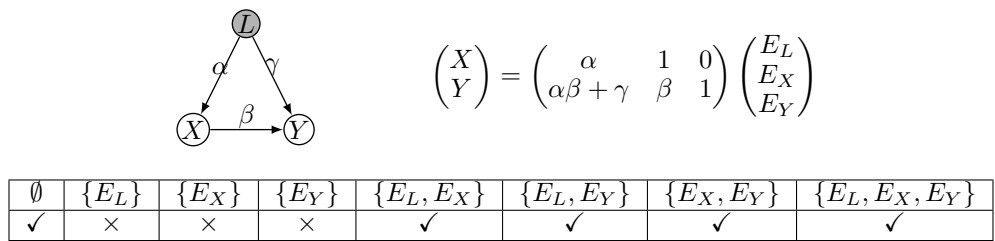

Table 3: All constructive noise combinations of the graph above.

| $\emptyset$ | $\{E_L\}$ | $\{E_X\}$ | $\{E_Y\}$ | $\{E_L, E_X\}$ | $\{E_L, E_Y\}$ | $\{E_X, E_Y\}$ | $\{E_L, E_X, E_Y\}$ |
|---|---|---|---|---|---|---|---|
| ✓ | × | × | × | ✓ | ✓ | ✓ | ✓ |

**Definition 13** (Latest Minimal bottleneck (LM bottleneck)). *Let* $\mathbf{J}, \mathbf{K}$ *and* $\mathbf{B}$ *be three subsets of* $\mathbf{V}$ *that are not necessarily disjoint. We say that a bottleneck* $\mathbf{B}$ *from* $\mathbf{J}$ *to* $\mathbf{K}$ *called minimal if every bottleneck* $\mathbf{B}'$ *from* $J$ *to* $K$ *has* $|\mathbf{B}'| \geq |\mathbf{B}|$. *Furthermore,* $\mathbf{B}$ *is the (topologically) latest minimal bottleneck (LM bottleneck) from* $\mathbf{J}$ *to* $\mathbf{K}$ *if for every minimal bottleneck* $\mathbf{B}'$ *from* $\mathbf{J}$ *to* $\mathbf{K}$, $\mathbf{B}$ *is the bottleneck from* $\mathbf{B}'$ *to* $\mathbf{K}$.

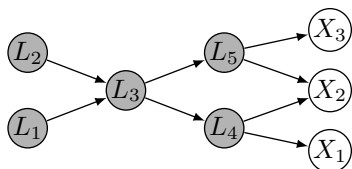

**Example 6.** *In the figure above,* $\{L_3\}$ *is a minimal bottleneck from* $\{L_1, L_2\}$ *to* $\mathbf{X}$. *More precisely, it is also the corresponding LM bottleneck.* $\{L_3, L_5\}$ *is a minimal bottleneck from* $\{L_1, L_5\}$ *to* $\mathbf{X}$ *but it is not the corresponding LM bottleneck. Instead, it should be* $\{L_4, L_5\}$.

**Definition 14.** *We define the LM bottleneck-dominated set of* $\mathbf{B}$ *with respect to* $\mathbf{K}$ *as the set of all nodes in* $\mathbf{V}$ *such that* $\mathbf{B}$ *is the LM bottleneck from the node to* $\mathbf{K}$. *Formally,*

$$\mathcal{D}_{\mathbf{B}, \to \mathbf{K}} := \{v \in \mathbf{V} | \mathbf{B} \text{ is the LM bottleneck from } v \text{ to } \mathbf{K}\} \tag{4}$$

This is the maximal set of nodes for which $\mathbf{B}$ serves as a bottleneck toward $\mathbf{K}$.

**Lemma 15.** *Let* $\mathbf{J}, \mathbf{K} \subseteq \mathbf{V}$ *that are not necessarily disjoint. The LM bottleneck from* $\mathbf{J}$ *to* $\mathbf{K}$ *always exists and is unique.*

*Proof.* Build the standard vertex–splitting network $G' = (\mathbf{V}', \mathbf{E}')$ with capacities as follows. For each $v \in \mathbf{V}$, create two nodes $v^-, v^+$ and add a unit–capacity edge $v^- \to v^+$. For each $u \to v \in \mathbf{E}$, add an infinite–capacity edge $u^+ \to v^-$. Add a source $s$ and a sink $t$; for each $j \in \mathbf{J}$ add an infinite–capacity edge $s \to j^-$, and for each $k \in \mathbf{K}$ add an infinite–capacity edge $k^+ \to t$.

Then for any $\mathbf{B} \subseteq \mathbf{V}$, $\mathbf{B}$ is a bottleneck from $\mathbf{J}$ to $\mathbf{K}$ $\iff$ $C(\mathbf{B}) := \{v^- \to v^+ : v \in \mathbf{B}\}$ is an $s$–$t$ cut in $G'$. Moreover, the capacity of $C(\mathbf{B})$ equals $|\mathbf{B}|$. Indeed, every path $j \rightsquigarrow k$ in $G$ lifts to a path $s \rightsquigarrow j^- \rightsquigarrow \cdots \rightsquigarrow k^+ \rightsquigarrow t$ in $G'$ that necessarily traverses the split edge $x^- \to x^+$ for each visited $x$; cutting precisely the split edges in $C(\mathbf{B})$ blocks all lifted $s$–$t$ paths iff every $j \rightsquigarrow k$ path in $G$ meets $\mathbf{B}$. Since only split edges have finite capacity, the cut capacity is $|\mathbf{B}|$.

Therefore, a minimal bottleneck (of smallest cardinality) exists because it corresponds to a minimum $s$–$t$ cut in the finite network $G'$.

Let $f$ be any maximum flow on $G'$ and let $R_f$ be the residual network. Define

$$T_f := \{x \in \mathcal{V}' : t \text{ can reach } x \text{ in } R_f\}, \qquad S_f := \mathcal{V}' \setminus T_f.$$

Standard max–flow theory implies that $(S_f, T_f)$ is a minimum $s$–$t$ cut, and that $T_f$ is inclusion–wise maximal among the sink sides of all minimum cuts (the "closest-to-$t$" minimum cut); in particular, $T_f$ is unique. For completeness: if $(S', T')$ is any minimum cut, then edges from $T'$ to $S'$ carry zero residual capacity and edges from $S'$ to $T'$ are saturated; hence every node reachable from $t$ in $R_f$ must lie in $T'$, so $T' \subseteq T_f$.

Map the $t$-closest minimum cut back to a vertex set:

$$\mathbf{B}^\star := \{\, v \in \mathbf{V} \,:\, v^- \in S_f \text{ and } v^+ \in T_f \,\}.$$

By construction, $C(\mathbf{B}^\star)$ is the cut $(S_f, T_f)$, hence $\mathbf{B}^\star$ is a minimal bottleneck.

Let $\mathbf{B}'$ be any other minimal bottleneck, and let $(S', T')$ be its corresponding minimum cut in $G'$. From the previous paragraph $T' \subseteq T_f$ (equivalently $S_f \subseteq S'$). Take any path $b' \rightsquigarrow k$ in $G$ with $b' \in \mathbf{B}'$ and $k \in \mathbf{K}$; its lift in $G'$ goes from $b'^- \in S' \supseteq S_f$ to $k^+ \in T_f$, hence must cross the cut $(S_f, T_f)$ through some split edge $v^- \to v^+$ with $v \in \mathbf{B}^\star$. Therefore every $b' \rightsquigarrow k$ path passes through $\mathbf{B}^\star$, i.e., $\mathbf{B}^\star$ is a bottleneck from $\mathbf{B}'$ to $\mathbf{K}$. Since $\mathbf{B}'$ was an arbitrary minimal bottleneck, $\mathbf{B}^\star$ is the latest minimal (LM) bottleneck.

If $\widetilde{\mathbf{B}}$ is another LM bottleneck with minimum cut $(\widetilde{S}, \widetilde{T})$, then by the same argument its sink side $\widetilde{T}$ must contain the sink side of every minimum cut, hence $\widetilde{T} = T_f$ by the maximality/uniqueness of $T_f$. Thus $\widetilde{\mathbf{B}} = \mathbf{B}^\star$. In summary, the LM bottleneck from $\mathbf{J}$ to $\mathbf{K}$ exists and is unique. $\square$

**Lemma 16.** *A variable set $\mathbf{V_b} \subseteq \mathbf{V}$ is an LM bottleneck from some variable set $\mathbf{V_s}$ to $\mathbf{X}$ if and only if $\mathbf{V_b}$ itself is the LM bottleneck from $\mathbf{V_b}$ to $\mathbf{X}$.*

*Proof.* If $\mathbf{V}_b$ is the LM bottleneck from some $\mathbf{V}_s$ to $\mathbf{X}$, then $\mathbf{V}_b$ is the LM bottleneck from $\mathbf{V}_b$ to $\mathbf{X}$. Since $\mathbf{V}_b$ is a bottleneck from $\mathbf{V}_s$ to $\mathbf{X}$, every $\mathbf{V}_s \rightsquigarrow \mathbf{X}$ path meets $\mathbf{V}_b$. Consequently $\mathbf{V}_b$ is trivially a bottleneck from $\mathbf{V}_b$ to $\mathbf{X}$ (every $v \in \mathbf{V}_b$–$\mathbf{X}$ path contains $v \in \mathbf{V}_b$ at its first node).

We show that $\mathbf{V}_b$ is minimal for the pair $(\mathbf{V}_b, \mathbf{X})$. Assume, for contradiction, that there exists a bottleneck $\mathbf{C}$ from $\mathbf{V}_b$ to $\mathbf{X}$ with $|\mathbf{C}| < |\mathbf{V}_b|$. Then for any $s \in \mathbf{V}_s$ and $x \in \mathbf{X}$, each $s \rightsquigarrow x$ path first hits $\mathbf{V}_b$ and, from that hit, must pass $\mathbf{C}$ (because $\mathbf{C}$ meets every $\mathbf{V}_b \rightsquigarrow \mathbf{X}$ path). Hence $\mathbf{C}$ is also a bottleneck from $\mathbf{V}_s$ to $\mathbf{X}$, contradicting the minimality of $\mathbf{V}_b$ for $(\mathbf{V}_s, \mathbf{X})$.

It remains to verify the latest property for $(\mathbf{V}_b, \mathbf{X})$. Let $\mathbf{C}$ be any minimal bottleneck from $\mathbf{V}_b$ to $\mathbf{X}$. We claim that every $\mathbf{C} \rightsquigarrow \mathbf{X}$ path meets $\mathbf{V}_b$. Indeed, otherwise there would exist $c \in \mathbf{C}$ and $x \in \mathbf{X}$ with a path $c \rightsquigarrow x$ avoiding $\mathbf{V}_b$. Concatenate a path $s \rightsquigarrow c$ with $s \in \mathbf{V}_s$ whose internal nodes avoid $\mathbf{V}_b$ (which exists because $\mathbf{V}_b$ is minimal for $(\mathbf{V}_s, \mathbf{X})$; otherwise $c$ would be redundant in $\mathbf{C}$), and then follow the $c \rightsquigarrow x$ path; this would give an $\mathbf{V}_s \rightsquigarrow \mathbf{X}$ path avoiding $\mathbf{V}_b$, contradicting that $\mathbf{V}_b$ is a bottleneck from $\mathbf{V}_s$ to $\mathbf{X}$. Thus $\mathbf{V}_b$ is a bottleneck from $\mathbf{C}$ to $\mathbf{X}$; since $\mathbf{C}$ was arbitrary minimal for $(\mathbf{V}_b, \mathbf{X})$, $\mathbf{V}_b$ is the LM bottleneck from $\mathbf{V}_b$ to $\mathbf{X}$.

If $\mathbf{V}_b$ is the LM bottleneck from $\mathbf{V}_b$ to $\mathbf{X}$, then $\mathbf{V}_b$ is the LM bottleneck from some $\mathbf{V}_s$ to $\mathbf{X}$. Take $\mathbf{V}_s := \mathbf{V}_b$. By assumption, $\mathbf{V}_b$ is a (latest) minimal bottleneck for $(\mathbf{V}_b, \mathbf{X})$; in particular it is a bottleneck from $\mathbf{V}_s$ to $\mathbf{X}$ and, for every minimal bottleneck $\mathbf{C}$ from $\mathbf{V}_s$ to $\mathbf{X}$, every $\mathbf{C} \rightsquigarrow \mathbf{X}$ path meets $\mathbf{V}_b$. Hence $\mathbf{V}_b$ is the LM bottleneck from $\mathbf{V}_s$ to $\mathbf{X}$. $\square$

**Theorem 4** (Graphical criteria of the constructible noise combination)**.** *Any noise combination $\alpha$ is constructible by $\mathbf{X}$ if and only if (i) $\exists \mathbf{T} \subseteq \mathbf{V}$ s.t. $\mathbf{T}$ is the LM bottleneck of $\mathbf{T}$ to $\mathbf{X}$ in $\mathcal{G}$. (ii) $\forall V_i \in \mathbf{V}$, $\alpha_i = 0 \iff \mathbf{T}$ is a bottleneck from $V_i$ to $\mathbf{X}$ in $\mathcal{G}$.*

*Proof.* Constructibility $\implies$ (i)–(ii). Assume $\alpha$ is constructible, let $\mathbf{S} := \{\, i \in \mathbf{V} \,:\, \alpha_i \neq 0 \,\}$ be the support of $\alpha$. By Lemma 15, the LM bottleneck $\mathbf{T}^\star$ from $\mathbf{S}$ to $\mathbf{X}$ exists and is unique; by Lemma 16, $\mathbf{T}^\star$ is also the LM bottleneck from $\mathbf{T}^\star$ to $\mathbf{X}$. This gives (i).

It remains to show (ii). Fix $i \in \mathbf{V}$. If $\alpha_i = 0$, then $\mathbf{T}^\star$ is a bottleneck from $V_i$ to $\mathbf{X}$. Suppose to the contrary that there exists a directed path $P : i \rightsquigarrow x$ avoiding $\mathbf{T}^\star$ (with $x \in \mathbf{X}$). Since $\mathbf{T}^\star$ is the LM bottleneck from $\mathbf{S}$ to $\mathbf{X}$, it is, by definition, the bottleneck from every minimal bottleneck for $(\mathbf{S}, \mathbf{X})$ to $\mathbf{X}$; in particular, $P$ can be concatenated with an $\mathbf{S} \rightsquigarrow i$ path that avoids $\mathbf{T}^\star$ up to $i$ (otherwise $i$ would be separated from $\mathbf{S}$ by $\mathbf{T}^\star$ and $\alpha_i$ would inherit a nonzero contribution through $i$'s first hit in $\mathbf{T}^\star$). Consequently there exists at least one directed path from $\mathbf{S}$ to $x$ that avoids $\mathbf{T}^\star$ and can be continued by $P$, contradicting that $\mathbf{T}^\star$ intercepts all $\mathbf{S} \rightsquigarrow \mathbf{X}$ paths. Hence every $i \rightsquigarrow \mathbf{X}$ path hits $\mathbf{T}^\star$, i.e., $\mathbf{T}^\star$ is a bottleneck from $V_i$ to $\mathbf{X}$.

If $\alpha_i \neq 0$, then $\mathbf{T}^\star$ is not a bottleneck from $V_i$ to $\mathbf{X}$. If every $i \rightsquigarrow x$ path met $\mathbf{T}^\star$, then any $\omega$ whose latent terms have been canceled via constraints indexed by $\mathbf{T}^\star$ would give $\nu_i = 0$ (all contributions must pass through $\mathbf{T}^\star$ and are nullified), contradicting $\alpha_i \neq 0$. Thus $i$ has a path to some $x \in \mathbf{X}$ that avoids $\mathbf{T}^\star$.

Combining the two implications yields (ii) with $\mathbf{T} = \mathbf{T}^\star$.

($\Leftarrow$) (i)–(ii) $\Longrightarrow$ constructibility. Assume (i)–(ii) hold for some $\mathbf{T} \subseteq \mathbf{V}$. Let

$$\mathbf{S} := \{ i \in \mathbf{V} : \alpha_i \neq 0 \} = \{ i \in \mathbf{V} : \mathbf{T} \text{ is not a bottleneck from } V_i \text{ to } \mathbf{X} \}.$$

By (i) and Lemma 16, $\mathbf{T}$ is the LM bottleneck from $\mathbf{T}$ to $\mathbf{X}$ and, therefore, from $\mathbf{S}$ to $\mathbf{X}$ as well (latest with respect to any minimal bottleneck for $(\mathbf{S}, \mathbf{X})$).

Consider the vertex–splitting network $G'$ used in Lemma 15. Let $(S_f, T_f)$ be the unique $t$-closest minimum cut in $G'$ (induced by any maximum flow); it induces $\mathbf{T}$ by $\mathbf{T} = \{v \in \mathbf{V} : v^- \in S_f, v^+ \in T_f\}$. Choose $|\mathbf{T}|$ distinct nodes $\{x_1, \ldots, x_{|\mathbf{T}|}\} \subseteq \mathbf{X}$ reached by the $|\mathbf{T}|$ vertex–disjoint paths guaranteed by Menger's theorem from $\mathbf{T}$ to $\mathbf{X}$ (tightness of the cut). Define $\omega$ supported on $\{x_1, \ldots, x_{|\mathbf{T}|}\}$ as the unique solution to the linear system that zeroes the contributions flowing through $\mathbf{T}$ (the $|\mathbf{T}| \times |\mathbf{T}|$ system is non-singular because the $\mathbf{T} \rightsquigarrow \{x_\ell\}$ paths are vertex-disjoint). Then 1) for any $i$ such that $\mathbf{T}$ is a bottleneck from $V_i$ to $\mathbf{X}$, every $i \rightsquigarrow \mathbf{X}$ path must traverse some $t \in \mathbf{T}$, hence its contribution to $\nu_i$ is canceled by construction; thus $\nu_i = 0$. 2) for any $i$ such that $\mathbf{T}$ is not a bottleneck from $V_i$ to $\mathbf{X}$, there exists a path $P : i \rightsquigarrow x$ that avoids $\mathbf{T}$. Since our constraints only cancel flows that pass through $\mathbf{T}$, the term corresponding to $P$ survives so $\nu_i \neq 0$.

Finally, impose additional linear constraints (orthogonality) on $\omega$ to remove latent terms (these constraints are independent of the $\mathbf{T}$-cancellation because the latter acts only on flows that cross $\mathbf{T}$), which is always possible as we only eliminate $|\mathbf{T}|$ directions associated with the cut while retaining degrees of freedom on $\mathbf{X}$. Thus $\alpha$ is constructible by $\mathbf{X}$. $\qquad\square$

**Corollary 1.** *Any noise combination $\alpha$ is constructible by $\tilde{\mathbf{X}} \subseteq \mathbf{X}$ if and only if (i) $\exists \mathbf{T} \subseteq \mathbf{V}$ s.t. $\mathbf{T}$ is the LM bottleneck of $\mathbf{T}$ to $\mathbf{X}$ in $\mathcal{G}$. (ii) $\forall V_i \in \mathbf{V}$, $\alpha_i = 0 \Longleftrightarrow \mathbf{T}$ is a bottleneck from $V_i$ to $\mathbf{X}$ in $\mathcal{G}$.*

### A.3.2  ILLUSTRATION OF NON-IDENTIFIABILITY ISSUE ON FIG. 1A AND 1B

| Z \ Y | $\{X_1\}$ | $\{X_2\}$ | $\{X_3\}$ | $\{X_1, X_2\}$ | $\{X_1, X_3\}$ | $\{X_2, X_3\}$ | $\{X_1, X_2, X_3\}$ |
|---|---|---|---|---|---|---|---|
| $\{X_1\}$ | 1 | 1 | 1 | 2 | 2 | 2 | 2 |
| $\{X_2\}$ | 1 | 1 | 1 | 2 | 2 | 2 | 3 |
| $\{X_3\}$ | 1 | 1 | 1 | 2 | 2 | 2 | 3 |

Table 4: TIN value of different $\mathbf{Y}$ and $\mathbf{Z}$ of Fig. 1a

| Z \ Y | $\{X_1\}$ | $\{X_2\}$ | $\{X_3\}$ | $\{X_1, X_2\}$ | $\{X_1, X_3\}$ | $\{X_2, X_3\}$ | $\{X_1, X_2, X_3\}$ |
|---|---|---|---|---|---|---|---|
| $\{X_1\}$ | 1 | 1 | 1 | 2 | 2 | 2 | 2 |
| $\{X_2\}$ | 1 | 1 | 1 | 2 | 2 | 2 | 3 |
| $\{X_3\}$ | 1 | 1 | 1 | 2 | 2 | 2 | 3 |

Table 5: TIN value of different $\mathbf{Y}$ and $\mathbf{Z}$ of Fig. 1b

*Proof.* We use $\mathcal{G}_1$ and $\mathcal{G}_2$ to represent the causal graph in Fig. 1a, and Fig. 1b, respectively. By some simple calculations, we can find that both $\mathcal{G}_1$ and $\mathcal{G}_2$ have no rank-deficiency constraints. Thus, for each pair of $(\mathbf{Z}, \mathbf{Y})$, $\text{rank}_{\mathcal{G}_1}(\Sigma_{\mathbf{Z},\mathbf{Y}}) = \min(\mathbf{Z}, \mathbf{Y}) = \text{rank}_{\mathcal{G}_2}(\Sigma_{\mathbf{Z},\mathbf{Y}})$. In addition, as we can see in Table 4 and 5, $\mathcal{G}_1$ and $\mathcal{G}_2$ have the same TIN value for each $(\mathbf{Z}, \mathbf{Y})$. As $\text{GIN}(\mathbf{Z}, \mathbf{Y})$ is satisfied if and only if $\text{TIN}(\mathbf{Z}, \mathbf{Y}) = \text{rank}(\Sigma_{\mathbf{Z},\mathbf{Y}}) < |\mathbf{Y}|$ (Dai et al., 2022), whether the GIN condition is satisfied for a certain pair $(\mathbf{Z}, \mathbf{Y})$ keeps the same in $\mathcal{G}_1$ and $\mathcal{G}_2$. $\qquad\square$

### A.3.3  PROOF OF LEMMA 1

*Proof.* $\text{NS}(\mathbf{Z}) = \text{Anc}(\mathbf{Z})$. By Theorem 4, $\text{Anc}(\mathbf{Z})$ is constructible. Therefore, according to the definition of constructible noise combination, we can always find a non-zero coefficient $\omega_2$ such that $\text{NS}(\omega_2^\top \mathbf{Z}) = \text{Anc}(\mathbf{Z})$. Since $\omega_1^\top \mathbf{Y} \perp\!\!\!\perp \mathbf{Z}$, we naturally obtain $\omega_1^\top \mathbf{Y} \perp\!\!\!\perp \omega_2^\top \mathbf{Z}$. $\qquad\square$

### A.3.4 PROOF OF REMARK 1

*Proof.* By Theorem 4, we can enumerate all constructive noise combinations by finding all LM bottlenecks. All LM bottlenecks can be identified by testing for Lemma 16. All constructive noise combinations by $\{X_1, X_2\}$, $\{X_1, X_3\}$ and $\{X_2, X_3\}$ in Fig. 1a are shown in Tab. 6. All constructive noise combinations by $\{X_1, X_2\}$, $\{X_1, X_3\}$ and $\{X_2, X_3\}$ in Fig. 1b are shown in Tab. 7.

| $\mathbf{Z} = \{X_1, X_2\}$ | $\emptyset$ | $\{L\}$ | $\{X_1\}$ | $\{X_2\}$ |
|---|---|---|---|---|
| | $\{E_L, E_1, E_2\}$ | $\{E_1, E_2\}$ | $\{E_L, E_2\}$ | $\{E_L, E_1\}$ |
| $\mathbf{Z} = \{X_1, X_3\}$ | $\emptyset$ | $\{L\}$ | $\{X_1\}$ | $\{X_3\}$ |
| | $\{E_L, E_1, E_2, E_3\}$ | $\{E_1, E_2, E_3\}$ | $\{E_L, E_2, E_3\}$ | $\{E_L, E_1, E_2\}$ |
| $\mathbf{Z} = \{X_2, X_3\}$ | $\emptyset$ | $\{L\}$ | $\{X_2\}$ | $\{X_3\}$ |
| | $\{E_L, E_1, E_2, E_3\}$ | $\{E_1, E_2, E_3\}$ | $\{E_L, E_3\}$ | $\{E_L, E_1, E_2\}$ |

Table 6: All constructive noise combinations by $\{X_1, X_2\}$, $\{X_1, X_3\}$ and $\{X_2, X_3\}$ in Fig. 1a. Each constructive noise combination is shown together with its corresponding LM bottleneck in a pairwise manner.

| $\mathbf{Z} = \{X_1, X_2\}$ | $\emptyset$ | $\{L\}$ | $\{X_1\}$ | $\{X_2\}$ |
|---|---|---|---|---|
| | $\{E_L, E_1, E_2, E_3\}$ | $\{E_1, E_2, E_3\}$ | $\{E_L, E_2, E_3\}$ | $\{E_L, E_1, E_3\}$ |
| $\mathbf{Z} = \{X_1, X_3\}$ | $\emptyset$ | $\{L\}$ | $\{X_1\}$ | $\{X_3\}$ |
| | $\{E_L, E_1, E_3\}$ | $\{E_1, E_3\}$ | $\{E_L, E_3\}$ | $\{E_L, E_1\}$ |
| $\mathbf{Z} = \{X_2, X_3\}$ | $\emptyset$ | $\{L\}$ | $\{X_2\}$ | $\{X_3\}$ |
| | $\{E_L, E_1, E_2, E_3\}$ | $\{E_1, E_2, E_3\}$ | $\{E_L, E_1, E_3\}$ | $\{E_L, E_2\}$ |

Table 7: All constructive noise combinations by $\{X_1, X_2\}$, $\{X_1, X_3\}$ and $\{X_2, X_3\}$ in Fig. 1b. Each constructive noise combination is shown together with its corresponding LM bottleneck in a pairwise manner.

From Tab. 6, when $\mathbf{Z} = \{X_1, X_2\}$, $\mathbf{Y} = \{X_2, X_3\}$, we can construct $\mathrm{NS}(\omega_1^\top \mathbf{Z}) = \{E_1, E_2\}$ and $\mathrm{NS}(\omega_2^\top \mathbf{Y}) = \{E_L, E_3\}$ with non-zero $\omega_1, \omega_2 \in \mathbb{R}^2$. In contrast, in Tab. 7, each pair of constructive noise combinations by $\mathbf{Z}$ and $\mathbf{Y}$ has shared noise components, thus cannot be independent. The conclusion for $\mathbf{Z} = \{X_1, X_3\}$ and $\mathbf{Y} = \{X_2, X_3\}$ can be analyzed similarly.

$\square$

### A.3.5 PROOF OF LEMMA 2

**Lemma 2** (Identifiability and indeterminacy of CICA). *Suppose Assumption 1 holds, let $\mathbf{X}$ be $m$ observed variables. If there exist $p_{min}(\mathbf{X})$ latent variables $\mathbf{L}$ such that for two valid CICA solutions $\mathbf{W}_1$ and $\mathbf{W}_2$, the components of $\mathbf{Z}^{(k)} := \mathbf{W}_k \mathbf{X}$ are mutually conditionally independent given $\mathbf{L}$ for $k \in \{1, 2\}$, then there uniquely exist a permutation matrix $\mathbf{P}_\pi$ (for some permutation $\pi$ of $[m]$) and a non-singular diagonal matrix $\mathbf{D}$ such that $\mathbf{W}_2 = \mathbf{P}_\pi \mathbf{D} \mathbf{W}_1$.*

*Proof.* Under Assumption 1 there exist an invertible $\mathbf{A} \in \mathbb{R}^{m \times m}$, a latent vector $\mathbf{L} \in \mathbb{R}^p$, a matrix $\mathbf{M} \in \mathbb{R}^{m \times p}$, and a noise $\mathbf{E} = (E_1, \ldots, E_m)^\top$ with mutually independent coordinates, $\mathbf{E} \perp\!\!\!\perp \mathbf{L}$, finite non-zero variances, and with at most one Gaussian coordinate, such that $\mathbf{X} = \mathbf{AS}$ and $\mathbf{S} = \mathbf{ML} + \mathbf{E}$. For $k \in \{1, 2\}$ write $\mathbf{Z}^{(k)} := \mathbf{W}_k \mathbf{X}$ and set $\mathbf{B}_k := \mathbf{W}_k \mathbf{A}$ (hence $\mathbf{Z}^{(k)} = \mathbf{B}_k \mathbf{S}$).

For every $\ell$,

$$\mathbf{Z}^{(k)} \mid \{\mathbf{L} = \ell\} = \mathbf{B}_k(\mathbf{ML} + \mathbf{E}) \mid \{\mathbf{L} = \ell\} = (\mathbf{B}_k \mathbf{M})\, \ell \;+\; \mathbf{B}_k \mathbf{E}.$$

Thus, for each $k$, the coordinates of $\mathbf{Z}^{(k)}$ are mutually independent given $\mathbf{L}$ if and only if the coordinates of $\mathbf{B}_k \mathbf{E}$ are mutually independent (a deterministic shift $\mathbf{B}_k \mathbf{M}\ell$ does not affect independence). In particular, we have an ICA model with independent sources $\mathbf{E}$ and mixing matrices $\mathbf{B}_k$.

($\Rightarrow$) Necessity. Assume (i) holds: there exists a specific latent vector $\mathbf{L}$ such that $\mathbf{Z}^{(k)}$ has mutually independent coordinates conditional on $\mathbf{L}$ for $k = 1, 2$. By the reduction above, both $\mathbf{B}_1 \mathbf{E}$ and

$\mathbf{B}_2\mathbf{E}$ have mutually independent coordinates. Since $\mathbf{E}$ has mutually independent coordinates with at most one Gaussian, the standard ICA identifiability theory (Comon, 1994) tells that the only invertible linear maps sending $\mathbf{E}$ to a vector with independent coordinates are generalized permutation matrices. That is, the invertible matrices $\mathbf{B}_1$ and $\mathbf{B}_2$ must both be of the form $\mathbf{B}_1 = \mathbf{P}_1\mathbf{D}_1$ and $\mathbf{B}_2 = \mathbf{P}_2\mathbf{D}_2$, where $\mathbf{P}_k$ are permutation matrices and $\mathbf{D}_k$ are non-singular diagonal matrices. Consequently, we can express $\mathbf{B}_2$ in terms of $\mathbf{B}_1$ as $\mathbf{B}_2 = \mathbf{P}_2\mathbf{D}_2(\mathbf{P}_1\mathbf{D}_1)^{-1}\mathbf{B}_1 = \mathbf{P}_2\mathbf{D}_2\mathbf{D}_1^{-1}\mathbf{P}_1^\top\mathbf{B}_1$. Let $\mathbf{P}_\pi = \mathbf{P}_2\mathbf{P}_1^\top$ and $\mathbf{D} = \mathbf{P}_1\mathbf{D}_2\mathbf{D}_1^{-1}\mathbf{P}_1^\top$, it follows that $\mathbf{P}_\pi$ is a permutation matrix, $\mathbf{D}$ is a non-singular diagonal matrix, and $\mathbf{B}_2 = \mathbf{P}_\pi\mathbf{D}\mathbf{B}_1$. Multiplying on the right by $\mathbf{A}^{-1}$ (recall $\mathbf{B}_k = \mathbf{W}_k\mathbf{A}$) yields $\mathbf{W}_2 = \mathbf{P}_\pi\mathbf{D}\mathbf{W}_1$.

($\Longleftarrow$) Sufficiency. Assume (ii) holds: $\mathbf{W}_2 = \mathbf{P}_\pi\mathbf{D}\mathbf{W}_1$ with $\mathbf{P}_\pi$ a permutation matrix and $\mathbf{D}$ a non-singular diagonal matrix. Let $\mathbf{L}$ be any latent vector for which $\mathbf{W}_1$ is a $p$-order CICA solution (which exists by assumption that $\mathbf{W}_1$ is a CICA solution). Then for every given condition $\mathbf{L} = \ell$,

$$\mathbf{Z}^{(2)} \mid \{\mathbf{L} = \ell\} = \mathbf{W}_2\mathbf{X} \mid \{\mathbf{L} = \ell\} = \mathbf{P}_\pi\mathbf{D}\mathbf{W}_1\mathbf{X} \mid \{\mathbf{L} = \ell\} = \mathbf{P}_\pi\mathbf{D}\mathbf{Z}^{(1)} \mid \{\mathbf{L} = \ell\}.$$

Since permutation and nonzero per-coordinate scaling preserve mutual independence of coordinates, the coordinates of $\mathbf{Z}^{(2)}$ are mutually independent given $\mathbf{L}$ whenever those of $\mathbf{Z}^{(1)}$ are. Hence (i) holds. Therefore, the two statements are equivalent. $\qquad\square$

### A.3.6 PROOF OF LEMMA 3

**Lemma 3.** *Let $\mathbf{X}$ be $m$ observed variables, and $\mathbf{W}$ be a $p$-order CICA solution of $\mathbf{X}$. Let $\mathbf{X}' = \mathbf{W}\mathbf{X}$, $\mathbf{Y}$ and $\mathbf{Z}$ are two subsets of $\mathbf{X}'$, then if $\max\{|\mathbf{Y}|, |\mathbf{Z}|\} > p$, $\omega_1^\top\mathbf{Y}' \perp\!\!\!\perp \omega_2^\top\mathbf{Z}'$ has a non-zero solution $(\omega_1, \omega_2)$ for $(\mathbf{Y}', \mathbf{Z}')$, where $\mathbf{Y}' = \{X_i \mid \sum_{X_k \in \mathbf{Y}} \mathbf{W}_{k,i} \neq 0\}$, $\mathbf{Z}'$ are defined similarly.*

*Proof.* $\mathbf{Y} = \mathbf{W}_{\mathbf{Y},:}\mathbf{Y}'$, $\mathbf{Z} = \mathbf{W}_{\mathbf{Z},:}\mathbf{Z}'$, $\mathbf{Y}$ and $\mathbf{Z}$ are conditional independent given $p$ latent variables. Since $\max\{|\mathbf{Y}|, |\mathbf{Z}|\} > p$, without losing generality, we assume $|\mathbf{Y}| > p$. Then we can find a non-zero $\omega_1$ that $\omega_1^\top\mathbf{Y} \perp\!\!\!\perp \mathbf{Z}$. By Lemma 1, there exist a non-zero $\omega_2$ that makes $\omega_1^\top\mathbf{Y} \perp\!\!\!\perp \omega_2^\top\mathbf{Z}$. Thus, $\omega_1^\top\mathbf{W}_{\mathbf{Y},:}\mathbf{Y}' \perp\!\!\!\perp \omega_2^\top\mathbf{W}_{\mathbf{Z},:}\mathbf{Z}'$. $\qquad\square$

### A.3.7 PROOF OF LEMMA 4

**Lemma 4.** *For an observed variable set $\mathbf{X}$ with $|\mathbf{X}| = m$, denote $p = p_{min}(\mathbf{X})$. Suppose $m \geq 2p + 2$, and set $\mathbf{X}' := \mathbf{W}\mathbf{X}$, then $\mathbf{W}$ is a $p$-order CICA solution of $\mathbf{X}$ if and only if for every pair of disjoint coordinate subsets $\mathbf{X_1}, \mathbf{X_2}$ of $\mathbf{X}'$ with $|\mathbf{X_1}| = |\mathbf{X_2}| = p + 1$, $\det(\Sigma_{\mathbf{X_1},\mathbf{X_2}}) = 0$, where $\Sigma := \mathrm{Cov}(\mathbf{X}')$ denotes the covariance matrix on $\mathbf{X}'$ and $\Sigma_{\mathbf{X_1},\mathbf{X_2}}$ is the $(p+1) \times (p+1)$ sub-matrix of $\Sigma$ with rows indexed by $\mathbf{X_1}$ and columns by $\mathbf{X_2}$.*

*Proof.* ($\Rightarrow$) Necessity. If $\mathbf{W}$ is a $p$-order CICA solution, there exist a $p$-dimensional latent vector $\mathbf{L}$ and independent noises $\mathbf{E} = (E_1, \ldots, E_m)$ such that $Z_i = a_i^\top\mathbf{L} + E_i$, $i = 1, \ldots, m$. Hence,

$$\Sigma = \underbrace{A\Sigma_{\mathbf{L}}A^\top}_{\text{rank} \leq p} + \underbrace{(\mathrm{diag}(\mathrm{Var}(E_i)))}_{\text{diagonal}}.$$

For disjoint $\mathbf{X}_1, \mathbf{X}_2$ the diagonal term vanishes, so $\Sigma_{\mathbf{X}_1,\mathbf{X}_2} = A_{\mathbf{X}_1}\Sigma_{\mathbf{L}}A_{\mathbf{X}_2}^\top$ has rank at most $p$. Therefore $\det(\Sigma_{\mathbf{X}_1,\mathbf{X}_2}) = 0$ for every such pair.

($\Longleftarrow$) Sufficiency. Assume that for every disjoint $\mathbf{X}_1, \mathbf{X}_2$ of size $p + 1$, $\mathrm{rank}(\Sigma_{\mathbf{X}_1,\mathbf{X}_2}) \leq p$ (equivalently, all $(p+1)$-minors vanish). By the trek separation theorem, for each such pair there exists a $t$-separating pair $(\mathbf{L}_1, \mathbf{L}_2)$ with $|\mathbf{L}_1| + |\mathbf{L}_2| \leq p$ that $t$-separates $\mathbf{X}_1$ from $\mathbf{X}_2$. Since $p = p_{\min}(X)$, no separator of size $< p$ works uniformly; hence the minimum size is exactly $p$ for all these pairs.

Consider some $(\mathbf{X}_A, \mathbf{X}_B)$ with $X_A \cap X_b = \emptyset$ and both $|X_A|$ and $|X_b|$ equals $p + 1$, let $(\mathbf{L}_1, \mathbf{L}_2)$ be a minimal $t$-separator for this pair, so $|\mathbf{L}_1| + |\mathbf{L}_2| = p$. We claim that $(\mathbf{L}_1, \mathbf{L}_2)$ $t$-separates every other disjoint $(\mathbf{X}_C, \mathbf{X}_D)$ with $|\cdot| = p + 1$ and is minimal for that pair as well. Suppose $(\mathbf{L}_1, \mathbf{L}_2)$ does not $t$-separate $\mathbf{X}_C$ from $\mathbf{X}_D$. Then there exists a trek from some $c \in \mathbf{X}_C$ to $d \in \mathbf{X}_D$ avoiding $\mathbf{L}_1 \cup \mathbf{L}_2$. Because $|\mathbf{L}_1| + |\mathbf{L}_2| = p$ while $|\mathbf{X}_A| = |\mathbf{X}_B| = p + 1$, Menger's theorem for treks imply that there are $p$ vertex-disjoint treks connecting $\mathbf{X}_A \setminus \{a\}$ to $\mathbf{X}_B \setminus \{b\}$ for some $a \in \mathbf{X}_A$, $b \in \mathbf{X}_B$, all avoiding $\mathbf{L}_1 \cup \mathbf{L}_2$. Together with the trek $c \rightsquigarrow d$ (also avoiding $\mathbf{L}_1 \cup \mathbf{L}_2$) we obtain $p + 1$ vertex-disjoint treks between the modified sets $\mathbf{X}'_A = (\mathbf{X}_A \setminus \{a\}) \cup \{c\}$ and $\mathbf{X}'_B = (\mathbf{X}_B \setminus \{b\}) \cup \{d\}$,

hence $\text{rank}(\Sigma_{\mathbf{X}'_A, \mathbf{X}'_B}) \geq p + 1$, contradicting our hypothesis. Thus $(\mathbf{L}_1, \mathbf{L}_2)$ $t$-separates every such pair. If for some $(\mathbf{X}_C, \mathbf{X}_D)$ there were a smaller separator $(\mathbf{L}'_1, \mathbf{L}'_2)$ with $|\mathbf{L}'_1| + |\mathbf{L}'_2| < p$, then $\text{rank}(\Sigma_{\mathbf{X}_C, \mathbf{X}_D}) \leq |\mathbf{L}'_1| + |\mathbf{L}'_2| < p$, again contradicting the assumption that all those ranks equal $p$ by minimality of $p = p_{\min}(X)$. Hence the same $(\mathbf{L}_1, \mathbf{L}_2)$ is a minimal $t$-separator (of size $p$) for every such pair. We fix this global separator $(\mathbf{L}_1, \mathbf{L}_2)$.

Let $i \neq j$ be nodes outside $\mathbf{L}_1 \cup \mathbf{L}_2$. If there existed a trek from $i$ to $j$ avoiding $\mathbf{L}_1 \cup \mathbf{L}_2$, then by the same replacement argument as above we could build $p+1$ vertex-disjoint treks between some disjoint $(p+1)$-subsets, forcing a cross-rank $\geq p + 1$, which is a contradiction. Therefore, every trek from $i$ to $j$ meets $\mathbf{L}_1 \cup \mathbf{L}_2$, i.e., all covariance between distinct observed coordinates flows through $(\mathbf{L}_1, \mathbf{L}_2)$. Equivalently, $\Sigma$ admits a decomposition $\Sigma = A\Sigma_{\mathbf{L}}A^\top + D$ with $\text{rank}(A\Sigma_{\mathbf{L}}A^\top) \leq p$ and diagonal $D$ collecting variances.

Under Assumption 1, noises are mutually independent and independent of the latents. The diagonal $D$ found in Step 2 implies that each observed coordinate has a unique private noise, and distinct coordinates share no private noise. Therefore, for some $p$-vector $\mathbf{L}$, $Z_i = a_i^\top \mathbf{L} + E_i$, $\mathbf{E} = (E_1, \ldots, E_m)$ mutually independent, $\mathbf{E} \perp\!\!\!\perp \mathbf{L}$. Thus, the coordinates of $Z$ are mutually independent given $\mathbf{L}$, i.e., $\mathbf{W}$ is a $p$-order CICA solution.

Combining both directions proves the equivalence. $\qquad\square$

### A.3.8 PROOF OF LEMMA 5

**Lemma 5.** *For an observed variable set $\mathbf{X}$ with $|\mathbf{X}| = m$, suppose that $p_{min}(\mathbf{X}) = 1$ and $m \geq 3$ hold, set $\mathbf{X}' \triangleq \mathbf{W}\mathbf{X}$, then the invertible matrix $\mathbf{W}$ is a 1-order CICA solution of $\mathbf{X}$ if and only if for every ordered triple $(X'_i, X'_j, X'_k)$ of $\mathbf{X}'$, $\{X'_i, X'_j\}$ and $X'_k$ satisfies the Triad constraint.*

*Proof.* ($\Rightarrow$) Necessity. If $\mathbf{W}$ is a 1-order CICA solution of $\mathbf{X}$, then there exists some latent $L$ that we have $X'_i = \tilde{m}_i L + \tilde{E}_i$ with $\tilde{E} = (\tilde{E}_1, \ldots, \tilde{E}_m)$ mutually independent and $\tilde{E} \perp\!\!\!\perp L$. For $i \neq j \neq k$, $\text{Cov}(X'_j, X'_k) = \tilde{m}_j \tilde{m}_k \text{Var}(L)$ and hence the pseudo-residual of $\{X_i, X_j\}$ relative to $X_k$ as

$$E_{(i,j|k)} := \text{Cov}(X'_j, X'_k) \cdot X'_i - \text{Cov}(X'_i, X'_k) \cdot X'_j$$
$$= \tilde{m}_k \text{Var}(L)\,(\tilde{m}_j X'_i - \tilde{m}_i X'_j) = \tilde{m}_k \text{Var}(L)\,(\tilde{m}_j \tilde{E}_i - \tilde{m}_i \tilde{E}_j),$$

which depends only on $(\tilde{E}_i, \tilde{E}_j)$ and is independent of $X'_k = \tilde{m}_k L + \tilde{E}_k$. Thus the Triad constraint holds for all distinct triples.

($\Leftarrow$) Sufficiency. Assume the Triad constraint holds for every distinct $(i, j, k)$.

Suppose, for the sake of contradiction, that there exist two variables $X'_i$ and $X'_k$ that share at least two independent components. Since the Triad constraint $E_{(i,j|k)} \perp\!\!\!\perp X'_k$ holds for any $j \notin \{i, k\}$, $X'_j$ must also contain the components shared by $X'_i$ and $X'_k$. Consequently, every pair of variables in $\mathbf{X}'$ shares at least two identical independent components. Let us denote two of these shared components as $L_1$ and $L_2$. Each $X'_i$ can then be expressed as $X'_i = \tilde{m}_i L_1 + \tilde{n}_i L_2 + \tilde{E}_i$, where $\tilde{E}_i$, $L_1$, and $L_2$ are mutually independent (note that the residual terms $\tilde{E}_i$ may be dependent across different variables). Varying the triplet $(i, j, k)$, we have:

$$E_{(i,j|k)} := \text{Cov}(X'_j, X'_k) \cdot X'_i - \text{Cov}(X'_i, X'_k) \cdot X'_j$$
$$= (\tilde{m}_j \tilde{m}_k \text{Var}(L_1) + \tilde{n}_j \tilde{n}_k \text{Var}(L_2))\,(\tilde{m}_i L_1 + \tilde{n}_i L_2 + \tilde{E}_i)$$
$$- (\tilde{m}_i \tilde{m}_k \text{Var}(L_1) + \tilde{n}_i \tilde{n}_k \text{Var}(L_2))\,(\tilde{m}_j L_1 + \tilde{n}_j L_2 + \tilde{E}_j)$$
$$= (\tilde{n}_j \tilde{m}_i - \tilde{n}_i \tilde{m}_j)\tilde{n}_k \text{Var}(L_2)L_1 + (\tilde{m}_j \tilde{n}_i - \tilde{m}_i \tilde{n}_j)\tilde{m}_k \text{Var}(L_1)L_2 + \cdots$$

The independence condition $E_{(i,j|k)} \perp\!\!\!\perp X'_k$ establishes that the coefficients of $L_1$ and $L_2$ must vanish, yielding $\tilde{n}_j \tilde{m}_i - \tilde{n}_i \tilde{m}_j = 0$ for any pair $(i, j)$. This implies that the vectors $(\tilde{m}_i, \tilde{n}_i)$ are collinear, meaning $\frac{\tilde{m}_1}{\tilde{n}_1} = \frac{\tilde{m}_2}{\tilde{n}_2} = \cdots = \frac{\tilde{m}_m}{\tilde{n}_m}$ (assuming non-zero denominators). This directly contradicts the rank faithfulness assumption. Therefore, any two variables $X'_i$ and $X'_k$ share at most one component.

Next, suppose that two variables $X'_i$ and $X'_k$ are independent. As $E_{(i,j|k)} \perp\!\!\!\perp X'_k$ holds for any $j \notin \{i, k\}$, it follows that $X'_j \perp\!\!\!\perp X'_k$. By extension, the variables in $\mathbf{X}'$ are pairwise independent.

According to the definition of CICA solution, this implies $p_{\min}(\mathbf{X}) = 0$, which contradicts the prerequisite $p_{\min}(\mathbf{X}) = 1$. Consequently, any pair of variables in $\mathbf{X}'$ must be dependent.

From the above, any pair of variables must be dependent but can share at most one component; thus, they must share exactly one independent component. Let us denote the single independent component shared by $X_i'$ and $X_j'$ as $L$. Following the previous logic, all other variables must also contain $L$. Because any pair of variables in $\mathbf{X}'$ is dependent, we can write $X_i' = \tilde{m}_i L + \tilde{E}_i$, where $\tilde{\mathbf{E}} = (\tilde{E}_1, \ldots, \tilde{E}_m)$ are mutually independent and $\tilde{\mathbf{E}} \perp\!\!\!\perp L$. Therefore, by definition, $\mathbf{W}$ is a 1-order CICA solution for $\mathbf{X}$.

Combining both directions proves the claim. $\qquad\square$

### A.3.9 PROOF OF LEMMA 6

**Lemma 6.** $\mathbf{I} - \mathbf{B}_{\mathbf{X},\mathbf{X}}$ *is a* $p_{min}(\mathbf{X})$*-order CICA solution of* $\mathbf{X}$ *with latent conditional set* $\mathrm{LPa}(\mathbf{X})$.

*Proof.* In the setting of our paper, $\mathbf{A}_{\mathbf{X},\mathbf{X}}^{-1} = \mathbf{I} - \mathbf{B}_{\mathbf{X},\mathbf{X}}$. In the proof of Theorem 3, we prove that $\mathbf{A}_{\mathbf{X},\mathbf{X}}^{-1}\mathbf{X}$ deletes all the outgoing edges from $\mathbf{X}$ graphically. Therefore, $\mathbf{A}_{\mathbf{X},\mathbf{X}}^{-1}\mathbf{X}$ is conditional independent given $\mathrm{LPa}(\mathbf{X})$. Given Condition 1 holds, $p_{min}(\mathbf{X}) = |\mathrm{LPa}(\mathbf{X})|$. Thus, $\mathbf{I} - \mathbf{B}_{\mathbf{X},\mathbf{X}}$ is a $p_{min}(\mathbf{X})$-order CICA solution of $\mathbf{X}$ with latent conditional set $\mathrm{LPa}(\mathbf{X})$. $\qquad\square$

### A.3.10 PROOF OF LEMMA 7

**Lemma 7.** *Suppose* $\mathbf{W}$ *is a* $p_{min}(\mathbf{X})$*-order CICA solution of* $\mathbf{X}$ *whose latent conditional set is* $\mathrm{LPa}(\mathbf{X})$*, then* $\mathbf{W} \sim \mathbf{I} - \mathbf{B}_{\mathbf{X},\mathbf{X}}$ *and we can uniquely recover the true causal matrix among observed variables* $\mathbf{B}_{\mathbf{X},\mathbf{X}}$ *from* $\mathbf{W}$.

*Proof.* By Lemma 2 and Lemma 6, we can find a permutation matrix $\mathbf{P}$ and non-singular diagonal matrix $\mathbf{D}$ that makes $\mathbf{W} = \mathbf{PD}(\mathbf{I} - \mathbf{B}_{\mathbf{X},\mathbf{X}})$. Subsequent proofs can be analogized to Lemma 1 in (Shimizu et al., 2006). $\qquad\square$

### A.3.11 PROOF OF LEMMA 8

**Lemma 8.** $\mathbf{I} - \mathbf{B}_{\mathbf{X},\mathbf{X}} \in \arg\min\{\|\mathbf{W}\|_0 : \mathbf{W}$ *is a* $p_{min}(\mathbf{X})$*-order CICA solution of* $\mathbf{X}\}$. .

*Proof.* For a vector $\omega \in \mathbb{R}^m$, denote

$$\mathbf{X}' = \alpha^\top \mathbf{E} = \omega^\top \mathbf{X} = \omega^\top \mathbf{A}\mathbf{E} = \omega^\top(\mathbf{I} - \mathbf{B}_{\mathbf{X},\mathbf{X}})^{-1}\mathbf{E}, \qquad \text{where } \alpha \in \mathbb{R}^{m+d}.$$

Since $\mathbf{A}_{\mathbf{X},\mathbf{X}} = (\mathbf{I} - \mathbf{B}_{\mathbf{X},\mathbf{X}})^{-1}$ is a non-singular matrix, let us denote the row indices corresponding to $\mathbf{X}$ as $\alpha^{\mathbf{X}}$ for convenience. We then have $\mathbf{A}_{\mathbf{X},\mathbf{X}}^\top \omega = \alpha^{\mathbf{X}}$, which yields

$$\omega = \mathbf{A}_{\mathbf{X},\mathbf{X}}^{-T}\alpha^{\mathbf{X}} = (\mathbf{I} - \mathbf{B}_{\mathbf{X},\mathbf{X}})^\top \alpha^{\mathbf{X}} = \alpha^{\mathbf{X}} - \mathbf{B}_{\mathbf{X},\mathbf{X}}^\top \alpha^{\mathbf{X}}.$$

Defining $\alpha^{\mathbf{L}}$ similarly, we obtain

$$\alpha^{\mathbf{L}} = \mathbf{A}_{\mathbf{L},\mathbf{X}}^\top \omega = \mathbf{A}_{\mathbf{L},\mathbf{X}}^\top (\mathbf{I} - \mathbf{B}_{\mathbf{X},\mathbf{X}})^\top \alpha^{\mathbf{X}}.$$

In summary, we can represent $\omega$ and $\alpha^{\mathbf{L}}$ as linear combinations of $\alpha^{\mathbf{X}}$:

$$\left[\frac{\alpha^{\mathbf{L}}}{\alpha^{\mathbf{X}}}\right] = \left[\frac{\mathbf{A}_{\mathbf{L},\mathbf{X}}^\top}{\mathbf{A}_{\mathbf{X},\mathbf{X}}^\top}\right]\omega \quad\implies\quad \left[\frac{\alpha^{\mathbf{L}}}{\omega}\right] = \left[\frac{\mathbf{A}_{\mathbf{L},\mathbf{X}}^\top \mathbf{A}_{\mathbf{X},\mathbf{X}}^{-\mathbf{T}}}{\mathbf{A}_{\mathbf{X},\mathbf{X}}^{-\mathbf{T}}}\right]\alpha^{\mathbf{X}} \tag{5}$$

Here, since we focus mainly on the sparsity of $\mathbf{W}$ (i.e., $\|\mathbf{W}\|_0$) rather than its specific values, we use 0 to represent a value of 0, and $\times$ to represent a nonzero value, following (Ghassami et al., 2020).

For $\mathbf{W} = \mathbf{A}_{\mathbf{X},\mathbf{X}}^{-1}$, denoting its corresponding noise coefficients of observed variables as $\alpha_{\mathbf{X}} = [\alpha_{\mathbf{X},1}, \alpha_{\mathbf{X},2}, \cdots, \alpha_{\mathbf{X},m}]$, we have $\alpha_{\mathbf{X},i} = [\underbrace{0, \cdots, 0}_{(i-1)\text{-times}}, \times, \underbrace{0, \cdots, 0}_{(m-i)\text{-times}}]^\top$ and $\alpha_{\mathbf{L},i} = [\times, \cdots, \times]^\top$.

Any other feasible solution, aside from $\mathbf{A}_{\mathbf{X},\mathbf{X}}^{-1}$, corresponds to choosing $d$ variables from the $d + m$ independent noises.

If $\mathbf{W}$ is a 1-order CICA solution whose latent conditional set is $\mathrm{LPa}(\mathbf{X})$, we can find a permutation matrix $\mathbf{P}$ and a non-singular diagonal matrix $\mathbf{D}$ such that $\mathbf{W} = \mathbf{PDA}_{\mathbf{X},\mathbf{X}}^{-1}$. Because $\mathbf{P}$ and $\mathbf{D}$ do not alter the sparsity pattern of $\mathbf{W}$, we can analyze $\mathbf{W} = \mathbf{A}_{\mathbf{X},\mathbf{X}}^{-1}$ directly for convenience.

For the $j$-th row of $\mathbf{W}$, there is exactly one $\times$ in the $j$-th column of the coefficient matrix $\alpha^{\mathbf{X}}$ (specifically at $\alpha_{:,j}^{\mathbf{X}}$). Because $\mathbf{W}$ acts as the unmixing matrix where $\mathbf{W}_{j,:} = \omega_j^\top$, we have

$$\mathbf{W}_{j,:} = (\alpha_{:,j}^{\mathbf{X}})^\top (\mathbf{I} - \mathbf{B}_{\mathbf{X},\mathbf{X}}).$$

For any $t \in [m]$,

$$\mathbf{W}_{j,t} = \sum_s \alpha_{s,j}^{\mathbf{X}} (\mathbf{I}_{s,t} - \mathbf{B}_{s,t}) = \alpha_{j,j}^{\mathbf{X}} (\mathbf{I}_{j,t} - \mathbf{B}_{j,t}). \tag{6}$$

Case (i): If $t = j$ ($\mathbf{I}_{j,j} \neq 0$), then $\mathbf{W}_{j,j} = \alpha_{j,j}^{\mathbf{X}} = \times$.

Case (ii): If $X_t \in \mathrm{Pa}(X_j)$ ($\mathbf{B}_{j,t} \neq 0$), then $\mathbf{W}_{j,t} = -\alpha_{j,j}^{\mathbf{X}} \mathbf{B}_{j,t} = \times$.

Case (iii): If $X_t$ does not fall into either of the two cases above, then $\mathbf{W}_{j,t} = 0$.

In summary,

$$\forall t \in [m], \mathbf{W}_{j,t} \neq 0 \iff X_t \in \{X_j\} \cup \mathrm{Pa}(X_j). \tag{7}$$

Now, if $\mathbf{W}$ is a 1-order CICA solution whose latent conditional set is not $\mathrm{LPa}(\mathbf{X})$, then $\mathbf{X}'$ is conditionally independent given a latent variable other than $L$. Without loss of generality, assume $\mathbf{X}'$ is conditionally independent given the exogenous noise of $X_k$, denoted $E_k$. Therefore, $E_k \in \mathrm{NS}(X_j')$ for any $j \in [m]$. In other words, the column vector $\alpha_{:,j}^{\mathbf{X}} \in \mathbb{R}^m$ has a $\times$ in its $k$-th position. Furthermore, exactly one column $\alpha_{:,j}^{\mathbf{X}}$ has a $\times$ in the $j$-th position for $j \in [m] \setminus \{k\}$. On the other hand, we have $\alpha^{\mathbf{L}} = [\underbrace{0, \cdots, 0}_{(k-1)\text{-times}}, \times, \underbrace{0, \cdots, 0}_{(m-k)\text{-times}}]$. Essentially, this scenario exchanges the positions of $E_L$ and $E_k$ compared to $\mathbf{W} \sim \mathbf{A}_{\mathbf{X},\mathbf{X}}^{-1}$.

For example, Equation (8) presents an example of the matrix $\alpha^{\mathbf{X}}$ when $k = 3$.

$$\alpha^{\mathbf{X}} = \begin{bmatrix} \times & 0 & 0 & 0 & \cdots & 0 \\ 0 & \times & 0 & 0 & \cdots & 0 \\ \times & \times & \times & \times & \cdots & \times \\ 0 & 0 & 0 & \times & \cdots & 0 \\ \vdots & \vdots & \vdots & \vdots & \ddots & \vdots \\ 0 & 0 & 0 & 0 & \cdots & \times \end{bmatrix} \tag{8}$$

Based on this structure of $\alpha^{\mathbf{X}}$, we can now check the sparsity of each row of $\mathbf{W}$.

For the $k$-th row of $\mathbf{W}$, we have exactly one $\times$ in $\alpha_{:,k}^{\mathbf{X}}$, located in the $k$-th row (i.e., $\alpha_{k,k}^{\mathbf{X}}$). In addition, $\alpha_{:,k}^{\mathbf{L}} = \times$. Therefore, the support of $\alpha_{:,k}^{\mathbf{L}}$ is exactly the same as in the scenario $\mathbf{W} \sim \mathbf{A}_{\mathbf{X},\mathbf{X}}^{-1}$.

For the $j$-th row of $\mathbf{W}$ with $j \neq k$, we have two $\times$ in $\alpha_{:,j}^{\mathbf{X}}$, located in the $j$-th and $k$-th rows (i.e., $\alpha_{j,j}^{\mathbf{X}}$ and $\alpha_{k,j}^{\mathbf{X}}$), respectively. Since $\mathbf{W}_{j,:} = (\alpha_{:,j}^{\mathbf{X}})^\top (\mathbf{I} - \mathbf{B}_{\mathbf{X},\mathbf{X}})$, for any $t \in [m]$,

$$\mathbf{W}_{j,t} = \sum_s \alpha_{s,j}^{\mathbf{X}} (\mathbf{I}_{s,t} - \mathbf{B}_{s,t}) = \alpha_{j,j}^{\mathbf{X}} (\mathbf{I}_{j,t} - \mathbf{B}_{j,t}) + \alpha_{k,j}^{\mathbf{X}} (\mathbf{I}_{k,t} - \mathbf{B}_{k,t}). \tag{9}$$

Case (i): If $t = j$ ($\mathbf{I}_{j,j} \neq 0, \mathbf{I}_{k,j} = 0$), then

$$\mathbf{W}_{j,j} = \alpha_{j,j}^{\mathbf{X}} (\mathbf{I}_{j,j} - \mathbf{B}_{j,j}) + \alpha_{k,j}^{\mathbf{X}} (\mathbf{I}_{k,j} - \mathbf{B}_{k,j}) = \alpha_{j,j}^{\mathbf{X}} - \alpha_{k,j}^{\mathbf{X}} \mathbf{B}_{k,j}.$$

On the other hand, we have the following.

$$\begin{aligned}
\alpha_{:,j}^{\mathbf{L}} &= \mathbf{A}_{\mathbf{L},\mathbf{X}}^{\top}(\mathbf{I} - \mathbf{B}_{\mathbf{X},\mathbf{X}}^{\top})\alpha_{:,j}^{\mathbf{X}} \\
&= \mathbf{A}_{\mathbf{L},\mathbf{X}}^{\top}(\mathbf{I}_{:,j} - \mathbf{B}_{\mathbf{X},j}^{\top})\alpha_{j,j}^{\mathbf{X}} + \mathbf{A}_{\mathbf{L},\mathbf{X}}^{\top}(\mathbf{I}_{:,k} - \mathbf{B}_{\mathbf{X},k}^{\top})\alpha_{k,j}^{\mathbf{X}} \\
&= (\mathbf{A}_{\mathbf{L},j}^{\top} - \mathbf{A}_{\mathbf{L},\mathbf{X}}^{\top}\mathbf{B}_{\mathbf{X},j}^{\top})\alpha_{j,j}^{\mathbf{X}} + (\mathbf{A}_{\mathbf{L},k}^{\top} - \mathbf{A}_{\mathbf{L},\mathbf{X}}^{\top}\mathbf{B}_{\mathbf{X},k}^{\top})\alpha_{k,j}^{\mathbf{X}} \\
&= 0
\end{aligned}$$

If $\mathbf{W}_{j,j} = 0$, given that $\alpha_{j,j}^{\mathbf{X}}$ and $\alpha_{k,j}^{\mathbf{X}}$ are non-zero, the following system of equations must have a non-zero solution $x_1 = \alpha_{j,j}^{\mathbf{X}}, x_2 = \alpha_{k,j}^{\mathbf{X}}$:

$$\begin{cases}
x_1 - & \mathbf{B}_{k,j}x_2 = 0 \\
(\mathbf{A}_{\mathbf{L},j}^{\top} - \mathbf{A}_{\mathbf{L},\mathbf{X}}^{\top}\mathbf{B}_{\mathbf{X},j}^{\top})x_1 + & (\mathbf{A}_{\mathbf{L},k}^{\top} - \mathbf{A}_{\mathbf{L},\mathbf{X}}^{\top}\mathbf{B}_{\mathbf{X},k}^{\top})x_2 = 0
\end{cases} \tag{10}$$

This requires the determinant of the coefficient matrix to be zero:

$$(\mathbf{A}_{\mathbf{L},k}^{\top} - \mathbf{A}_{\mathbf{L},\mathbf{X}}^{\top}\mathbf{B}_{\mathbf{X},k}^{\top}) + (\mathbf{A}_{\mathbf{L},j}^{\top} - \mathbf{A}_{\mathbf{L},\mathbf{X}}^{\top}\mathbf{B}_{\mathbf{X},j}^{\top})\mathbf{B}_{k,j} = 0.$$

Here, $\mathbf{A}_{\mathbf{L},k}^{\top} - \mathbf{A}_{\mathbf{L},\mathbf{X}}^{\top}\mathbf{B}_{\mathbf{X},k}^{\top}$ measures the total causal effect of $L$ on $X_k$ without passing through other observed variables, while $(\mathbf{A}_{\mathbf{L},j}^{\top} - \mathbf{A}_{\mathbf{L},\mathbf{X}}^{\top}\mathbf{B}_{\mathbf{X},j}^{\top})\mathbf{B}_{k,j}$ measures the total causal effect of $L$ on $X_k$ that does not pass through $\mathbf{X} \setminus \{X_j\}$ and ends with $X_j \to X_k$. Therefore, this sum implies the causal effect of $L$ on $X_k$ is zero given all observed variables other than $X_j$ and $X_k$. In other words, $L \perp\!\!\!\perp X_k \mid \mathbf{X} \setminus \{X_k, X_j\}$ and $\text{Rank}(\Sigma_{L,X_k|\mathbf{X}\setminus\{X_k,X_j\}}) = 0$. However, this rank constraint is not generic and violates the rank faithfulness assumption. Therefore, we must have $\mathbf{W}_{j,j} \neq 0$, arriving at a contradiction.

Case (ii): If $t = k$ ($\mathbf{I}_{j,k} = 0, \mathbf{I}_{k,k} \neq 0$), then

$$\mathbf{W}_{j,k} = \alpha_{j,j}^{\mathbf{X}}(\mathbf{I}_{j,k} - \mathbf{B}_{j,k}) + \alpha_{k,j}^{\mathbf{X}}(\mathbf{I}_{k,k} - \mathbf{B}_{k,k}) = -\alpha_{j,j}^{\mathbf{X}}\mathbf{B}_{j,k} + \alpha_{k,j}^{\mathbf{X}}.$$

Similar to Case (i), if $\mathbf{W}_{j,k} = 0$, we have

$$(\mathbf{A}_{\mathbf{L},j}^{\top} - \mathbf{A}_{\mathbf{L},\mathbf{X}}^{\top}\mathbf{B}_{\mathbf{X},j}^{\top}) + (\mathbf{A}_{\mathbf{L},k}^{\top} - \mathbf{A}_{\mathbf{L},\mathbf{X}}^{\top}\mathbf{B}_{\mathbf{X},k}^{\top})\mathbf{B}_{j,k} = 0,$$

which means the causal effect of $L$ on $X_j$ is zero given all observed variables other than $X_j$ and $X_k$. This implies $\text{Rank}(\Sigma_{L,X_j|\mathbf{X}\setminus\{X_k,X_j\}}) = 0$. As this violates the rank faithfulness assumption, we have $\mathbf{W}_{j,k} \neq 0$.

Case (iii): If $X_t \in \text{Pa}(X_j) \setminus \{X_k\}$ ($\mathbf{B}_{j,t} \neq 0$), then $\mathbf{W}_{j,t} = -\alpha_{j,j}^{\mathbf{X}}\mathbf{B}_{j,t} - \alpha_{k,j}^{\mathbf{X}}\mathbf{B}_{k,t}$. Similar to Case (i), if $\mathbf{W}_{j,t} = 0$, we have

$$(\mathbf{A}_{\mathbf{L},k}^{\top} - \mathbf{A}_{\mathbf{L},\mathbf{X}}^{\top}\mathbf{B}_{\mathbf{X},k}^{\top})\mathbf{B}_{j,t} = (\mathbf{A}_{\mathbf{L},j}^{\top} - \mathbf{A}_{\mathbf{L},\mathbf{X}}^{\top}\mathbf{B}_{\mathbf{X},j}^{\top})\mathbf{B}_{k,t}.$$

This implies $\text{Rank}(\Sigma_{\{L,X_t\},\{X_k,X_j\}|\mathbf{X}\setminus\{X_t,X_k,X_j\}}) = 1$, which violates the rank faithfulness assumption. Therefore, $\mathbf{W}_{j,t} \neq 0$.

Case (iv): If $X_t \in \text{Pa}(X_k) \setminus \{X_j\}$ ($\mathbf{B}_{k,t} \neq 0$), then $\mathbf{W}_{j,t} = -\alpha_{j,j}^{\mathbf{X}}\mathbf{B}_{j,t} - \alpha_{k,j}^{\mathbf{X}}\mathbf{B}_{k,t}$. Similar to Case (iii), if $\mathbf{W}_{j,t} = 0$, we obtain

$$(\mathbf{A}_{\mathbf{L},k}^{\top} - \mathbf{A}_{\mathbf{L},\mathbf{X}}^{\top}\mathbf{B}_{\mathbf{X},k}^{\top})\mathbf{B}_{j,t} = (\mathbf{A}_{\mathbf{L},j}^{\top} - \mathbf{A}_{\mathbf{L},\mathbf{X}}^{\top}\mathbf{B}_{\mathbf{X},j}^{\top})\mathbf{B}_{k,t}.$$

Thus, we can prove $\mathbf{W}_{j,t} \neq 0$ because $\text{Rank}(\Sigma_{\{L,X_t\},\{X_k,X_j\}|\mathbf{X}\setminus\{X_t,X_k,X_j\}}) = 1$ violates the rank faithfulness assumption.

Case (v): If $X_t$ does not fall into any of the four cases above, then $\mathbf{W}_{j,t} = 0$.

Therefore, from Case (i)$\sim$(v), we have

$$\forall t \in [m] \setminus \{k\}, \mathbf{W}_{j,t} \neq 0 \iff X_t \in \{X_j, X_k\} \cup \text{Pa}(X_j) \cup \text{Pa}(X_k). \tag{11}$$

Combined with the scenario $t = k$, in summary, we have

$$\forall t \in [m], \mathbf{W}_{j,t} \neq 0 \iff X_t \in \{X_j, X_k\} \cup \text{Pa}(X_j) \cup \text{Pa}(X_k). \tag{12}$$

Since $\{X_j\} \cup \text{Pa}(X_j) \subseteq \{X_j, X_k\} \cup \text{Pa}(X_j) \cup \text{Pa}(X_k)$, we have $\|\mathbf{A}_{\mathbf{X},\mathbf{X}}^{-1}\|_0 \leq \|\mathbf{W}\|_0$. Therefore,

$$\mathbf{I} - \mathbf{B}_{\mathbf{X},\mathbf{X}} \in \arg\min\{\|\mathbf{W}\|_0 : \mathbf{W} \text{ is a } p_{min}(\mathbf{X})\text{-order CICA solution of } \mathbf{X}\}.$$

$\square$

### A.3.12    PROOF OF LEMMA 9

**Lemma 9**. *If Condition 2 holds, $\mathbf{W} \in \arg\min\{\|\tilde{\mathbf{W}}\|_0 : \tilde{\mathbf{W}}$ is a $p_{min}(\mathbf{X})$-order CICA solution of $\mathbf{X}\}$ if and only if $\mathbf{W} \sim \mathbf{I} - \mathbf{B}_{\mathbf{X},\mathbf{X}}$.*

*Proof.* During the proof in Lemma 8 we obtain the following results. If $\mathbf{W}$ is a 1-order CICA solution whose latent conditional set is $\mathrm{LPa}(\mathbf{X})$, it follows that for any row index $j$,

$$\forall t \in [m], \mathbf{W}_{j,t} \neq 0 \iff X_t \in \{X_j\} \cup \mathrm{Pa}(X_j).$$

If $\mathbf{W}$ is a 1-order CICA solution whose latent conditional set is the exogenous noise of $X_k$, instead of $\mathrm{LPa}(\mathbf{X})$, it follows that for any row index $j$,

$$\forall t \in [m], \mathbf{W}_{j,t} \neq 0 \iff X_t \in \{X_j, X_k\} \cup \mathrm{Pa}(X_j) \cup \mathrm{Pa}(X_k).$$

If $X_k \in \mathrm{Pa}(X_j)$ and $\mathrm{Pa}(X_k) = \emptyset$, we have

$$\{X_j, X_k\} \cup \mathrm{Pa}(X_j) \cup \mathrm{Pa}(X_k) = \{X_j\} \cup \mathrm{Pa}(X_j),$$

Thus, $\mathbf{W}_{j,t}$ has the exact same sparsity pattern in two secnario. If Condition 2 holds, then there exists a $X_j$ such that the constraint $X_k \in \mathrm{Pa}(X_j)$ and $\mathrm{Pa}(X_k) = \emptyset$ does not hold,

$$\{X_j\} \cup \mathrm{Pa}(X_j) \subsetneq \{X_j, X_k\} \cup \mathrm{Pa}(X_j) \cup \mathrm{Pa}(X_k),$$

then the CICA solution whose latent conditional set is $\mathrm{LPa}(\mathbf{X})$ has a strictly small number of non-zero entries. $\qquad\square$

### A.3.13    PROOF OF THEOREM 1

**Theorem 1.** *All latent variables in $\mathrm{LPa}(\mathbf{X})$ can be identified. Besides, the causal edges of $\mathrm{LPa}(\mathbf{X})$ to $\mathbf{X}$ and the causal edges between the observed variables are also identifiable.*

*Proof.* By Lemma 8, if Condition 2 is satisfied, we can identify $\mathbf{I} - \mathbf{B}_{\mathbf{X},\mathbf{X}}$ by adding sparsity constraints and induce the causal structure. On the other hand, if Condition 2 is not satisfied, $\mathbf{I} - \mathbf{B}_{\mathbf{X},\mathbf{X}}$ is not identifiable. That is, we can find another $p$-order CICA solution $\mathbf{W}'$ with the same number of non-zero entries as $\mathbf{I} - \mathbf{B}_{\mathbf{X},\mathbf{X}}$. Review the results obtained in the proof of Lemma 8,

$$\forall t \in [m], \mathbf{W}_{j,t} \neq 0 \iff X_t \in \{X_j, X_k\} \cup \mathrm{Pa}(X_j) \cup \mathrm{Pa}(X_k).$$

If $X_k \in \mathrm{Pa}(X_j)$ and $\mathrm{Pa}(X_k) = \emptyset$,

$$\{X_j, X_k\} \cup \mathrm{Pa}(X_j) \cup \mathrm{Pa}(X_k) = \{X_j\} \cup \mathrm{Pa}(X_j),$$

then $\mathbf{W}_{t,j}$ has the exact same sparsity pattern. If the constraint $X_k \in \mathrm{Pa}(X_j)$ and $\mathrm{Pa}(X_k) = \emptyset$ holds for every $X_j$ (Condition 2 does not hold), then the whole $\mathbf{W}$ exist exactly the same sparsity pattern. In other words, although $\mathbf{W}'$ has different parameters with $\mathbf{I} - \mathbf{B}_{\mathbf{X},\mathbf{X}}$, their support matrix remains the same. Therefore, in both cases, the causal structure among observed variables $\mathbf{B}_{\mathbf{X},\mathbf{X}}$ within a causal cluster is identifiable. Given Condition 1 holds, $p_{min}(\mathbf{X}) = |\mathrm{LPa}(\mathbf{X})|$, thus we can identify each latent variable in $\mathrm{LPa}(\mathbf{X})$. Putting all these partial results together, all the latent variables in $\mathrm{LPa}(\mathbf{X})$, the causal edges of $\mathrm{LPa}(\mathbf{X})$ to $\mathbf{X}$ and the causal edges between the observed variables can be identified. $\qquad\square$

### A.3.14    PROOF OF LEMMA 10

**Lemma 10**. *Let $L$ be a latent variable discovered in the current iteration. Denote $\mathbf{S} = \mathrm{Ch}(L)$. Let $S_k$ have the highest causal order in $\mathbf{S}$ whose index in $\mathbf{S}$ is $k$, and $\mathbf{W}$ be the sparsest $p_{min}(\mathbf{S})$-order CICA solution of $\mathbf{S}$. $\mathbf{P}$ is the permutation matrix that makes $\mathbf{PW}$ have non-zero diagonal elements, simultaneously. Denote $\mathbf{Z} = \mathbf{PWS}$, then the value of $Z_k$ can be a suitable surrogate for $L$.*

*Proof.* By Lemma 8, if Condition 2 is satisfied, we can identify $\mathbf{I} - \mathbf{B}_{\mathbf{S},\mathbf{S}}$ by adding sparsity constraints and induce the causal structure. Then $\mathbf{PW}$ deletes all outgoing edges from $\mathbf{S}$ and makes $Z_k$ a pure child of $L$. As shown in (Xie et al., 2024), it can be a suitable surrogate for $L$. On the other hand, if Condition 2 is not satisfied, $\mathbf{I} - \mathbf{B}_{\mathbf{S},\mathbf{S}}$ is not identifiable. Review the results obtained in the proof of Lemma 8,

$$\forall t \in [m], \mathbf{W}_{j,t} \neq 0 \iff X_t \in \{X_j, X_k\} \cup \mathrm{Pa}(X_j) \cup \mathrm{Pa}(X_k).$$

If $X_k \in \mathrm{Pa}(X_j)$ and $\mathrm{Pa}(X_k) = \emptyset$,

$$\{X_j, X_k\} \cup \mathrm{Pa}(X_j) \cup \mathrm{Pa}(X_k) = \{X_j\} \cup \mathrm{Pa}(X_j),$$

thus $\mathbf{W}_{t,j}$ has the exactly same sparsity pattern. If the constraint $X_k \in \mathrm{Pa}(X_j)$ and $\mathrm{Pa}(X_k) = \emptyset$ holds for every $X_j$ (Condition 2 does not hold), then the whole $\mathbf{W}$ exist exactly same sparsity pattern. In other words, although $\mathbf{W}'$ has different parameters with $\mathbf{I} - \mathbf{B}_{\mathbf{S},\mathbf{S}}$, their support matrix remains the same. Essentially, $\mathbf{W}'\mathbf{S}$ can be interpreted as swapping the roles of $L$ and $E_k$ on $\mathbf{I} - \mathbf{B}_{\mathbf{S},\mathbf{S}}\mathbf{S}$. Although $L$ is not contained in the latent conditional set, it is still included in $Z_k$. Therefore, in both cases, $Z_k$ can be a suitable surrogate for $L$. $\qquad\square$

### A.3.15 PROOF OF THEOREM 2

**Theorem 2.** *If Condition 1 holds, then the underlying causal graph $G$ is fully identifiable, including both the location of latent variables $\mathbf{L}$ and the recovery of causal relationships $\mathcal{E}$.*

*Proof.* Denote $\mathrm{Dis}(V_i)$ the length of the longest direct path from $V_i$ to $\mathbf{X}$. $\forall X_i \in \mathbf{X}, \mathrm{Dis}(X_i) = 0$. We collect $\mathbf{Y_k} = \{V_i | \mathrm{Dis}(V_i) \leq k\}$. The proof is based on mathematical induction:

(1) Base: for $k = 1$, we use Theorem 1 to identify the common latent parents of observed variables and related causal edges. In other words, we can correctly identify the induced sub-graph of $G$ with nodes in $\mathbf{Y_1}$.

(2) Induction: assume we have correctly identified the induced sub-graph of $G$ with nodes in $\{V_i | \mathrm{Dis}(V_i) \leq k\}$, then using Lemma 10 to find the suitable surrogate for latent variables in $\mathbf{Y_k} \setminus \mathbf{Y_{k-1}}$, we can continue to use Theorem 1 to local the latent variables in $\mathbf{Y_{k+1}}$ and related causal edges, which concludes the induction.

Therefore, the underlying causal graph $G$ is fully identifiable, including both latent variables and their causal relationships. $\qquad\square$

### A.3.16 PROOF OF THEOREM 3

**Theorem 3.** *Let the graph obtained after removing all the outgoing edges of $\mathbf{X}$ in $\mathcal{G}$ be named by $\mathcal{G}'$, which form several connected components of observed variables $\mathbf{X}'_{C_1}, \mathbf{X}'_{C_2}, \cdots, \mathbf{X}'_{C_k}$, where $k$ be the number of connected components in $\mathcal{G}'$. For an ISA solution $\mathbf{W}$, let $\mathbf{W}\mathbf{X} = (\mathbf{Z}_1^\top, \ldots, \mathbf{Z}_k^\top)^\top$. Then there is a permutation $\pi$ of $[k]$ s.t. for any $i \in [k]$, $\exists \mathbf{W}_i \in \mathrm{GL}(|C_i|)$ makes $\mathbf{Z}_{\pi(i)} = \mathbf{W}_i \mathbf{X}'_{C_i}$.*

*Proof.* Based on the Schur complement, we have

$$\mathbf{A}_{\mathbf{X},\mathbf{X}}^{-1} = (\mathbf{I} - \mathbf{B}_{\mathbf{X},\mathbf{X}}) - \mathbf{B}_{\mathbf{X},\mathbf{L}}(\mathbf{I} - \mathbf{B}_{\mathbf{L},\mathbf{L}})^{-1}\mathbf{B}_{\mathbf{L},\mathbf{X}} \qquad (13)$$

Denote $\mathbf{Z} = \mathbf{A}_{\mathbf{X},\mathbf{X}}^{-1}\mathbf{X}$. Then we have $\mathrm{NS}(\mathbf{Z}_i) = \{E_j | L_j$ has a directed path to $X_i$ whose intermediate nodes, if exist, are all latent nodes$\}$. The reasons are as follows.

$$\begin{aligned}
Z_i &= \sum_{j=1}^{|X|} (\mathbf{A}_{\mathbf{X},\mathbf{X}}^{-1})_{i,j} X_j \\
&= \sum_{j=1}^{|X|} ((\mathbf{I} - \mathbf{B}_{\mathbf{X},\mathbf{X}}) - \mathbf{B}_{\mathbf{X},\mathbf{L}}(\mathbf{I} - \mathbf{B}_{\mathbf{L},\mathbf{L}})^{-1}\mathbf{B}_{\mathbf{L},\mathbf{X}})_{i,j} X_j \\
&= X_i - \sum_{X_j \in \mathbf{X} \setminus \{X_i\}} (\mathbf{B}_{i,j} + \mathbf{B}_{i,\mathbf{L}}(\mathbf{I} - \mathbf{B}_{\mathbf{L},\mathbf{L}})^{-1}\mathbf{B}_{\mathbf{L},j}) X_j
\end{aligned} \qquad (14)$$

Considering all directed paths into $X_i$, we categorize them into different groups according to the topologically last observed nodes before $X_i$ on this path. For example, if there is a path $P_1 : X_t \to L_1 \to X_k \to L_2 \to X_i$, we put this path into the group corresponding to $X_k$, named $\mathcal{G}[X_k]$. If there are no observed nodes before $X_i$ in this path, we put this path in the group corresponding to $\emptyset$, named $\mathcal{G}[\emptyset]$. In total, there are $|X|$ groups: $\bigcup_{X_k \in \mathbf{X} \setminus \{X_i\}} \mathcal{G}[X_k] \cup \mathcal{G}[\emptyset]$.

$X_i$ is a cumulative sum of all directed paths into $X_i$. The contribution of each directed path in this sum is the noise of the start point times the path coefficient. Obviously, any path will be placed

in the group $\bigcup_{X_k \in \mathbf{X} \setminus \{X_i\}} \mathcal{G}[X_k] \cup \mathcal{G}[\emptyset]$. Then, consider what the subtrahend in the last line of Equ. (14) denotes. $\mathbf{B}_{i,j}$ denotes the direct causal effect from $X_j$ to $X_i$, $\mathbf{B}_{i,\mathbf{L}}(\mathbf{I} - \mathbf{B}_{\mathbf{L},\mathbf{L}})^{-1}\mathbf{B}_{\mathbf{L},j}$ denotes the indirect causal effect from $X_j$ to $X_i$ through latent variables. Consequently, $(\mathbf{B}_{i,j} + \mathbf{B}_{i,\mathbf{L}}(\mathbf{I} - \mathbf{B}_{\mathbf{L},\mathbf{L}})^{-1}\mathbf{B}_{\mathbf{L},j})X_j$ includes all causal effects in $X_i$ from $\mathrm{Anc}(X_i)$ whose last observed node before $X_i$ in the causal path is $X_j$. This term is exactly the sum of causal effects on $X_i$ by paths in $\mathcal{G}[X_j]$. As a consequence, $Z_i$ equals the sum of causal effects on $X_i$ by the paths in $\mathcal{G}[\emptyset]$. That is, those directed paths whose intermediate nodes are all latent.

Therefore, $\mathbf{A}_{\mathbf{X},\mathbf{X}}^{-1}$ deletes all the outgoing edges from $\mathbf{X}$ and forms several connected components which correspond to the subspace in ISA's definition. Since ISA does not pose any constraints within a subspace, any invertible matrix is valid. Since ISA exists block permutation indeterminacy (Theis, 2006), then we can conclude that there is a permutation $\pi$ of $[k]$ s.t. for any $i \in [k]$, $\exists \mathbf{W}_i \in \mathrm{GL}(|C_i|)$ makes $\mathbf{Z}_{\pi(i)} = \mathbf{W}_i \mathbf{X}'_{C_i}$. □

### A.3.17 PROOF OF REMARK 2

**Remark** 2. *The two causal graphs in Fig. 1a and Fig. 1b cannot be identified by ISA.*

*Proof.* For the causal graphs in Fig. 1a and 1b, after removing all the outgoing edges of $\mathbf{X}$, $X_1, X_2, X_3$ are still connected due to the existence of $L$. According to Theorem 3, $\forall \mathbf{W} \in \mathrm{GL}(3)$ is an ISA solution in both causal graphs. Consequently, the two causal graphs in Fig. 1a and Fig. 1b cannot be identified by ISA. □

## B ILLUSTRATIONS OF ALGORITHMS

### B.1 MERGING RULES

**Proposition 1** (Merging Rules). *Let $\mathbf{A}$ be the active variable set and $C_1$ and $C_2$ be two causal clusters. $C_1$ and $C_2$ share the common latent parent, if one of the following rules holds.*

*R1. 1) $C_1$ and $C_2$ are both pure clusters, and 2) for any subset $\tilde{C} \subseteq C_1 \cup C_2$ with $|\tilde{C}| = 2$, $(\mathbf{A} \setminus \tilde{C}, \tilde{C})$ follows the GIN condition.*

*R2. 1) One of the clusters is a pure cluster and the other is not, e.g., $C_1$ is pure and $C_2$ is impure, and 2) $\forall V_i \in C_1$ and $\forall V_j \in C_2$, $(\mathbf{A} \setminus \{C_2, V_i\}, \{V_i, V_j\})$ follows the GIN condition.*

*R3. 1) $C_1$ and $C_2$ both are impure clusters, and 2) for $\forall \tilde{C} \subseteq C_1 \cup C_2$ with $|\tilde{C}| = 2$, $(\mathbf{A} \setminus \{C_1 \cup C_2\}, \tilde{C})$ follows the GIN condition.*

*Otherwise, $C_1$ and $C_2$ do not share the common latent parent.*

Proposition 1 establishes a set of GIN-based criteria for determining whether two identified causal clusters ($C_1$ and $C_2$) originate from the same latent parent node during each iteration. This proposition defines three mutually exclusive merging rules (R1, R2, R3) based on the purity attribute of clusters (pure or impure). This proposition is crucial for the algorithm's bottom-up reconstruction of the latent structure. During causal discovery, the algorithm first identifies local causal clusters, but these clusters may merely represent different observed subsets of the same latent variable. Through Proposition refmerge, the algorithm can reaggregate these dispersed observational evidence clusters, thereby correctly inferring the number and coverage of latent variables and avoiding the erroneous splitting of a single latent variable into multiple ones.

**Corollary 2.** *Let $L_1$ be a latent variable that was introduced in previous iterations, $C_2$ be a new cluster, and $\mathbf{A}$ be the active variable set in the current iteration. Suppose cluster $C_1$ was a subset of $\mathrm{Ch}(L_1)$ found in previous iterations. Then $C_1$ and $C_2$ share the common latent parent $L_1$ if setting $\mathbf{A} = \mathbf{A} \cup C_1 \setminus L_1$ be the active set, one of the three rules in Proposition 2 holds. Otherwise, $C_1$ and $C_2$ do not share the common latent parent.*

Corollary 2 is a natural extension of Proposition 1 during the recursive process, designed to address the connection between newly discovered causal clusters and known latent variables. Corollary 2 prevents the algorithm from "over-generating" latent variables. Without this check, the algorithm might create a new latent variable node for every occurrence of the same latent factor.

## B.2 Pseudo Code

---

**Algorithm 2** CICA-LiNGAM

---

**Require:** Observed variables $\mathbf{X}$.
**Ensure:** Fully identified causal structure $G$.
 1: Initialize active variable set $\mathbf{A} = \mathbf{X}$ and $G = \emptyset$.
 2: **while** $\mathbf{A} \neq \emptyset$ **do**
 3:   $\mathbf{C} \leftarrow$ FindCausalClusters($\mathbf{A}$); (see Algorithm 3)
 4:   $G \leftarrow$ SparseCICA($\mathbf{C}, G$); (see Algorithm 4)
 5:   $G \leftarrow$ DetermineLatentVariables($\mathbf{C}, \mathbf{A}, G$). (see Algorithm 5)
 6:   $\mathbf{A} \leftarrow$ UpdateActiveData($\mathbf{A}, G$). (see Algorithm 6)
 7: **end while**
 8: Return $G$.

---

---

**Algorithm 3** Finding Causal Clusters

---

**Require:** Active variable set $\mathbf{A}$.
**Ensure:** The set of causal clusters $\mathbf{C}$.
 1: Initialize $\mathbf{C} = \emptyset$ and the group size GrLen = 2;
 2: **while** $|\mathbf{A}| \geq$ GrLen $+ 1$ **do**
 3:   **repeat**
 4:     Select a subset $\mathbf{Y}$ from $\mathbf{A}$ such that $|\mathbf{Y}| =$ GrLen;
 5:     **if** $(\mathbf{A} \setminus \mathbf{Y}, \tilde{\mathbf{Y}})$ follows GIN condition for $\forall \tilde{\mathbf{Y}} \in \mathbf{Y}$ such that $|\tilde{\mathbf{Y}}| = 2$ **then**
 6:       Add $\mathbf{Y}$ into $\mathbf{C}$;
 7:     **end if**
 8:   **until** All subsets with group length GrLen in $\mathbf{A}$ have been selected;
 9:   $\mathbf{A} = \mathbf{A} \setminus \mathbf{C}$; GrLen = GrLen $+ 1$;
10: **end while**
11: Return $\mathbf{C}$;

---

Algorithm 3 constitutes the first step of each iteration in the CICA-LiNGAM algorithm. It employs an incremental subset search strategy, executing four sequential steps in each iteration: i) Enumerating candidate subsets of a specified size from the current active variable set (Line 4); ii) Verifying whether a specific independence structure exists between the subset and its complement using the GIN condition to determine if it constitutes a causal cluster (Line 5); iii) Confirming qualified subsets as causal clusters and removing them from the active set (Line 6); iv) Gradually increasing the subset size to continue searching for larger clusters among the remaining variables (Line 9).

---

**Algorithm 4** Sparse CICA

---

**Require:** The set of causal clusters $\mathbf{C}$, and partial graph $G$.
**Ensure:** Updated partial graph $G$.
 1: **for** each $C_i \in \mathbf{C}$ **do**
 2:   $\mathbf{W} \leftarrow$ sparsest CICA solution on $C_i$;
 3:   $\mathbf{P} \leftarrow$ the permutation matrix that makes $\mathrm{diag}(\mathbf{PW})$ non-zero simultaneously;
 4:   $\tilde{\mathbf{W}} \leftarrow$ divide each row of $\mathbf{PW}$ by its corresponding diagonal element;
 5:   Compute an estimate $\hat{\mathbf{B}}$ using $\hat{\mathbf{B}} = \mathbf{I} - \tilde{\mathbf{W}}$;
 6:   Update $G := G \cup \{j \rightarrow i | \hat{\mathbf{B}}_{i,j} \neq 0\}$;
 7: **end for**
 8: Return $G$;

---

Algorithm 4 constitutes the second and most crucial step in each iteration of the CICA-LiNGAM algorithm. It performs structural estimation within each identified causal cluster, executing four sequential steps per iteration: i) Solving for the sparsest CICA solution on the current causal cluster (Line 2); ii) Finding a permutation matrix to eliminate permutation ambiguity and ensure nonzero diagonal elements (Line 3); iii) Eliminate scaling uncertainty through row normalization to compute

the corresponding adjacency matrix (Lines 4–5); iv) Update the local causal graph structure based on nonzero connection coefficients (Line 6).

---

**Algorithm 5** Determine Latent Variables

---

**Require:** A cluster set $\mathbf{C}$, active variable set $\mathbf{A}$, and partial graph $G$.
**Ensure:** Updated partial graph $G$.
1: $\mathbf{C} \leftarrow$ Merge clusters from $\mathbf{C}$ according to Rules $R1$ and $R2$ of Proposition 1;
2: **for** each $C_i \in \mathbf{C}$ **do**
3:    **if** $L_j$ and $C_i$ satisfy $R3$ of Corollary 2 **then**
4:       $G \leftarrow G \cup \{L_j \rightarrow V_i \mid V_i \in C_i\}$;
5:    **else**
6:       Introduce a new latent variable $L_k$ to $\mathbf{L}$;
7:       $G \leftarrow G \cup \{L_j \rightarrow V_i \mid V_i \in C_i\}$;
8:    **end if**
9: **end for**
10: Return $G$;

---

Algorithm 5 constitutes the third step of each iteration in the CICA-LiNGAM algorithm. It aims to resolve the connection between latent variables and observed clusters, executing three sequential steps: i) Merge causal clusters potentially sharing the same latent parent node based on Proposition 1 (Line 1); ii) Determine whether the current cluster shares a parent node with latent variables discovered in previous iterations (Line 3); iii) Based on the determination, either establish connections between existing latent variables and clusters, or introduce new latent variables and construct causal edges pointing to variables within the cluster (Lines 4–7).

---

**Algorithm 6** Update Active Data

---

**Require:** Current active variable set $\mathbf{A}$, partial graph $G$.
**Ensure:** Updated active variable set $\mathbf{A}$.
1: **if** no new latent variable introduced in $G$ **then**
2:    $\mathbf{A} \leftarrow \varnothing$;
3: **else**
4:    **for** each new latent variable $L_i \in G$ **do**
5:       Initialize the value of $L_i$ according to Lemma 10;
6:       Add $L_i$ into $\mathbf{A}$ and delete $\mathrm{Ch}(L_i)$ from $\mathbf{A}$;
7:    **end for**
8: **end if**

---

Algorithm 6 maintains the set of active variables for the next recursive round. It sequentially executes three steps: i) Check whether new latent variables were introduced in the current iteration; if not, terminate the algorithm (Lines 1–2); ii) If new latent variables exist, initialize their values using the observed child node with the highest causal order as a proxy according to Lemma 10 (Line 5); iii) Add the new latent variable to the active set while removing its child already explained by latent variables, thereby achieving bottom-up recovery of the causal structure (line 6).

B.3   Discussion of Optimization Criterion of CICA

**Definition 15** (Cumulant (Brillinger, 2001)). *Let $X = (X_1, X_2, \ldots, X_n)$ be a random vector of length $n$. The $k$-th order cumulant tensor of $X$ is defined as a $n \times \cdots \times n$ ($k$ times) table, $\mathcal{C}^{(k)}$, whose entry at position $(i_1, \cdots, i_k)$ is*

$$\mathcal{C}^{(k)}_{i_1, \cdots, i_k} = \mathrm{cum}(X_{i_1}, \ldots, X_{i_k}) = \sum_{(D_1, \ldots, D_h)} (-1)^{h-1}(h-1)! \mathbb{E}\left[\prod_{j \in D_i} X_j\right] \cdots \mathbb{E}\left[\prod_{j \in D_h} X_j\right],$$

*where the sum is taken over all partitions $(D_1, \ldots, D_h)$ of the set $\{i_1, \ldots, i_k\}$.*

A $p$-dimensional shared subspace leaves a low-rank fingerprint not only in covariance but also in higher-order cumulants. In the covariance view, identifiability comes from the fact that cross-covariance blocks live in a space of rank at most $p$; equivalently, all $(p + 1)$-minors vanish. The same logic transfers to cumulants: when we form cumulant matrices by linearly contracting the fourth-order cumulant tensor, the contribution of the shared factors still spans at most $p$ independent directions. Hence, these cumulant blocks also satisfy a rank deficiency property.

This viewpoint treats cumulants as providing additional low-rank views of the same latent structure. Because there are many ways to contract a cumulant tensor, we obtain many rank constraints without needing two large disjoint coordinate subsets, which loosens the requirement on $m$. At the same time, the framework strictly contains the second-order case: if we "degrade" the cumulant to order two, we recover the original covariance criterion. In short, moving from covariance to cumulants preserves the rank-deficiency principle while supplying more constraints and thereby stronger identifiability with fewer observed variables.

## C  ADDITIONAL INFORMATION ON EXPERIMENTS

### C.1  COMPUTING INFRASTRUCTURE

The computing devices and platforms are listed as follows.

- OS: Microsoft Windows 11.
- CPU: AMD Ryzen 7 4800H with Radeon Graphics, 2900 Mhz.
- Memory: 16G.
- Python 3.8.18.

### C.2  IMPLEMENTATION DETAILS

We use the solution of ICA as an initialization for CICA optimization. LiNGIC (Li et al., 2026) serves as the dependence measure for testing Triad constraints. Besides, the L1 norm of the demixing matrix is adopted as a convex relaxation of the L0 norm to encourage a sparsest solution.

### C.3  REAL-WORLD EXPERIMENTS

#### C.3.1  TEACHER'S BURNOUT STUDY

Barbara Byrne conducted a study to investigate the impact of organizational (role ambiguity, role conflict, work overload, classroom climate, decision making, superior support, peer support) and personality (self-esteem, external locus of control) on three facets (emotional exhaustion, depersonalization, and personal accomplishment) of burnout in full-time elementary teachers (Byrne, 2016). The data set consists of 32 observed variables with 599 samples. The details of latent factors and their indicators are shown in Table 8 (See Chapter 6, Page 191 in (Byrne, 2016) for more details). As in practice, the ground-truth latent structure is usually hard to know, here we use the hypothesized model given in (Byrne, 2016) as a reference.

**Locating latent variables.**  We run our algorithm with the prior knowledge that the underlying graph contains only the one-factor cluster. The final output of the measurement model is shown above. Here we rename the name of the latent variables in RLCD's output for easier comparison. Compare to the reference model given in (Byrne, 2016), our method merges DM and SS into one latent factor and keeps other clusters correctly identified. Notice that (Dong et al., 2023) arises more errors in clustering step $(L_1, L_2, L_4)$. A possible reason is that $L_1$ only have two measurement variables and are incapable of correctly locating by their method. These results further verify the efficacy of our algorithm. Besides, the structural model learning results (causal graph on latent variables) of our method and RLCD are:

**Inferring latent variable structure.**  The F1 score of our results is 0.522. In contrast, RLCD obtains 0.364. In the output results of the RLCD, most of the edges connected to RC are incorrect.

| Latent Factors | Children (Indicators) |
|---|---|
| Role Ambiguity (RA) | $RA_1, RA_2$ |
| Emotional Exhaustion (EE) | $EE_1, EE_2, EE_3$ |
| Depersonalization (DP) | $DP_1, DP_2$ |
| Role Conflict (RC) | $RC_1, RC_2, WO_1, WO_2$ |
| Self-Esteem (SE) | $SE_1, SE_2, SE_3$ |
| Personal Accomplishment (PA) | $PA_1, PA_2, PA_3$ |
| Peer Support (PS) | $PS_1, PS_2$ |
| Classroom (CC) | $CC_1, CC_2, CC_3, CC_4$ |
| Decision Making (DM) | $DM_1, DM_2$ |
| Superior Support (SS) | $SS_1, SS_2$ |
| External Locus of Control (ELC) | $ELC_1, ELC_2, ELC_3, ELC_4, ELC_5$ |

Table 8: The latent factors and their indicators in teacher's burnout study.

| Ours | | RLCD | |
|---|---|---|---|
| $L_1 \sim \{RA_1, RA_2\}$ | $\checkmark$ | $L_1 \sim \{RA_1, RA_2, RC_1, EE_1\}$ | $\times$ |
| $L_2 \sim \{EE_1, EE_2, EE_3\}$ | $\checkmark$ | $L_2 \sim \{EE_2, EE_3\}$ | $\times$ |
| $L_3 \sim \{DP_1, DP_2\}$ | $\checkmark$ | $L_3 \sim \{DP_1, DP_2\}$ | $\checkmark$ |
| $L_4 \sim \{RC_1, RC_2, WO_1, WO_2\}$ | $\checkmark$ | $L_4 \sim \{RC_2, WO_1, WO_2\}$ | $\times$ |
| $L_5 \sim \{SE_1, SE_2, SE_3\}$ | $\checkmark$ | $L_5 \sim \{SE_1, SE_2, SE_3\}$ | $\checkmark$ |
| $L_6 \sim \{PA_1, PA_2, PA_3\}$ | $\checkmark$ | $L_6 \sim \{PA_1, PA_2, PA_3\}$ | $\checkmark$ |
| $L_7 \sim \{CC_1, CC_2, CC_3, CC_4\}$ | $\checkmark$ | $L_7 \sim \{CC_1, CC_2, CC_3, CC_4\}$ | $\checkmark$ |
| $L_8 \sim \{DM_1, DM_2, SS_1, SS_2\}$ | $\times$ | $L_8 \sim \{DM_1, DM_2, SS_1, SS_2\}$ | $\times$ |
| $L_9 \sim \{ELC_1, ELC_2, ELC_3, ELC_4, ELC_5\}$ | $\checkmark$ | $L_9 \sim \{ELC_1, ELC_2, ELC_3, ELC_4, ELC_5\}$ | $\checkmark$ |

Table 9: The measurement model results of our method and RLCD (Dong et al., 2023).

The possible reason is that some latent factors can not be discovered correctly, which further causes some unobserved confounding between latent variables. Note that previous method can not identify $SE \rightarrow ELC$ in principle, as they form an impure structure on latent variables. By solving CICA on SE and ELC using their observed descendants, our method can recover the causal direction $SE \rightarrow ELC$, which supports the necessitate of introducing two-sided projection.

### C.3.2 BIG FIVE PERSONALITY

**Dataset Description.** The Big Five personality dataset is rooted in the Five-Factor Model (FFM), a seminal theoretical framework in personality psychology to characterize individual personality differences, proposed by American psychologists Paul Costa and Robert McCrae (Costa & McCrae, 1992). This dataset encompasses five core personality dimensions, namely **O**penness, **C**onscientiousness, **E**xtraversion, **A**greeableness, and **N**euroticism, abbreviated as the O-C-E-A-N model. Each dimension is operationally measured by 10 psychometric items, which are designed to capture the nuanced traits underlying each factor. For example, the Openness dimension includes items like "I am intrigued by abstract ideas", while the Conscientiousness dimension features items such as "I am diligent in fulfilling responsibilities".

The data were collected via the online interactive personality testing platform hosted on `https://openpsychometrics.org`, a widely recognized and ethically compliant public data acquisition channel in psychological research. The survey implementation adhered to established ethical norms in empirical psychology, including informed consent and anonymous participation. After data cleaning and validation, the final dataset utilized in this study comprises approximately 20,000 valid samples, covering 50 psychological measurement indicators (10 items per dimension across the five factors). Prior to subsequent analyses, we performed standardization on the data to ensure each variable follows a distribution with a mean of 0 and a variance of 1.

**Measurement Model Learning.** To determine the causal structure in the Big Five personality data, we first employed the GIN algorithm (Xie et al., 2022) to construct a measurement model. The core objective was to identify observed items that highly correspond to each personality dimension. During the clustering process, some items may reflect multiple personality dimensions: for instance,

| Ours | | RLCD | |
|---|---|---|---|
| $RA \rightarrow PA$ | $\checkmark$ | $RA \rightarrow DM/SS$ | $\times$ |
| $EE \rightarrow SE$ | $\checkmark$ | $SE \rightarrow DP$ | $\checkmark$ |
| $SE \rightarrow ELC$ | $\checkmark$ | $SE \rightarrow PA$ | $\checkmark$ |
| $DM/SS \rightarrow SE$ | $\checkmark$ | $DP \rightarrow PA$ | $\checkmark$ |
| $RC \rightarrow DP$ | $\times$ | $DP \rightarrow CC$ | $\times$ |
| $CC \rightarrow EE$ | $\checkmark$ | $RC \rightarrow DP$ | $\times$ |
| $ELC \rightarrow PA$ | $\times$ | $RC \rightarrow SE$ | $\times$ |
| $ELC \rightarrow DP$ | $\times$ | $RC \rightarrow ELC$ | $\checkmark$ |
| $RC \rightarrow EE$ | $\checkmark$ | $RC \rightarrow RA$ | $\times$ |
| $EE \rightarrow ELC$ | $\times$ | | |

Table 10: The structural model results of our method and RLCD (Dong et al., 2023).

item $O_9$ ("I spend time reflecting on things") has dual connotations. On one hand, it reflects in-depth thinking about abstract and complex issues, which is consistent with the cognitive exploration traits of Openness; on the other hand, it involves reviewing and being prudent about one's own behaviors and tasks, aligning with the rigorous and self-disciplined traits of Conscientiousness. For item $A_{10}$ ("I make people feel at ease."), on one hand, the sense of interpersonal security brought by empathy and friendliness is in line with the cognitive exploration traits of Agreeableness; from the perspective of Extraversion, the enthusiasm and talkativeness of extroverts can easily alleviate awkwardness. Such variables cannot correspond to a specific cluster and are therefore not included in the output of the measurement model. After screening via the GIN algorithm, the final output of the measurement model is as follows:

- **Openness**: $L_1\{O_2, O_4, O_7\}$, $L_2\{O_3, O_5, O_6, O_{10}\}$, $L_3\{O_1, O_8\}$;
- **Conscientiousness**: $L_4\{C_1, C_2, C_3, C_4, C_5, C_6, C_7, C_8, C_9, C_{10}\}$;
- **Extraversion**: $L_5\{E_1, E_2, E_4, E_5, E_6, E_7, E_8, E_9, E_{10}\}$;
- **Agreeableness**: $L_6\{A_1, A_2, A_3, A_4, A_5, A_6, A_7, A_8, A_9\}$;
- **Neuroticism**: $L_7\{N_1, N_2, N_3, N_4, N_5, N_6, N_7, N_8, N_9, N_{10}\}$.

The measurement model reveals that the latent variables $L_4$, $L_5$, $L_6$, and $L_7$ serve as unitary representations for Conscientiousness, Extraversion, Agreeableness, and Neuroticism, respectively, explaining the shared variance in their corresponding item responses. In contrast, the Openness dimension exhibits a more granular internal structure, decomposing into three distinct sub-clusters: $L_1$, $L_2$, and $L_3$. These sub-clusters correspond to the core components of "Cognitive exploration", namely abstract reasoning, creative imagination, and linguistic-cognitive complexity.

**Causal Analysis Within Clusters.** After obtaining the measurement model, we further applied our algorithm to uncover causal relationships within the clusters. We found several new conclusions that were not revealed by (Dong et al., 2023).

**(i) Openness**: In the Openness dimension, "difficulty in understanding" is the direct cause of "lack of interest" ($O_2 \rightarrow O_4$). When a person repeatedly fails to understand abstract content, it will directly weaken their willingness to explore this field, whereas if they can understand it easily, they will be more likely to develop interest. Imagination is the core source of creative output: on one hand, "vivid imagination" will directly give rise to "excellent and unique ideas" ($O_3 \rightarrow O_5$), and conversely, a lack of imagination will directly restrict the quality of ideas; on the other hand, the breadth of imagination also directly determines the quantity of ideas, and "vivid imagination" will be transformed into "a constant stream of ideas" ($O_3 \rightarrow O_{10}$). In addition, vocabulary reserve is the foundation of the complexity of language expression: "a rich vocabulary" will directly endow people with the ability to use complex and rare words ($O_1 \rightarrow O_8$), while a poor vocabulary cannot support the use of difficult words.

**(ii) Conscientiousness**: In the Conscientiousness dimension, The intrinsic core trait of "liking order" directly drives individuals to maintain the orderly state of life and work through the behavior of "following a schedule" ($C_7 \rightarrow C_9$); while the behavioral tendency of "paying attention to details" directly translates into the specific manifestation of "being exacting in work" ($C_3 \rightarrow C_{10}$) — a high

sensitivity to details directly acts on the control of omissions in work, thereby presenting a rigorous work state.

**(iii) Extraversion**: In the Extraversion dimension, on one hand, the intrinsic mindset of "feeling comfortable around people" serves as the core prerequisite for active social interaction — if an individual feels at ease in crowds, this mindset will directly prompt them to initiate conversations actively ($E_3 \to E_5$), and at the same time, it will directly drive them to interact with multiple people in social scenarios such as parties ($E_3 \to E_7$); on the other hand, the core tendency of "not liking to draw attention to oneself" is the direct trigger for social avoidance behaviors — the aversion to others' attention will directly guide the individual to choose a low - key position "keeping in the background" ($E_8 \to E_4$), and this sense of aversion will also directly suppress their desire to express themselves in front of strangers ($E_8 \to E_{10}$).

**(iv) Agreeableness**: In the Agreeableness dimension, in which $A_4$ ("I sympathize with others' feelings.") plays a key mediating role: "feeling others' emotions" is the prerequisite for generating "sympathizing with others' feelings" ($A_9 \to A_4$) — only by accurately capturing others' emotional states can one further put oneself in others' shoes and generate emotional resonance, while the inability to perceive emotions will directly lead to a lack of empathy. On this basis, "sympathizing with others' feelings", as a mediating variable, becomes the direct driving force for altruistic behavior — a deep resonance with others' feelings will directly prompt individuals to take time out for others ($A_4 \to A_8$); conversely, if such empathy ($A_4$) is lacking, even if one can perceive others' emotions, it will directly reduce the willingness to engage in the altruistic behavior of active companionship.

**(v) Neuroticism**: In the Neuroticism dimension, on one hand, the core trait of "changing mood a lot" is directly externalized as the specific manifestation of "having frequent mood swings" ($N_7 \to N_8$); on the other hand, the emotional tendency of "getting stressed out easily" exerts a direct impact through the accumulation of sustained states ($N_1 \to N_{10}$) — being in a stressed state for a long time will directly lead to the continuous superposition of negative emotions, which in turn gives rise to the emotional outcome of "often feeling blue".

**Structural Model Learning.**    Following the learning of the measurement model and cluster causal analysis, we further recovered the causal structure among latent variables. While some of our findings are generally consistent with (Dong et al., 2023), we present here only the newly discovered structural learning results.

**Causal relationship**: $\{L_1 \to L_2, L_1 \to L_3, L_4 \to L_6, L_6 \to L_5, L_4 \to L_1, L_1 \to L_5, L_7 \to L_5\}$.

**(i) ($L_1 \to L_2, L_1 \to L_3$)**: In the Openness dimension, "Abstract cognitive ability and interest orientation($L_1$)" serve as the prerequisite for fostering "creative potential ($L_2$)" and "complexity of language expression ($L_3$)". Only by overcoming difficulties in understanding abstract concepts and maintaining interest in them can one provide cognitive support for the operation of imagination and the accumulation of vocabulary. On this basis, $L_1$ directly drives the manifestation of $L_2$ and $L_3$: strong abstract cognitive ability translates into rich imagination and excellent creative output, while a positive orientation toward abstract thinking enhances the depth of vocabulary reserves and the ability to use complex words; conversely, deficiencies in $L_1$ regarding abstract cognition will directly restrict the development of creativity and the complexity of language expression.

**(ii) ($L_7 \to L_5$)**: "Emotional Instability ($L_7$)" exerts a negative regulatory effect on "social participation tendency ($L_5$)". Emotional fluctuations and feelings of anxiety directly suppress people's desire to interact, thereby leading to social avoidance behaviors such as staying in the background and being quiet around strangers.

# D    RELATED WORK

Existing methods for handling causal discovery in the presence of latent confounders can be categorized into the following folds. Here we list the papers focusing on linear continuous variables,

**Conditional independence constraints-based.**    This line of work (Spirtes et al., 2000; Colombo et al., 2012; Akbari et al., 2021; Triantafillou & Tsamardinos, 2015) uses conditional independence tests to infer causal graphs. The core idea is to find patterns of conditional independence among variables to reveal the underlying causal structure. By testing for independence among observed

variables, these methods can discover the causal skeleton and orient some of the edges. These approaches can handle both linear and nonlinear causal relationships.

**Rank deficiency-based.** This line of work (Silva et al., 2002; 2006; Kummerfeld & Ramsey, 2016; Huang et al., 2022; Li et al., 2024) uses rank constraints of covariance matrices to locate latent variables and infer the causal skeleton. The core idea is that in linear causal models, the covariance matrix or its submatrices exhibit specific rank properties. By analyzing these rank deficiencies, it is possible to reveal the connection patterns between latent and observed variables.

**Matrix decomposition-based.** This line of work proposes to identify the causal structure of latent variables by decomposing the covariance or precision matrix into matrices with specific structures, such as low-rank and sparse. Specifically, the low-rank matrix captures the causal relationships from latent variables to observed variables, while the sparse matrix represents the direct causal relationships among observed variables. Representative works include (Chandrasekaran et al., 2011; 2012; Anandkumar et al., 2013; Frot et al., 2019).

**Overcomplete independent component analysis (OICA)-based.** This line of work leverages Overcomplete Independent Component Analysis (OICA) to handle problems with latent variables. OICA allows more source signals than observed signals, and thus can be used to learn the causal structure with latent variables. Related work in this area includes (Shimizu et al., 2009; Entner & Hoyer, 2010; Adams et al., 2021; Dai et al., 2026).

**Generalized independent noise (GIN)-based.** This line of work (Cai et al., 2019; Xie et al., 2020; Dai et al., 2022; Xie et al., 2023; Chen et al., 2022; 2023; Jin et al., 2023; Li et al., 2024; Xie et al., 2024) extends the independent noise condition to handle scenarios with latent variables. The core idea is that, for non-Gaussian linear causal mechanisms, higher-order statistics can be leveraged to bring asymmetry and identify causal directions.

**Higher-order cumulant-based.** This line of work (Cai et al., 2023; Chen et al., 2024a; Schkoda et al., 2024; Chen et al., 2025a) leverages higher-order cumulants to identify the causal structure when latent variables are present. For non-Gaussian distributions, cumulants can capture richer structural information than covariance alone. These studies show that the cumulant tensors of observed variables have specific rank constraints that can reveal the causal skeleton of latent variables.

**Score-based.** This line of work (Agrawal et al., 2023; Ng et al., 2024) frames the learning of causal structure as a search problem, aiming to find the graph structure that best fits the data. They typically define a scoring function to measure a graph's goodness of fit, then use search algorithms (like hill-climbing or greedy search) to find the highest-scoring graph.

## E    THE USE OF LARGE LANGUAGE MODELS (LLMs)

We used ChatGPT to refine writing only. The prompt was: "I am preparing a paper for submission to an international conference and would like your help to check for any grammatical issues and refine the wording or sentence structure where necessary to ensure conciseness and precision." Edits were applied paragraph-by-paragraph, and all outputs were verified and revised by the authors; no scientific content, analyses, or references were generated by the tool.

