# OpenReview forum: "Conditional Independent Component Analysis for Estimating Causal Structure with Latent Variables"
_ICLR.cc/2026/Conference — ICLR 2026 Poster_

### Official Review · Reviewer_8zWU · 2025-10-27

**Soundness:** 3
**Presentation:** 2
**Contribution:** 3
**Rating:** 4
**Confidence:** 3

**Summary:**

This paper proposes a new algorithm for identifying latent variables and causal structures in linear causal models, especially for models with unknown latent variables. A new CICA principle is proposed as a key tool for identification, which utilizes two-sided projections instead of one-sided projections, thus allowing for wider identifiable cases. Experiments on synthetic data show that the proposed algorithm performs better in both variable and structure identification, compared with several baseline methods.

**Strengths:**

1.	A novel identifiability principle is proposed (CICA), allowing for identification of some cases that is originally unidentifiable. I think this is the major merit of this paper.
2.	A complete identification algorithm is presented, achieving better results on both latent variable identification and structure discovery than several baselines.
3.	Examples are provided for better illustration of certain Lemmas or Conditions.

**Weaknesses:**

1.	The major weakness lies in unsatisfying experimental results. In Tab. 1, causal discovery metrics (COR, F1-Score) are quite low for the proposed methods. If I understand it correctly, COR < 0.7 or F1-Score < 0.8 indicate a structure identification result that is far from optimal, > 0.9 results are expected for a theoretically-supported algorithm. Meanwhile, error in latent variables is supposed to decrease while sample size grows. However, the proposed method does not exhibit such monotonicity, making it questionable for the stability of proposed algorithm.
2.	Another weakness is about the clarity. In the main algorithm (Alg. 1), only a rough process is given, and no implementation details is provided in this manuscript. Some subprocesses is based on Lemma or Theorem which does not contain a constructive solution itself. I suggest the authors provide details for each subprocess in Appendix.

**Questions:**

1.	Since the proposed CICA principle is one of the major contributions, why not compare your method with GIN and TIN algorithms in experiments?
2.	In Fig. 2, the CICA is given in a form conditioned on L. However, in the rest of this paper, it is more common that the independence is presented without conditioning on L. Which is right?
3.	In Eqn. 1, the variable V=AE is introduced as part of problem setup. However, I find V is seldomly mentioned in the rest parts. Is V necessary here? Or it should be mentioned separately somewhere else.
4.	The solution in line 234 seems to be incorrect, see if it should be $\omega_1=[-(au+b),a], \omega_2=[-v, 1]$.
5.	In Condition 2, what does $LPa(X_i)$ mean? According to your definition in line 106, LPa takes a subset S as input and returns common latent parents of any *two* nodes in S. However, $X_i$ is a single node here.

---

> ### Author Response · Authors · 2025-11-21
> **Response to Reviewer 8zWU - Part 1/2 (W1 & W2 & Q1)**
>
> We sincerely appreciate the reviewer's helpful feedback and careful reading! Please see below for our response.
>
> ---
>
> **(W1)** The reviewer argues the performance is not satisfying enough (expected F1-Score > 0.9). The reviewer queries why the error in latent variables does not monotonically decrease while the sample size grows.
>
> **R:** We thank the reviewer for the careful reading. We would like to first emphasize that latent variable causal discovery is still under active development; as can be shown from the results, we still consistently perform better. Besides, we also highlight several specific factors that may explain the observed performance.
>
> In latent variable modeling, metrics like F1 scores are sensitive to the error of latent variables. A single error in clustering observed variables to latents cascades into massive errors in the structural edges. In our estimation algorithm, as we presented in Algorithm 3 (pages 26 in the Appendix), we identify latent variables and cluster observed variables based on GIN condition. Consequently, the performance of our method inherently depends on the accuracy of the GIN test. To our knowledge, how to more accurately test whether the GIN condition holds remains an open problem with room for improvement. Since we do not see this part as our main contribution in this work, we follow [1] and adopt the definition-based GIN testing procedure. In other words, more robust implementations of GIN in the future could be directly incorporated into our framework and potentially yield even better results.
>
> In our Case 4, which is a highly impure scenario, the accuracy of GIN tests may collapse, leading to instability in the “Error in Latent Variables” metric. To mitigate the instability of GIN testing, we now follow [2] with the prior knowledge that the underlying graph contains only the 1-latent set to remove unnecessary GIN tests. This prior knowledge is also applied to other comparisons. By using this adjustment, we resolved inconsistencies in certain metrics and achieved better results in Case 4. We have updated our new results in [updated submission].
>
> ---
>
> **(W2)** The reviewer suggests providing details for each subprocess of the main algorithm to improve clarity.
>
> **R:** We thank the reviewer for this constructive suggestion. We have expanded Appendix B in our revised manuscript. This appendix now contains step-by-step pseudocode for the key subprocesses of our CICA-LINGAM algorithm (Algorithm 1). We hope these additions provide sufficient details for each subprocess and fully address the reviewer's concern, making the paper's algorithmic contribution much clearer.
>
> ---
>
> **(Q1)** The reviewer is curious about why not compare our method with the GIN and TIN-based algorithms in experiments.
>
> **R:** We thank the reviewer for this question, which allows us to clarify the scope of our experimental comparisons.
>
> **Regarding GIN:** We fully agree that GIN is a highly relevant baseline principle. **We did compare against GIN-based methods:** In our experiments (Table 1), we included **PO-LINGAM** and **LaHME** as baselines. These are state-of-the-art methods that explicitly use the GIN condition. Our results in Table 1 demonstrate that in the fully impure scenarios (e.g., Cases 2 and 4), which our CICA-based method is designed to solve, these GIN-based methods fail to produce correct results.
>
> **Regarding TIN:** We did not include TIN as an experimental baseline for the following reason:
>
> TIN is a **metric**. $\operatorname{TIN}(\mathbf{Z, Y})$ quantifies the dimension of the subspace $\lbrace \omega|\omega^{\top} \mathbf{Y} \perp \mathbf{Z} \rbrace$. To the best of our knowledge, the original TIN paper [3] focuses on introducing this metric, its theoretical properties and how to estimate the value from samples, but it **does not provide a practical, end-to-end algorithm** based on this value to recover a full causal graph. Therefore, a direct experimental comparison as a baseline was not feasible.
>
> While a direct experimental comparison with TIN is not feasible, our analysis in Section 3.1 (proof in Appendix A.3.2, pages 17) does explain in detail why both the GIN and TIN conditions are fundamentally insufficient to solve the impure structure problem illustrated in our Figure 1, which is what necessitates the introduction of our CICA principle.
>
> ---
> References:
>
> [1] Feng Xie, Ruichu Cai, Biwei Huang, Clark Glymour, Zhifeng Hao, and Kun Zhang. Generalized Independent Noise Condition for Estimating Latent Variable Causal Graphs. NeurIPS 2020.
>
> [2] Feng Xie, Biwei Huang, Zhengming Chen, Ruichu Cai, Clark Glymour, Zhi Geng, and Kun Zhang. Generalized Independent Noise Condition for Estimating Causal Structure with Latent Variables. Journal of Machine Learning Research, 25(2024): 1-61.
>
> [3] Haoyue Dai, Peter Spirtes, Kun Zhang. Independence Testing-Based Approach to Causal Discovery under Measurement Error and Linear Non-Gaussian Models. NeurIPS 2022.

---

> ### Author Response · Authors · 2025-11-21
> **Response to Reviewer 8zWU - Part 2/2 (Q2 - Q5)**
>
> We sincerely appreciate the reviewer's helpful feedback and careful reading! Please see below for our response.
>
> ---
>
> **(Q2)** The reviewer is curious about the relationship between the CICA form ($Z_i \perp Z_j | \mathbf{L}$, conditioned on $\mathbf{L}$) with the "Two-Sided Projection" form ($\omega_{1}^{\top}\mathbf{Y} \perp \omega_{2}^{\top}\mathbf{Z}$, without conditioning on $\mathbf{L}$).
>
> **R:** We sincerely thank the reviewer for pointing out this important point. Next, we will explain this in three steps: 1) Why do we discuss $\omega_1^{\top}\mathbf{Y} \perp \omega_2^{\top}\mathbf{Z}$? 2) Why we seek for $Z_i \perp Z_j | \mathbf{L}$ instead? 3) What's their connection?
>
> 1. The "Two-Sided Projection" $\omega_1^{\top}\mathbf{Y} \perp \omega_2^{\top}\mathbf{Z}$ is the motivation (Section 3.1).
>
>    Prior methods like GIN and TIN fail on distinguishing two graphs in Figure 1 as they use only "one-sided projections" $\omega^{\top}\mathbf{Y} \perp \mathbf{Z}$. Remark 1 shows that "two-sided projections" $\omega_1^{\top}\mathbf{Y} \perp \omega_2^{\top}\mathbf{Z}$ leave additional identifiable traces that enable recovering the causal structure. Intuitively, each "two-sided projection" $\omega_1^{\top}\mathbf{Y} \perp \omega_2^{\top}\mathbf{Z}$ may bring a discriminative signal to obtain better identifiability. Thus, we hope to collect as many "two-sided projections" as possible.
>
> 2. $Z_i \perp Z_j | \mathbf{L}$ is our proposed principle (Section 3.2).
>
>    Why do we not directly brute-force search all possible "two-sided projections"?
>
>    We have two approaches to achieve this: OICA or direct search for each "two-sided projection" ($\omega_1, \omega_2$). Their problems are: 1) computationally prohibitive and prone to local optima; or 2) difficult to guarantee that all feasible ($\omega_1, \omega_2$) have been found.
>
>    Instead, we propose to optimize CICA ($Z_i \perp Z_j | \mathbf{L}$). CICA has at least three advantages: 1) By using rank deficiency as optimization proxy, CICA is much easier to optimize (similarly to an ICA problem), without the need to estimate the distribution of $L$. 2) The CICA solution $\mathbf{W}$ naturally induce all necessary two-sided projections ($\omega_{1}^{\top}\mathbf{Y} \perp \omega_{2}^{\top}\mathbf{Z}$), which is discussed in Lemma 3. 3) The CICA solution $\mathbf{W}$ contains the complete set of independence relationships required to identify the graph, which we present in our later identifiability theory.
>
> 3. We prove in Lemma 3 that any solution satisfying our CICA principle will naturally induce the "two-sided projections".
>
>    For a $p$-order CICA solution $\mathbf{W}$, $\mathbf{Z=WX}$, $|\mathbf{Z}|=|\mathbf{X}|=m$. So we have $Z_i \perp Z_j | \mathbf{L}\ (\forall i \neq j)$, $|\mathbf{L}|=p$. If we have $s(s > p)$ variables $\mathbf{Z_S}$, there exists a non-empty null space that can project the data orthogonal to the latents, i.e., $\lbrace\omega\in \mathbb{R}^s|\omega^{\top} \mathbf{Z_S}\perp L\rbrace \neq \emptyset$. Once $\mathbf{L}$ is cancelled out by such a projection, the remaining signal consists only of independent noise terms. Thus, we observe the independence $\omega^{\top} \mathbf{Z_S} \perp \mathbf{Z_{\bar{S}}}$ ($\mathbf{\bar{S}}:=[m]\setminus{\mathbf{S}}$) without conditioning on $\mathbf{L}$. Consider $\mathbf{Z_S}=\mathbf{W_{S,:}X}, \mathbf{Z_{\bar{S}}}=\mathbf{W_{\bar{S},:}X}$, we have $\omega^{\top} \mathbf{W_{S,:}X} \perp \mathbf{W_{\bar{S},:}X}$, which is essentially "two-sided projections" $\omega_1^{\top}\mathbf{Y} \perp \omega_2^{\top}\mathbf{Z}$.
>
> Thanks again for this insightful comment! We have revised the transition between Sections 3.1 and 3.2 to clarify it.
>
> ---
>
> **(Q3-Q5)** Other helpful comments, including:
>
> * **Necessity of introducing the notation $V$**:  $V$ represents the complete set of all variables in the causal graph, including both the latent variables ($L$) and the observed variables ($X$). Defining $V$ provides a unified way to refer to any variable in the graph. For instance, when specifying parents as $\operatorname{Pa}(V_i)$, it avoids treating latent and observed variables separately. Without defining $V$, our notation would be much more cumbersome. Finally, we characterize many graph properties over the full causal structure in Appendix A.3.1. Here, it is more convenient to define a generic variable $V$ as many properties are shared for both latent and observed variables.
>
> * **Incorrect equation in Example 1**: Thank you for carefully pointing out this typo. We have fixed it in the updated submission.
> * **Definition of $\operatorname{LPa}(X_i)$**: Thank you for carefully pointing out this mistake. Now Condition 2 becomes: For any $X_i \in \mathbf{X}$, $\exists X_j \in \mathbf{X}\setminus\{X_i\}$ with $\operatorname{LPa}(\{X_i,X_j\})\neq \emptyset$, $X_i \not\rightarrow X_j$. We have fixed it in the updated submission.

---

> ### Comment · Reviewer_8zWU · 2025-11-25
>
> Thank you for your response.
>
> While most of my other concerns have been addressed, I still have questions regarding the experimental results after reading the rebuttal. I checked the results reported in original papers of CDHS and PO-LiNGAM and find their COR scores are mostly > 0.9. Since PO-LiNGAM also relies on GIN test, how do you explain such large gap compared with results in your paper? If the cases in your paper are harder to solve, can you find simpler cases to prove the effectiveness of your method (achieving OCR>0.9)? If GIN test is not reliable, why can CDHS and PO-LiNGAM achieve such high scores in their cases? In one word, I remain skeptical about the effectiveness of your CICA-LiNGAM method with current low COR score.

---

> > ### Author Response · Authors · 2025-11-27
> > **Follow-Up Response to Reviewer 8zWU (Part 1/2)**
> >
> > We sincerely appreciate the reviewer's helpful feedback and careful reading! Please see below for our response.
> >
> > We address the reviewer's concern through the following steps:
> >
> > ---
> >
> >    **(Q6.1)**: CDHS and PO-LiNGAM have competitive results in their original papers, but they achieve unsatisfactory performance in our setting.
> >
> >    **R**: The discrepancy arises because the experimental settings are different. In the original CDHS and PO-LiNGAM's experiments  ([CDHS](https://openreview.net/pdf?id=fGhr39bqZa), [PO-LiNGAM](https://openreview.net/pdf?id=nHkMm0ywWm)), the data-generating process either fully or closely satisfies the purity assumption, under which the induced graph among the observed variables is nearly empty. In contrast, our experiments involve highly impure structures. The observed variables remain densely connected even if the latent variable $L$ is removed. This setting is harder and falls outside the identifiability conditions required by both CDHS and PO-LiNGAM.
> >
> >    Concretely, CDHS locates latent variables based on "Homologous Surrogate" (see Definition 1 in [CDHS](https://openreview.net/pdf?id=fGhr39bqZa)). They require that each latent variable has a corresponding homologous surrogate. However, in our Cases 2, 3, and 4, this assumption does not hold. Consequently, CDHS can not correctly identify latent variables. As we said in the response to **(Q1)**, a single error in clustering observed variables/identifying latent variables cascades into massive errors in the structural edges. This explains why CDHS exhibited degraded performance in our experiments.
> >
> >    As for PO-LiNGAM, their identifiability condition requires each atomic unit (see Definition 3 in [PO-LiNGAM](https://openreview.net/pdf?id=nHkMm0ywWm)) with size $\|V\|= t$ has at least $t+1$ atomic units as pure children (see Definition 2 in [PO-LiNGAM](https://openreview.net/pdf?id=nHkMm0ywWm)). In PO-LiNGAM's implementation, the algorithm relies on finding enough pure children sharing the same latent parents to locate latent variables. However, in our Cases 2, 3, and 4, the latent variables have only one pure child, preventing PO-LiNGAM from forming sufficiently large pure child sets. Consequently, PO-LiNGAM can not correctly locate latent variables, leading to poor estimation performance in our setting.
> >
> >    In summary, both CDHS and PO-LiNGAM's identifiability guarantee needs additional assumptions that are violated in most settings in our experiments. Their estimation algorithms based on these specific assumptions will consequently fail.
> >
> > ---
> >
> >    **(Q6.2)**: Regarding why GIN tests seem reliable in PO-LiNGAM's original paper.
> >
> >    **R:** The key reason is that verifying the GIN condition becomes more challenging in **impure structures** than in the (nearly) pure structures used in the original PO-LiNGAM experiments.
> >
> >    Recall that the [GIN test](https://proceedings.neurips.cc/paper/2020/file/aa475604668730af60a0a87cc92604da-Paper.pdf ) verifies whether a constructed surrogate variable (a specific linear combination $E_{Y||Z} = \omega^T Y$) is statistically independent of a variable set $Z$. The vector $\omega$ is estimated to mathematically cancel out the influence of hypothesized common latent confounders.
> >
> >    Statistically, the estimation of $\omega$ involves inverting covariance matrices or computing null spaces. In impure structures, direct causal edges between observed variables induce strong multicollinearity among observed variables, typically increasing the condition number $\kappa(\Sigma)$ of the covariance matrix. According to standard perturbation bounds, a large condition number amplifies small finite-sample errors into massive deviations in the estimated vector $\omega$. This prevents the surrogate variable from effectively canceling out the latent confounders.
> >
> >    Algorithmically, PO-LiNGAM’s original experiments rely on nearly pure structures, where each latent variable has multiple redundant pure children. Even if one individual GIN test is wrong, the algorithm can still probably identify latent variables through other pure child sets. In contrast, impure structures have less redundant information and place higher accuracy demands on GIN tests. If the key GIN test used to determine the existence of a hidden variable fails, the algorithm will never be able to establish the existence of that hidden variable.
> >
> >    In summary, under impure structures, both the statistical and algorithmic challenges become more severe, explaining the degraded performance observed in our experiments.

---

> > ### Author Response · Authors · 2025-11-27
> > **Follow-Up Response to Reviewer 8zWU (Part 2/2)**
> >
> > **(Q6.3)** The reviewer requests a simpler case to prove the effectiveness of our method and achieve better metrics.
> >
> > **R:** Here we present another synthetic experiment. The underlying causal structure is as follows:
> > $$
> > \begin{align*}
> > L_1 \rightarrow L_2, \qquad L_1\rightarrow\lbrace X_1,X_2,X_3\rbrace, \qquad L_2\rightarrow\lbrace X_4,X_5,X_6\rbrace,
> > \qquad X_1 \rightarrow X_2.
> > \end{align*}
> >  $$
> > The structure nearly satisfies the purity assumption. Other details remain the same as those in Section 5. In the table below, we use ELV, COR, and F1 as abbreviations for Error in Latent Variables, Correct-Ordering Rate, and F1-score, respectively.
> >
> >    | **Graph**  | **Method** | **ELV ($\downarrow$)** |               |               | **COR ($\uparrow$)** |               |               | **F1 ($\uparrow$)** |               |               |
> >    | ---------- | ---------- | -------------------------------------------- | ------------- | ------------- | -------------------------------------- | ------------- | ------------- | ------------------------- | ------------- | ------------- |
> >    |            |            | **5k**                                       | **10k**       | **20k**       | **5k**                                 | **10k**       | **20k**       | **5k**                    | **10k**       | **20k**       |
> >    | **Case 5** | CDHS       | **0.45±0.74**                                | **0.25±0.43** | 0.20±0.40     | 0.69±0.45                              | 0.75±0.43     | 0.80±0.40     | 0.69±0.45                 | 0.75±0.43     | 0.80±0.40     |
> >    |            | PO-LiNGAM  | 0.50±0.50                                    | 0.35±0.48     | 0.10±0.30     | 0.50±0.50                              | 0.65±0.48     | 0.90±0.30     | 0.50±0.50                 | 0.65±0.48     | 0.90±0.30     |
> >    |            | LaHME      | 0.90±0.30                                    | 0.90±0.30     | 1.10±0.30     | 0.05±0.16                              | 0.07±0.21     | 0.00±0.00     | 0.08±0.25                 | 0.09±0.26     | 0.00±0.00     |
> >    |            | **Ours**   | 0.50±0.50                                    | 0.35±0.48     | **0.10±0.30** | **0.72±0.61**                          | **0.79±0.52** | **0.93±0.36** | **0.72±0.61**             | **0.79±0.52** | **0.93±0.36** |
> >
> > Our proposed algorithm achieves optimal performance across most metrics. The performance improvement over PO-LiNGAM primarily comes from accurately recovering the edge $X_1 \rightarrow X_2$, which PO-LiNGAM is not able to recover under its identifiability assumptions. This simpler case highlights the effectiveness of our approach in nearly pure settings.
> >
> > ---
> >
> > Once again, we are grateful for the reviewer's valuable comments, and sincerely hope that you will find your suggestions and concerns well addressed by the response above and the relevant sections in the paper. Your further feedback would be greatly appreciated, and we look forward to the opportunity to respond to it.

---

> > > ### Comment · Reviewer_8zWU · 2025-11-28
> > >
> > > After reading the author response, I decide to raise my score to 6 (currently unable to operate). It is the unsatisfaction on overall performance (e.g., COR) that stops me from further increasing my score. I lean towards acceptance, since a novel principle (CICA) is proposed, while experimental results show some advantage. The unsatisfying performance may due to limitation of existing GIN test method as explained by the authors, I encourage the authors to fix this problem so that the experimental results would be more convincing.

---

### Official Review · Reviewer_Lp3A · 2025-10-28

**Soundness:** 4
**Presentation:** 3
**Contribution:** 4
**Rating:** 6
**Confidence:** 3

**Summary:**

This paper proposes Conditional Independent Component Analysis (CICA), a new statistical causal discovery method that can handle latent variables. Traditional causal discovery methods involving latent variables often require a strong assumption that each latent variable has at least one pure child node to ensure identifiability.
In contrast, the proposed approach proves that identifiability can be achieved without the pure-child assumption, by jointly adjusting the projections of two sets of observed variables to find directions that are conditionally independent given the latent variables, together with a sparsity condition on the causal structure.

Building on this theoretical result, the authors introduce the CICA-LiNGAM algorithm, which implements the framework in practice.
They also show that, unlike Independent Subspace Analysis (ISA), which assumes independence only across subspaces, CICA can explicitly model conditional autonomy concerning latent structures.

Experiments on simulated data demonstrate that the proposed method estimates causal structures more accurately than existing statistical causal discovery approaches that account for latent variables.

**Strengths:**

The main strength of this paper is that it removes the pure-child assumption, which is required by most existing statistical causal discovery methods dealing with latent variables, through a natural and well-motivated reformulation. This represents a significant contribution to the causal discovery research community. The ability to handle latent variables more naturally greatly expands the practical applicability of causal discovery to real-world problems where unobserved factors cannot be eliminated.

The theoretical development is carefully and clearly presented, making the reasoning easy to follow.

In addition, the paper uses clear illustrative examples that help readers understand the key ideas, and the simulation experiments convincingly demonstrate that the proposed method outperforms conventional approaches.

**Weaknesses:**

While the paper makes a strong theoretical contribution by proving that the sparsest solution is unique and identifiable, it does not sufficiently discuss how much denser alternative feasible solutions could differ from the sparsest one. This issue is important for understanding the algorithm’s stability and robustness. Although it is plausible that such alternative solutions would have similar structures due to the algorithmic design, explicitly discussing this point, and, if possible, providing empirical evaluation, would greatly strengthen the paper’s practical credibility.

In addition, the minimal sparsity principle, the assumption that the true causal structure is the sparsest among all feasible ones, is theoretically reasonable in the context of identifiability proofs. However, this assumption appears to be motivated mainly by the need for identifiability rather than by empirical or theoretical evidence that real causal systems are indeed maximally sparse. In this sense, the preference for sparsity may be viewed more as a convenient modeling choice than as a natural property of causal mechanisms. A clearer discussion of this limitation, or of the conditions under which sparsity aligns with real-world causal structures, would further improve the interpretability and credibility of the work.

**Questions:**

Could you discuss more clearly how much denser alternative feasible solutions might differ from the sparsest one, and, if possible, include an evaluation experiment to illustrate this?

---

> ### Author Response · Authors · 2025-11-21
> **Response to Reviewer Lp3A - Part 1/2 (W2)**
>
> We sincerely appreciate the reviewer's encouragement and insightful feedback. Please see below for our response.
>
> ---
>
> **(W2)** The reviewer is curious whether the minimal sparsity principle is a convenient modeling choice or a natural property of causal mechanisms. Further, the reviewer requests a clearer discussion of the conditions under which sparsity aligns with real-world causal structures.
>
> **R:** We sincerely thank the reviewer for this deep question. To begin with, we would like to clarify a key point where the reviewer might have had some misunderstandings: the **"minimal sparsity principle" is not a prior assumption/convenient choice** we impose on the model. Rather, it is a **provable theoretical property** that emerges from the CICA framework itself, which we then exploit for identifiability.
>
> Here is the distinction:
>
> 1. We do **not** assume, as a general rule, that "real-world causal structure is maximally sparse." On the contrary, it can be arbitrarily dense.
> 2. Instead, our paper first identifies a new indeterminacy unique to CICA (Lemma 2), the conditional set $\mathbf{L}$. As we illustrate in **Figure 3**, multiple solutions (e.g., $\mathbf{W}_1$ and $\mathbf{W}_2$) can perfectly satisfy the CICA definition, making them equivalent from a conditional independence perspective.
> 3. The central challenge, then, is how to distinguish the solution ($\mathbf{W_1} \sim \mathbf{I-{B_{X,X}}}$) that corresponds to the ground-truth causal structure from the "spurious" (no direct connections with the ground-truth) ones ($\mathbf{W}_2$).
>
> 4. Our key theoretical contribution, **Lemma 8**, is a proof that the true causal structure ($\mathbf{I}-\mathbf{B_{X,X}}$) is provably the sparsest solution among all valid $p_{min}(\mathbf{X})$-order CICA solutions. The other valid CICA solutions (like $\mathbf{W}_2$) are provably denser ($\|\mathbf{W}_1\|_0 \leq \|\mathbf{W}_2\|_0$). More specifically, $\operatorname{Supp}(\mathbf{W}_1) \subseteq \operatorname{Supp}(\mathbf{W}_2)$. No further conditions are required other than the basic assumptions in our paper.
> 5. **Condition 2** and **Lemma 9** established when $\|\mathbf{W}_1\|_0 < \|\mathbf{W}_2\|_0$ or $\operatorname{Supp}(\mathbf{W}_1) \subsetneq \operatorname{Supp}(\mathbf{W}_2)$.
>
> Therefore, our preference for sparsity is not a "convenient modeling choice" or a prior about nature. It is the provable, discriminative signal that the CICA solution space provides. Our algorithm uses this provable property as a selection criterion to break the ambiguity and uniquely identify the true causal graph.
>
> We want to thank the reviewer again for this point. In the new version, we have revised Section 3.4 and made this point more explicit. Hope this addresses the reviewer's concern.

---

> ### Author Response · Authors · 2025-11-21
> **Response to Reviewer Lp3A - Part 2/2 (Q1 & W1)**
>
> We sincerely appreciate the reviewer's encouragement and insightful feedback. Please see below for our response.
>
> ---
>
> **(Q1 & W1)** The reviewer has concerns that the paper does not sufficiently discuss how much denser alternative feasible solutions could differ from the sparsest one.
>
> **R:** We sincerely thank the reviewer for this excellent question. In what follows, we answer from two perspectives. We first present the theoretical results by formalizing a concise characterization on this gap, and by simulation observations to confirm this is clear enough.
>
> **Theoretical Results:** As a kind reminder, the alternative feasible CICA solutions are not random graphs. They are specific, structured matrices (like $\mathbf{W_2}$ in Fig. 3) that arise provably from making a different, valid choice for the latent conditional set (e.g., swapping the latent $L$ for an exogenous noise $E_k$). Our proof of **Lemma 8 (Appendix A.3.11)** provides a precise characterization of this density gap:
>
> The true solution corresponding to ground-truth causal structure ($\mathbf{W_1} \sim \mathbf{I-B_{X,X}}$) has a sparsest structure where row $j$ has a non-zero support of $\operatorname{Supp}(\mathbf{W_{1;j,:}}) = \lbrace X_j \rbrace \cup \operatorname{Pa}(X_j)$.
>
> An alternative solution ($\mathbf{W_2}$, resulting from swapping $L$ and $E_k$) has a provably denser structure. Its row $j$ has a support of $\operatorname{Supp}(\mathbf{W_{2;j,:}}) = \lbrace X_j, X_k \rbrace \cup \operatorname{Pa}(X_j) \cup \operatorname{Pa}(X_k)$. This means the support of $\mathbf{W_2}$'s $j$-th row is essentially the union of the supports of the $j$-th and $k$-th rows of the true solution $\mathbf{W_1}$. This gap is a fundamental, structural increase in density.
>
> **Simulation Results:** To provide the precise evaluation the reviewer requested, we conduct a comprehensive simulation to quantify this gap. We analyze all possible causal graphs with 1 latent variable ($L$) and $n$ observed variables ($X_1, ..., X_n$), for $n \in \{3, 4, 5, 6\}$. To make the enumeration feasible, we assume a causal order $X_1 \rightarrow ... \rightarrow X_n$ and then enumerate all $2^{n(n-1)/2}$ possible graphs by randomizing all possible causal edges between the observed variables. For each of these possible graphs, we compute the exact sparsity (number of non-zero entries) for all $n+1$ possible valid CICA solutions:
>
> The results demonstrate a clear sparsity gap. We report the total sum of non-zero entries for each of the $n+1$ solutions (sorted from sparsest to densest) aggregated across all enumerated graphs:
>
> |  n   | #graphs | Sum0 (True) |  Sum1  |  Sum2  |  Sum3  |  Sum4  |  Sum5  |  Sum6  | #Average Increased Non-zero Entries |
> | :--: | :-----: | :---------: | :----: | :----: | :----: | :----: | :----: | :----: | :--------------------------------: |
> |  3   |    8    |     36      |   43   |   50   |   59   |        |        |        |               0.875                |
> |  4   |   64    |     448     |  531   |  602   |  682   |  809   |        |        |               1.297                |
> |  5   |  1024   |    10240    | 12015  | 13368  | 14857  | 16802  | 19758  |        |               1.733                |
> |  6   |  32768  |   442368    | 513675 | 563799 | 619008 | 685666 | 769179 | 895521 |               2.176                |
>
> This comprehensive enumeration provides clear evidence for the gap between the true (sparsest) solution and the second sparsest alternative solution. For example, when $n=6$, the number of non-zero entries increases 2.176 per graph (the average number of edges between observed variables in the 6-node graph is 7.5). This confirms that the sparsity gap is a structural and generic property, and provides practical stability and robustness guarantees for our method in finding the true causal graph.
>
> Once again, we are grateful for the reviewer's valuable comments, and sincerely hope that you will find your suggestions and concerns well addressed by the response above and the relevant sections in the paper. Your further feedback would be appreciated, and we hope for the opportunity to respond to it.

---

### Official Review · Reviewer_K2L3 · 2025-10-28

**Soundness:** 3
**Presentation:** 3
**Contribution:** 1
**Rating:** 2
**Confidence:** 3

**Summary:**

This paper introduces Conditional Independent Component Analysis (CICA), a novel method for identifying latent variables and causal structure in linear non-Gaussian acyclic models. Unlike existing approaches that require "purity" assumptions (where latent variables must have enough pure children), CICA uses two-sided projections to extract components that are conditionally independent given latent variables. The authors prove that under some conditions, the sparsest CICA solution combined with appropriate row permutation recovers the full causal structure and demonstrate their approach on synthetic data and a personality psychology dataset. The method offers tractable optimization through rank-deficiency constraints rather than higher-order statistics, achieving superior performance particularly in "fully impure" scenarios where existing methods struggle.

**Strengths:**

* The paper presents a reasonable extension of prior work by generalizing from one-sided to two-sided projections in the ICA framework, addressing the one-sided limitations.
* The theoretical analysis and identifiability results are rigorous as far as I can tell, with clear conditions under which full causal structure recovery is guaranteed.
* Unlike overcomplete ICA methods that require EM algorithms and higher-order cumulants, CICA provides tractable optimization criteria through rank-deficiency constraints, making the approach computationally feasible while maintaining theoretical guarantees.
* While the material is mathematically dense, the theoretical developments are presented clearly with good use of examples.

**Weaknesses:**

* The paper seems to be mostly driven by theoretical feasibility rather than a practical necessity of an improved approach. It is unclear to me whether the setting and the proposed method have any practical relevance. The assumptions in Equation (1), linearity and non-Gaussianity seem prohibitive for most problems. While those are common assumptions in the Causality literature, the authors would have to provide strong evidence that their method is useful in relevant problems despite those strong assumptions.
* The real-world experiment on personality data fails to validate the method convincingly. The discovered causal relationships lack corroboration with psychological literature and rely solely on intuitive plausibility (e.g., "difficulty understanding causes lack of interest"), with many arrows equally plausible in reverse. The authors provide no justification for why personality data would exhibit the specific dependency structures that necessitate their two-sided projection approach over existing methods like Dong et al. Critical technical assumptions (linearity, non-Gaussianity) are never verified for the data, and the fundamental assumption that latent personality factors follow a DAG structure is also not well-justified. The real-world results appear to be post-hoc interpretations rather than meaningful validation.

**Minor:**

* L121: LiNGAM isn't defined.
* Figure 2 contains little relevant information. That space might be better spent making a visual overview of the method.

**Questions:**

* Is there any concrete, non-theoretical evidence, that extending the method to two-sided projections helps solve relevant problems for which previous approaches fail?
* How does your method relate to Causal Component Analysis [1]?

References:

[1] Wendong, Liang, et al. Causal component analysis. NeurIPS 2023

---

> ### Author Response · Authors · 2025-11-21
> **Response to Reviewer K2L3 - Part 1/4 (W1)**
>
> We sincerely appreciate the reviewer's helpful feedback. Please see below for our response.
>
> ---
>
> **(W1)** The reviewer has concerns that linearity and non-Gaussianity are prohibitive for most problems.
>
> **R:** We appreciate the reviewer for their feedback and attention on real applications. Below, we provide a point-by-point response.
>
> **Regarding Linearity and non-Gaussianity assumption:**
>
> We first clarify that the Linearity and non-Gaussianity assumptions are not prohibitive. Rather, they represent the standard theoretical guarantee for causal discovery and serve as the necessary starting point for establishing provable identifiability. We address the reviewer's concern through the following three logical levels:
>
> **1. Linearity:** To provide a rigorous identification conclusion for a statistical model, essential parameterization assumptions are indispensable. Linear assumptions represent an important direction for model identification. Under linear assumptions, we can derive clean and elegant identification guarantees without requiring auxiliary information (needed in the non-linear scenario [1, 2]). The linearity assumption is a principled simplification used as the foundation for countless classic methods (e.g., linear regression, PCA, CCA, LDA) across statistics [3, 4], economics [5, 6], machine learning [7, 8, 9] and blind source separation [10, 11]. In almost all fields of statistical learning, understanding the linear case in principle is a prerequisite for solving the much more complex nonlinear case. Empirically, linear models are robust approximations of complex systems in many fields. Many economic works assume a linear relationship and still have meaningful results [6]. Many nonlinear relationships can be well approximated as linear after appropriate transformations (e.g., log transformations in biological applications [12]).
>
> **2. Non-Gaussianity:** Non-Gaussianity is also not a prohibitive assumption. Non-Gaussian distributions are ubiquitous. Non-Gaussian data are far more prevalent than Gaussian ones in the real world [13]. Besides, as established by the classical Darmois-Skitovich Theorem and ICA theory [10, 11], linear models with Gaussian noise are fundamentally unidentifiable due to rotational symmetry. Introducing non-Gaussianity provably breaks this symmetry, allowing for the unique recovery of causal direction from observational data [14]. Therefore, non-Gaussianity is a commonly used assumption in statistics and machine learning, particularly the basic assumption ground for ICA and blind source separation. For causal discovery, non-Gaussianity has demonstrated usefulness in many real-world scenarios (e.g., real-world physics and sociology data [15]).
>
> ---
> References:
>
> [1] Aapo Hyvarinen, Hiroaki Sasaki, Richard Turner. Nonlinear ICA using auxiliary variables and generalized contrastive learning. AISTATS 2019.
>
> [2] Yujia Zheng, Ignavier Ng, Kun Zhang. On the identifiability of nonlinear ICA: Sparsity and beyond. NeurIPS 2022.
>
> [3] Emmanuel J. Candes, Benjamin Recht. Exact matrix completion via convex optimization. Communications of the ACM (2012).
>
> [4] Emmanuel J. Candes Terence Tao. Near Optimal Signal Recovery From Random Projections: Universal Encoding Strategies? IEEE transactions on information theory (2006).
>
> [5] Imbens, Guido W., and Joshua D. Angrist. Identification and Estimation of Local Average Treatment Effects. Econometrica (1994).
>
> [6] Wooldridge, Jeffrey M. Econometric analysis of cross section and panel data. MIT Press, 2010.
>
> [7] Emmanuel J. Candes, Xiaodong Li, Yi Ma, John Wright. Robust Principal Component Analysis? Journal of the ACM (2011).
>
> [8] John Wright, Arvind Ganesh, Shankar Rao, Yigang Peng, Yi Ma. Robust Principal Component Analysis: Exact Recovery of Corrupted Low-Rank Matrices via Convex Optimization. NeurIPS 2009.
>
> [9] Robust Recovery of Subspace Structures by Low-Rank Representation. Guangcan Liu, Zhouchen Lin, Shuicheng Yan, Ju Sun, Yong Yu, Yi Ma. IEEE transactions on pattern analysis and machine intelligence (2012).
>
> [10] Aapo Hyvärinen. Fast and robust fixed-point algorithms for independent component analysis. IEEE Transactions on Neural Networks 1999.
>
> [11] Aapo Hyvärinen, Oja Erkki. Independent component analysis: algorithms and applications. Neural networks (2000).
>
> [12] McDonald, John H. Handbook of biological statistics (2009).
>
> [13] Peter Spirtes, Kun Zhang. Causal discovery and inference: concepts and recent methodological advances. Applied informatics (2016).
>
> [14] Shimizu Shohei, Takanori Inazumi, Yasuhiro Sogawa, Aapo Hyvarinen, Yoshinobu Kawahara, Takashi Washio, Patrik O. Hoyer, Kenneth Bollen. DirectLiNGAM: A direct method for learning a linear non-Gaussian structural equation model. Journal of Machine Learning Research (2011).
>
> [15] Shohei Shimizu, Patrik O Hoyer, Aapo Hyvärinen, Antti Kerminen. A linear non-Gaussian acyclic model for causal discovery. Journal of Machine Learning Research (2006).

---

> ### Author Response · Authors · 2025-11-21
> **Response to Reviewer K2L3 - Part 2/4 (W1 & Q1 & W3 & W4)**
>
> We sincerely appreciate the reviewer's helpful feedback. Please see below for our response.
>
> ---
>
> **(W1 & Q1)** The reviewer has concerns that linearity and non-Gaussianity are prohibitive for most problems.
>
> **3. Extension to non-linear cases:** Our current use of "rank deficiency" (Lemma 4) is the tractable proxy for optimizing CICA in the linear data. In particular, the idea of characterizing conditional independent given latent variables using rank-deficiency constraint that go beyond the linear setting. To see this, related results in the linear-Gaussian setting [16] and in discrete models [17, 18] have shown how ranks can characterize graphical structural properties across different data types. Although the specific techniques for characterizing and estimating such rank constraints vary, these works share the same underlying intuition: use a testable quantity on the observed data to capture the minimal separator between two variable sets in the causal graph. Therefore, we can replace our current rank-deficiency proxy on the covariance matrix with tensor-rank or matrix rank conditions on the contingency tables [17, 18]. Of course, as a trade-off, stronger assumptions (e.g., full-rank or completeness conditions in [17], auxiliary variables [1], structural sparsity [2]) must also be introduced.
>
> **Regarding Non-theoretical Evidence for Motivation:**
>
> Finally, in light of your comment, we have added a motivating example:
>
> - Let **$L$** be a latent factor, e.g., General Intelligence and Personality.
> - Let **$X_1$** be "Hours Studied", **$X_2$** be "Exam Score", **$X_3$** be "Job Offer Salary".
>
> A highly plausible real-world causal structure is:
>
> - General Intelligence and Personality ($L$) confounds all three observed variables.
> - Hours Studied ($X_1$) also causally affects Exam Score ($X_2$).
> - Exam Score ($X_2$) in turn causally affects Job Offer Salary ($X_3$).
>
> This is an impure structure, as all observed variables are confounded and causally linked. In this realistic scenario, most existing state-of-the-art methods (like GIN or TIN we mentioned) provably fail as they cannot find the non-trivial independence patterns needed for identification. This means using existing discovery tools would be unable to distinguish this true structure ($X_1 \rightarrow X_2 \rightarrow X_3$) from an alternative one (e.g., $X_1 \rightarrow X_3 \rightarrow X_2$), potentially concluding that Job Offer Salary ($X_3$) directly causes Exam Score ($X_2$). The practical necessity of our CICA-based framework is to solve this exact identification problem. Our two-sided projection approach is the specific tool required to break this ambiguity. Hope you find this example helpful.
>
> ---
>
> **(W3, W4)** Other helpful comments, including:
>
> * **LiNGAM isn't defined:** Thank you for pointing out this mistake. LiNGAM is an abbreviation for the linear, non-Gaussian acyclic model. We have fixed it in the updated submission.
> * **Little relevant information contained in Figure 2:** Thanks for your advice. We have moved it to the Related Work Section in the Appendix. The core purpose of this figure is to provide a "theoretical evolution" diagram that visually situates our CICA principle in the context of prior work and conceptual support for our motivation. It clearly contrasts the one-sided projections ($\omega^{\top}\mathbf{Y} \perp \mathbf{Z}$) relied upon by GIN and TIN with the more informative two-sided conditional projections ($\omega_{1}^{\top}\mathbf{Y} \perp \omega_{2}^{\top}\mathbf{Z}|\mathbf{L}$) introduced by our CICA principle.
>
> ---
>
> References:
>
> [1] Aapo Hyvarinen, Hiroaki Sasaki, Richard Turner. Nonlinear ICA using auxiliary variables and generalized contrastive learning. AISTATS 2019.
>
> [2] Yujia Zheng, Ignavier Ng, Kun Zhang. On the identifiability of nonlinear ICA: Sparsity and beyond. NeurIPS 2022.
>
> [17] Zhengming Chen, Ruichu Cai, Feng Xie, Jie Qiao, Anpeng Wu, Zijian Li, Zhifeng Hao, Kun Zhang. Learning Discrete Latent Variable Structures with Tensor Rank Conditions. NeurIPS 2024.
>
> [18] Lingjing Kong, Guangyi Chen, Biwei Huang, Eric P. Xing, Yuejie Chi, Kun Zhang. Learning Discrete Concepts in Latent Hierarchical Models. NeurIPS 2024.

---

> ### Author Response · Authors · 2025-11-21
> **Response to Reviewer K2L3 - Part 3/4 (W2)**
>
> **(W2)** The reviewer argues the real-world experiment on personality data fails to validate the method convincingly.
>
> **R:** Thanks for your question. Based on the reviewer's concern, we provide another real-world experiment on the Teacher’s Burnout Study. It is a study conducted by Barbara Byrne to investigate the impact of organizational and personality on three facets (emotional exhaustion, depersonalization, and personal accomplishment) of burnout in full-time elementary teachers. The data set consists of 32 observed variables with 599 samples. Here, we use the model given by the domain expert in [19] as a reference. The measurement model learning results of our method and RLCD [20] are:
>
> | Ours                                                        | Correctness | RLCD                                                        | Correctness |
> | ----------------------------------------------------------- | :--------: | ----------------------------------------------------------- | :--------: |
> | $L_1 \sim\{R A_1, R A_2\}$                                  | $\surd$     | $L_1 \sim\{RA_1, RA_2, RC_1,EE_1\}$                         | $\times$    |
> | $L_2 \sim \{EE_1, EE_2, EE_3\}$                             | $\surd$     | $L_2\sim\{EE_2,EE_3\}$                                      | $\times$    |
> | $L_3 \sim\{D P_1, D P_2\}$                                  | $\surd$     | $L_3 \sim\{D P_1, D P_2\}$                                  | $\surd$     |
> | $L_4 \sim\{R C_1, R C_2, W O_1, W O_2\}$                    | $\surd$     | $L_4 \sim\{RC_2, WO_1, WO_2\}$                              | $\times$    |
> | $L_5 \sim\{S E_1, SE_2, SE_3\}$                             | $\surd$     | $L_5 \sim\{SE_1, SE_2, SE_3\}$                              | $\surd$     |
> | $L_6 \sim\{P A_1, P A_2, P A_3\}$                           | $\surd$     | $L_6 \sim\{PA_1, PA_2, PA_3\}$                              | $\surd$     |
> | $L_7 \sim\{C C_1, C C_2, C C_3, C C_4\}$                    | $\surd$     | $L_7 \sim\{C C_1, C C_2, C C_3, C C_4\}$                    | $\surd$     |
> | $L_8 \sim \{D M_1, D M_2, S S_1, S S_2\}$                   | $\times$    | $L_8 \sim \{D M_1, D M_2, SS_1,SS_2\}$                      | $\times$    |
> | $L_{9} \sim\{E L C_1, E L C_2, E L C_3, E L C_4, E L C_5\}$ | $\surd$     | $L_{9} \sim\{E L C_1, E L C_2, E L C_3, E L C_4, E L C_5\}$ | $\surd$     |
>
> Here, we rename the names of the latent variables in RLCD's output for easier comparison. Compared to the reference model given in [19], our method merges DM and SS into one latent factor and keeps other clusters correctly identified. Notice that [20] arises more errors in the clustering step ($L_1, L_2, L_4$). A possible reason is that $L_1$ only have two measurement variables and are incapable of correctly locating by their method. These results further verify the efficacy of our algorithm. Besides, the structural model learning results (causal graph on latent variables) of our method and RLCD are:
>
> | Ours                   | Correctness | RLCD                   | Correctness |
> | ---------------------- | :-----------: | ---------------------- | :---------: |
> | $RA \rightarrow PA$    | $\surd$     | $RA \rightarrow DM/SS$ | $\times$    |
> | $EE \rightarrow SE$    | $\surd$     | $SE \rightarrow DP$    | $\surd$     |
> | $SE \rightarrow ELC$   | $\surd$     | $SE \rightarrow PA$    | $\surd$     |
> | $DM/SS \rightarrow SE$ | $\surd$     | $DP \rightarrow PA$    | $\surd$     |
> | $RC \rightarrow DP $   | $\times$    | $DP\rightarrow CC$     | $\times$    |
> | $CC \rightarrow EE$    | $\surd$     | $RC \rightarrow DP $   | $\times$    |
> | $ELC \rightarrow PA$   | $\times$    | $RC\rightarrow SE$     | $\times$    |
> | $ELC \rightarrow DP$   | $\times$    | $RC \rightarrow ELC$   | $\surd$     |
> | $RC\rightarrow EE$     | $\surd$     | $RC\rightarrow RA$     | $\times$    |
> | $EE \rightarrow ELC$   | $\times$    |                        |             |
>
> The F1 score of our results is 0.522. In contrast, RLCD obtains 0.364. In the output results of the RLCD, most of the edges connected to RC are incorrect. The possible reason is that some latent factors can not be discovered correctly, which further causes some unobserved confounding between latent variables. Note that the previous method can not identify $SE \rightarrow ELC$ in principle, as they form an impure structure on latent variables. By solving CICA on SE and ELC using their observed descendants, our method can recover the causal direction $SE \rightarrow ELC$, which supports the necessitate of introducing two-sided projection.
>
> ---
> References:
>
> [19] Barbara M Byrne. Structural Equation Modeling With AMOS: Basic Concepts, Applications, and Programming. Routledge, 2016.
>
> [20] Xinshuai Dong, Biwei Huang, Ignavier Ng, Xiangchen Song, Yujia Zheng, Songyao Jin, Roberto Legaspi, Peter Spirtes, Kun Zhang. A Versatile Causal Discovery Framework to Allow Causally-Related Hidden Variables. ICLR 2024.

---

> ### Author Response · Authors · 2025-11-21
> **Response to Reviewer K2L3 - Part 4/4 (Q2)**
>
> **(Q2)** The reviewer requests a discussion between Causal Component Analysis [21] and our paper.
>
> **R:** [21] is a nice work which introduces an intermediate problem between independent component analysis and causal representation learning: recover causally related latent variables $\mathbf{Z}$ from non-linear mixtures $\mathbf{X} = f(\mathbf{Z})$ when the causal graph $G$ among the latent variables $\mathbf{Z}$ is assumed to be known.
>
> Please also let us highlight that, despite having similar names, our work and Causal Component Analysis (CauCA) [21] address fundamentally different questions, which we summarize in the following table:
>
> |                   |                   Ours           |                            CauCA                             |
> | :--------------------: | ----------------------------------------------------------- | ------------------------------------------------------------ |
> |       Goal        |   Causal discovery based on the solution of proposed CICA    | Learn the unknown unmixing function $f$ and the causal mechanisms |
> |       Data        |                A single observational dataset                |               Multiple interventional datasets               |
> |   Causal Graph    |                           Unknown                            |                            Known                             |
> | Main contribution | 1. Introducing a novel CICA principle, that extracts components that are conditionally independent given latent variables. 2. Establishing an identification theory and an estimation algorithm that recover the underlying causal structure based on the sparsest CICA solutions. | 1. Providing identifiability proofs that the unmixing function $f$ is identifiable up to element-wise scaling if one has access to a perfect stochastic intervention on every latent variable. 2. Proposing a likelihood-based estimation procedure using normalizing flows to learn the non-linear unmixing function and the causal mechanisms. |
>
> We have added this clarification to our related work section.
>
> ---
>
> [21] Liang Wendong, Armin Kekic, Julius von Kügelgen, Simon Buchholz, Michel Besserve, Luigi Gresele, Bernhard Schölkopf. Causal component analysis. NeurIPS 2023.

---

### Official Review · Reviewer_479s · 2025-11-01

**Soundness:** 3
**Presentation:** 3
**Contribution:** 3
**Rating:** 6
**Confidence:** 3

**Summary:**

The paper tackles causal discovery with latent confounders in settings where the standard “pure measurement” assumption breaks down—i.e., observed variables may also be causally linked, making many existing methods unreliable. The authors introduce Conditional Independent Component Analysis (CICA), which seeks two-sided linear projections whose outputs are conditionally independent given the latent factors. They show that CICA admits a characterization through rank-deficiency constraints and that, among all CICA-consistent solutions, the one aligned with the true causal model is the sparsest. Building on this, they design an algorithm that recursively recovers a latent-augmented DAG under mild structural assumptions. On synthetic data with non-pure, inter-connected observed variables, the method outperforms recent latent-causal baselines.

**Strengths:**

* The motivation is clear: the paper targets exactly the regime where measurement purity is violated because observed variables also have edges, a setting in which many latent-causal methods break down.
* The theoretical contribution is novel and well argued: introducing two-sided conditional independence provides the additional degrees of freedom needed to distinguish structures that are otherwise observationally equivalent, and the insight that the “correct” CICA solution is the sparsest one is both elegant and convincing.
* Experiments on several deliberately hard, impure scenarios demonstrate consistent advantages over existing baselines.

**Weaknesses:**

* The main limitation—admittedly a common one in this area—is that all experiments are synthetic and closely matched to the paper’s own generative assumptions. There is no semi-real or real benchmark (e.g., latent-variable causal discovery datasets, or at least mixed real+synthetic scenarios) to demonstrate robustness under model misspecification.
* Several recent works with very similar problem settings are not discussed. For instance, the ICLR’24 paper “Causal Structure Recovery with Latent Variables under Milder Distributional and Graphical Assumptions” is highly relevant and should be compared.

**Questions:**

* How robust is CICA to departures from the linear-mixing-with-independent-noise assumption? In particular, if the observables are produced by mildly nonlinear transformations of the latents, do the proposed rank-deficiency constraints still identify meaningful structure, or do they degenerate to spurious CICA solutions?
* The method largely assumes that p (the number of conditioning latents) is known in advance. How should p be chosen in practice from data? Is there a concrete, data-driven procedure—e.g., progressively increasing p while monitoring rank deficiency or CI test statistics—to determine an appropriate order?

---

> ### Author Response · Authors · 2025-11-21
> **Response to Reviewer 479s - Part 1/2 (W1 & W2)**
>
> We sincerely appreciate the reviewer's constructive comments and helpful feedback. Please see below for our response.
>
> ---
>
> **(W1)** The reviewer has concerns that the paper has no semi-real or real benchmark.
>
> **R:** Thank you for this point, we would like to clarify that in fact, we did, **include a detailed real-world experiment in Appendix C.2 (starting on page 26)**. Due to the space constraints of the main paper, it is placed in the appendix, making it less visible. In this experiment, we applied our CICA-LINGAM algorithm to the "Big Five" personality dataset, a psychology benchmark comprising approximately 20,000 valid samples. In light of your comment, we have made this point explicit in the update submission. Hope you find the updated version clearer on this point.
>
> In this analysis, we construct a measurement model, identifying latent factors corresponding to personality dimensions. Then we discover novel and interpretable within-cluster causal relationships among the observed items (e.g., "difficulty in understanding" $O_2$ $\rightarrow$ "lack of interest" $O_4$). Finally, we recover the structural causal model among the latent variables (e.g., "Abstract cognitive ability" $L_1$ $\rightarrow$ "creative potential" $L_2$). This experiment effectively demonstrates our method's ability to handle complex, real-world data, particularly in identifying the impure structures that existing methods often fail to capture. Besides, we provide another real-world experiment on the Teacher’s Burnout Study. Please kindly see our response to Reviewer K2L3 (W2) and Appendix C.2 for more details.
>
> ---
>
> **(W2)** The reviewer requests a discussion between [1] and our paper.
>
> **R:** We sincerely thank the reviewer for pointing out this relevant work. [1] is a nice work focusing on the same problem as our work. We have added a detailed discussion of it in our Related Work section. In what follows, we provide a summary comparison.
>
> While both our paper and [1] aim to recover causal structures with latent variables by relaxing strong assumptions like purity assumptions, we respectfully clarify that our CICA framework is fundamentally different and addresses a more general and challenging class of causal structures that [1] can not solve.
>
> 1. **Difference in methodological tools:**
>
>    In [1]'s most relevant case (Case II, non-Gaussian), its identification theory is based on Lemma 3 to identify "pseudo-pure pairs", which involves finding a linear combination of variables that is independent of a single variable (e.g., $L(O_1, O_2, O_3) \perp O_1$). This is a form of the "one-sided projection" ($\omega^{\top}\mathbf{Y} \perp \mathbf{Z}$) discussed in our paper.
>
>    Our paper's central motivation (Section 3.1) is that this entire class of "one-sided projection" tools (including GIN, TIN, and the one used by Li et al. (2024)) is provably insufficient for the "fully impure" structures in our Figure 1. Our CICA principle ($Z_i \perp Z_j | \mathbf{L}$) is introduced to obtain a more powerful "two-sided projection" ($\omega_{1}^{\top}\mathbf{Y} \perp \omega_{2}^{\top}\mathbf{Z}$) and find additional identifiable traces for causal structure.
>
> 2. **Difference in structural limitations:**
>
>    The identifiability results of [1] are based on its Assumption 1, which requires that each latent variable has at least one generalized pure pair as children. While relaxing the full purity assumption, its framework still relies on searching for "generalized pure pairs" by testing tetrad constraints as anchors. In our motivating example (Figure 1(a), $L$ confounds $X_1, X_2, X_3$ and $X_1 \rightarrow X_2 \rightarrow X_3$) is a "fully impure" structure. Here, $L$ has no generalized pure pairs. As a result, the identification procedure of [1] cannot be started.
>
>    Our paper solves "fully impure" structures. Our key theoretical contribution (Lemma 8) proves that the true causal structure can still be identified from the sparsest CICA solution even in the absence of "generalized pure pairs". This further demonstrates that these challenging impure structures fall outside the scope of [1], highlighting the distinct and necessary contribution of our CICA framework.
>
> ---
>
> References:
> [1] Xiu-Chuan Li, Kun Zhang, Tongliang Liu. Causal Structure Recovery with Latent Variables under Milder Distributional and Graphical Assumptions. ICLR 2024.

---

> ### Author Response · Authors · 2025-11-21
> **Response to Reviewer 479s - Part 2/2 (Q1 & Q2)**
>
> We sincerely appreciate the reviewer's constructive comments and helpful feedback. Please see below for our response.
>
> ---
>
> **(Q1)** The reviewer is curious about whether CICA and the proposed rank-deficiency constraints degenerate if the data follow a nonlinear generation process.
>
> **R:** Thanks for your insightful question. Our current framework only applies to the linear non-Gaussian setting. Therefore, if the data generation process involves strong nonlinearities, the resulting CICA solutions would likely be spurious. That said, we believe the key intuition we developed behind the specific technique can have promising future results. In particular, the idea of characterizing conditional independent given latent variables using a rank-deficiency constraint goes beyond the linear setting.
>
> To see this, related results in the linear-Gaussian setting [2] and in discrete models [3, 4] have shown how ranks can characterize graphical structural properties across different data types. Although the specific techniques for characterizing and estimating such rank constraints vary, these works share the same underlying intuition: using a testable quantity on the observed data to capture the minimal separator between two variable sets in the causal graph. Therefore, we can replace our current rank-deficiency proxy on the covariance matrix with tensor-rank or matrix rank conditions on the contingency tables [3, 4]. Of course, as a trade-off, stronger assumptions such as full-rank or completeness conditions [3] must also be introduced.
>
> From another view of the linear non-Gaussian method's original history, its "regression residual independence" intuition has motivated many useful results like (non-linear) additive noise model (ANM) [5], post non-linear (PNL) model [6], and so on. We hope ours can as well in the latent variable discovery.
>
> Thanks again for this insightful comment!
>
> ---
>
> **(Q2)** The reviewer raises a question about how to determine the value of $p_{min}(\mathbf{S})$.
>
> **R:** We sincerely thank the reviewer for this insightful question. The reviewer is correct that our CICA optimization criteria (Lemma 4 and 5) require the order $p = p_{min}(\mathbf{S})$ to be known for a given set of variables $\mathbf{S}$. However, we would like to clarify that $p$ is **not** assumed in advance in causal discovery. Instead, the value of $p$ is **determined directly from the data** during the clustering phase of our CICA-LINGAM algorithm (Algorithm 1, line 3), as emphasized in lines 258-260.
>
> Specifically, as we have clarified in our **revised Lemma 11**, our algorithm first identifies each causal cluster $\mathbf{S}$ and determines the number of latent parents for $\mathbf{S}$ by testing GIN conditions, meanwhile. As the number of common latent parents $|LPa(\mathbf{S})|$ equals $p_{min}(\mathbf{S})$, we naturally obtain the value of $p_{min}(\mathbf{S})$ in a data-driven way.
>
> We have incorporated the revised, more explicit Lemma 11 and added this explanation to Section 3.4 to make this data-driven procedure unambiguous in the new version.
>
> ---
>
> References:
>
> [2] Seth Sullivant, Kelli Talaska, Jan Draisma. Trek separation for Gaussian graphical models. Annals of statistics, 38(3), 1665-1685.
>
> [3] Zhengming Chen, Ruichu Cai, Feng Xie, Jie Qiao, Anpeng Wu, Zijian Li, Zhifeng Hao, Kun Zhang. Learning Discrete Latent Variable Structures with Tensor Rank Conditions. NeurIPS 2024.
>
> [4] Lingjing Kong, Guangyi Chen, Biwei Huang, Eric P. Xing, Yuejie Chi, Kun Zhang. Learning Discrete Concepts in Latent Hierarchical Models. NeurIPS 2024.
>
> [5] Patrik O. Hoyer, Dominik Janzing, Joris M. Mooij, Jonas Peters, Bernhard Schölkopf. Nonlinear causal discovery with additive noise models. NeurIPS 2008.
>
> [6] Kun Zhang, Aapo Hyvarinen. On the identifiability of the post-nonlinear causal model. UAI 2009.

---

### Author Response · Authors · 2025-12-03
**Summary of the Discussion for the Area Chair**

Dear Area Chair,

Thanks for handling our paper! The following provides a summary of the reviews, our responses, and discussions to facilitate your final decision.

---

First, we are grateful that the reviewers recognized our work's soundness, novelty, and contribution. E.g.,

- "the theoretical contribution is novel and well argued" (Reviewer 479s)
- "the theoretical analysis and identifiability results are rigorous as far as I can tell, with clear conditions ..." (Reviewer K2L3)
- "... it removes the pure-child assumption, which is required by most existing causal discovery methods dealing with latent variables ... This represents a significant contribution to the causal discovery research community" (Reviewer Lp3A)
- "A novel identifiability principle is proposed." (Reviewer 8zWU)

We then provide a brief overview of the reviewers' main concerns and our responses.

---

**Reviewer 479s** (Initial rating: 6): The reviewer raises concerns regarding the absence of a real benchmark and the relationship of our work to [[L+24]](https://openreview.net/pdf?id=MukGKGtgnr).

**Response summary:**

* We clarified that our paper **actually includes** a comprehensive real-world experiment (see Appendix C.2), and we additionally provide another real-world evaluation on the Teacher’s Burnout Study.
* We provide a detailed discussion distinguishing our methodological tools and structural assumptions from those in [L+24]. We respectfully emphasize that our approach is designed for a broader class of causal structures, making the two methods fundamentally different in scope.

---

**Reviewer K2L3** (Initial rating: 2): The reviewer expresses concerns that the linearity and non-Gaussianity assumptions may be restrictive. The reviewer also requests concrete evidence for the necessity of two-sided projections and argues that the real-world experiment on personality data does not provide sufficiently convincing validation.

We respectfully note that although the reviewer acknowledged the contribution and novelty of our work, **the score of 2 was assigned without sufficient evidence to support such a low rating. Several concerns on assumptions are not applicable to our setting**. We clarified these points in detail in our response, and **we hope the AC will consider our explanations when evaluating the paper**.

**Response summary:**

* We clarified that the linearity assumption is a principled simplification used across statistics, economics, and machine learning. Further, non-Gaussianity serves as the foundation behind ICA and blind source separation, and has repeatedly demonstrated its practical value in causal discovery applications.
* We added a motivating example in the rebuttal to illustrate the necessity of two-sided projections.
* We provided another real-world experiment on the Teacher’s Burnout Study and used domain expert knowledge as a reference. Our method achieves a clearly higher F1 score than the baseline. Some edges recovered by our method, which previous approaches cannot identify in principle, further support the necessity of introducing two-sided projections.

---

**Reviewer Lp3A** (Initial rating: 6): The reviewer asks whether sparsity is an inherent property or merely a modeling choice, and requests conditions under which sparsity aligns the true causal structure, as well as how denser alternative solutions deviate from the sparsest one.

**Response summary:**

* We clarified that sparsity is a theoretically established property rather than a modeling choice, and specified when the sparsest solution coincides with the true causal structure (Lemma 8).
* We clarified a precise graphical characterization of the gap between alternative solutions and the sparsest one. Supplemented simulations quantitatively confirm that the gap is clear.

---

**Reviewer 8zWU** (Initial rating: 4; promised to lift the rating to 6 after two rounds of discussion): The reviewer requests a simpler case to prove the effectiveness of our method and achieve better metrics, wonders about the gap between the baselines’ results in our paper and that reported in their original papers, and suggests providing details for each subprocess of the main algorithm.

**Response summary:**

* We present another synthetic experiment in our rebuttal response, highlighting the effectiveness of our approach in nearly pure settings.
* We clarified that our experiments involve highly impure structures, which are harder and fall outside the identifiability conditions assumed by many baselines. Besides, we elaborated that verifying the GIN condition becomes more challenging in impure structures, which accounts for the degraded performance of GIN-based methods observed in our experiments.
* We expanded Appendix B in our revised manuscript to include step-by-step pseudocode.
* We also respectfully noted that the reviewer explicitly stated: **"After reading the author response, I decide to raise my score to 6"**.

---

### Meta-Review · Area_Chair_CfR4 · 2026-01-07

**Summary:**

Reviewers agree that this paper proposes a novel method for causal discovery. However, they also find experimental results not satisfying enough for providing very convincing practical benefits. Authors are encouraged to update the paper with new results in the rebuttal and further strengthen the discussion on the practical implications.

**Reviewer Concerns:**

In the rebuttal, the authors provided more experimental results to make the real world experiments stronger and clarifies the causes of the unsatisfying results. To answer the reviewers' questions, they also compare their work with more related literature and explained why their work is in a different scope or setting with previous work. I regard these concerns resolved.

I think what is still outstanding is, this is a theory-driven work and practitioners may find it less appealing as assumptions may sound strong (even though standard in the causal literature) and experimental results not satisfying. So more discussion on the practical implications would be helpful.

**Reviewer Scores:**

Reviewer 8zWU mentioned that after the discussion, they will raise the score to 6.

Given the rebuttal is fairly clear to me, I predict Reviewer K2L3 may raise the score to 4.

The other two reviewers may maintain their scores.

---

### Decision · Program_Chairs · 2026-01-26

Accept (Poster)